# Do You Really Need Public Data?
# Surrogate Public Data for Differential Privacy on Tabular Data

**Shlomi Hod**[*]
Boston University

**Lucas Rosenblatt**[*]
New York University

**Julia Stoyanovich**
New York University

## Abstract

Differentially private (DP) machine learning often relies on the availability of public data for tasks like privacy-utility trade-off estimation, hyperparameter tuning, and pretraining. While public data assumptions may be reasonable in text and image data, they are less likely to hold for tabular data due to tabular data heterogeneity across domains. We propose leveraging powerful priors to address this limitation; specifically, we synthesize realistic tabular data directly from schema-level specifications – such as variable names, types, and permissible ranges – without ever accessing sensitive records. To that end, this work introduces the notion of *"surrogate" public data* – datasets generated independently of sensitive data, which consume no privacy loss budget and are constructed solely from publicly available schema or metadata. Surrogate public data are intended to encode plausible statistical assumptions (informed by publicly available information) into a dataset with many downstream uses in private mechanisms. We automate the process of generating surrogate public data with large language models (LLMs); in particular, we propose two methods: direct record generation as CSV files, and automated structural causal model (SCM) construction for sampling records. Through extensive experiments, we demonstrate that surrogate public tabular data can effectively replace traditional public data when pretraining differentially private tabular classifiers. To a lesser extent, surrogate public data are also useful for hyperparameter tuning of DP synthetic data generators, and for estimating the privacy-utility tradeoff.

## 1 Introduction

Differential privacy (DP) is a mathematical framework for protecting individuals' privacy in statistical analysis and machine learning [38], and was deployed in multiple recent high-stakes releases and systems [3, 55, 83, 21, 113, 40] (see [35] for a more complete list). It is common in the design of differentially private algorithms to assume access to a *relevant public dataset* that can guide hyperparameter tuning, pretraining, or performance improving mechanisms [11, 13, 71, 123]. Executing these tasks with *sensitive* data would require an additional allocation of the privacy budget, resulting in weaker overall privacy guarantees or reduced utility. However, using this assumed *public* data in a private mechanism avoids additional privacy budget consumption. This leads to the following informal definition of *public* data in our work:

**Public Data (informal)**

A dataset is considered *public* if a computation taking it as input does not consume privacy loss budget with respect to any fixed private, sensitive dataset.

---

[*]Equal Contribution.

39th Conference on Neural Information Processing Systems (NeurIPS 2025) Track on Datasets and Benchmarks.

For text and image domains, assuming public data availability is often reasonable: public image collections or large-scale textual corpora are readily available, and it has been shown that even out-of-distribution data can serve as a valuable prior in these contexts, whether through pretraining or foundation models [85, 42]. However, this assumption does not often hold in a tabular data setting. Tabular data is heterogeneous, high-dimensional, subject to strict privacy or legal restrictions, and has few universal priors [84]. In many real-world domains like healthcare, finance, and government administration, tabular data encodes sensitive information that drives *high-stakes decisions*. It is thus rare to find truly public, non-sensitive samples with sufficient alignment to a private distribution to be used for private hyperparameter tuning or pretraining.

Nevertheless, recent theoretical insights confirm that if a public dataset is "close enough" to a sensitive data distribution, then private learning can still achieve strong utility, even when the public and private datasets are not perfectly matched [11]. In practice, however, identifying or constructing such a surrogate is often far from trivial. Real-world deployments of differentially private methods face numerous hurdles related to data availability [30, 31]. As an example, a recent release of Israel's Live Birth Registry [55] underscores the challenges of obtaining an end-to-end differential privacy guarantee.

Public data served two purposes for Hod and Canetti [55]: it helped constrain the hyperparameter space within a computationally locked-down enclave environment, and it enabled the estimation of the privacy-utility trade-off when allocating privacy budget. Yet, in general, sensitive datasets (e.g., birth records) are not readily available as public data. Hod and Canetti [55] reported finding only one open-access birth dataset worldwide (in the U.S.); without it, estimating the necessary parameter settings for their release would have been significantly more challenging.

A recent practical guide for differentially private machine learning recommends that "the simplest approach, when possible, is to do all model architecture search and hyperparameter tuning on a proxy public dataset (with a distribution similar to the private data), and only use the private training dataset to train the final DP model" [89].

These two examples highlight a fundamental challenge; many differentially private algorithms require informed decisions *a priori* that,

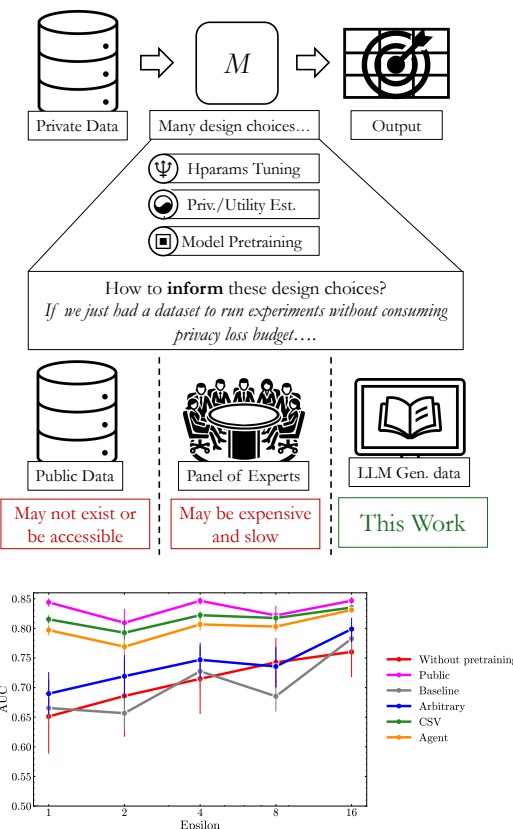

Figure 1: **(Top)** The premise of this work: Can we utilize LLMs to generate surrogate public data to solve DP auxiliary tasks? **(Bottom)** The answer is **yes**; for example, pretraining on *surrogate* public data generated through various LLM based methods (green, orange) nearly matches the performance of pretraining on regular public data (magenta), and outperforms no pretraining (red), or pretraining on baselines (gray, blue). Results on the EDAD dataset.

ideally, do not consume extra privacy budget. This leads us to consider a class of *DP auxiliary tasks*, which we define informally as:

---

**Differential Privacy Auxiliary Task**

A *differential privacy auxiliary task*, with respect to a differentially private mechanism for conducting an analysis of interest, is a required decision or procedure for execution. The auxiliary task may or may not incur privacy loss. Examples include hyperparameter tuning, setting $\varepsilon$, mechanism initialization, model selection, etc.

---

Motivated by the example of Hod and Canetti [55] and the recommendation of Ponomareva et al. [89], we imagine a world where we could convene a panel of domain experts, and ask them to manually encode an approximate data-generating process. In the birth registry example, epidemiologists and bio-statisticians could approximate high-level relationships among the birth-related variables (e.g., premature birth correlated with infant weight), yielding a sufficiently similar distribution. From this data generating process, one could then generate "public" samples for tasks such as hyperparameter tuning, privacy-utility calibration, or model pretraining. Indeed, for many tabular settings that must accommodate strict privacy and legal constraints, we hypothesize that such an expert-driven approach could offer a practical surrogate to *traditional* public data [52].

> **Surrogate Public Data**
>
> We consider a dataset generated independently of a sensitive dataset, consuming no privacy loss budget, and based only on publicly available schema or metadata to be a *surrogate* public dataset.

Surrogate public data is positioned in contrast to *"traditional" public data*, which shares a similar generation process (often *naturally occurring* then *collected*) as the private data. Then, the main question of this paper is: how useful is *surrogate* public (1) relative to *traditional* public data, or (2) relative to the *lack* of any public data? Is, for example, *automating* the process of expert panel data generation with large language models (LLMs) a suitable surrogate?

**Analogy to image or text priors.** Unlike images or text, where universal coordinate systems (like RGB grids or token sequences) and large public data source provide reusable priors, tabular pipelines face schema multiplicity (each table defines bespoke categorical codes and column sets), weak cross-domain priors (theres no obvious analog to "edges or "syntax), and potentially scarce, restricted public samples. Our goal is then to instantiate *in-domain* tabular surrogate public data from schema/metadata only, so auxiliary DP tasks can use the data while consuming *no* additional privacy budget. Just as synthetic image priors (e.g., GAN-based pretraining) can warm-start DP-SGD for image classifiers [101], our surrogate public data can play the analogous role for tables when no matched public sample exists.

To investigate these questions, we evaluate automated data generation approaches that leverage LLMs [19, 122, 66]. LLMs are trained on enormous and diverse datasets, including vast amounts of tabular data [19, 54] as well as scientific literature [88] that captures rich structural and contextual knowledge of relationships between variables. This allows for the *direct* generation of realistic tabular records, along with the *indirect* generation of coherent, causally informed relationships among variables that can lead to the generation of reasonable tabular data. Specifically, we draw inspiration from causal and Bayesian modeling methods – DAG-based generative models, analogous to *structural causal models* or *Bayes nets* – but do not strictly rely on or guarantee correctness of any *true* causal structure. Rather, our goal is to capture plausible dependencies among variables using only schema-level metadata (such as variable names, types, allowable ranges, and domain constraints). With this approach, we attempt to bridge the gap left by the unavailability of suitable public tabular data in arbitrary settings. But how can we best utilize LLMs?

Recent work on causal modeling with LLMs suggests they can encode causal information [65] and can be used to generate data with casual structure, for example, simulating counterfactuals [22]. We take this direction as inspiration, but leave open whether it is *important* for the generated data to have a realistic *causal* structure or effects. We can use causal principles as a way to encourage, but *not* guarantee, consistent, higher-order dependencies among variables – with the hope of ultimately generating more coherent tabular datasets. We can also, of course, directly request synthetic records from the LLM. We compare these approaches for generating *surrogate* public data with much simpler baselines, such as uniform sampling or arbitrarily defined Bayesian networks over the domain. This leads us to our main contributions.

## 1.1 Contributions

**(1.) Methods for generating *surrogate* public data (Section 3).** We introduced an agent-based strategy with a black-box LLM access assumption to automatically construct a plausible structural causal model for surrogate public data generation. We also introduce a number of simpler baselines methods for comparison.

**(2.) Benchmark of DP auxiliary tasks with surrogate public data (Section 4).** Auxiliary DP tasks are part of a wider private pipeline. Consequently, evaluating the usefulness of surrogate public data

requires a careful design across the DP downstream task, datasets, baselines, comparison conditions, and aggregated metrics. In this work, we propose such a benchmark framework and provide an extensible, method-agnostic implementation.

**(3.) In-depth experimental results evaluating surrogate public data on some DP auxiliary tasks (Section 5).** We find that pretraining with LLM-generated surrogate public data can *substantially* improve differentially private classification performance; this holds true in the low dataset size regime in particular. Additionally, we show that LLM-generated surrogate public data can be useful for hyperparameter tuning of private data synthesizers. We further present a complicated story on using surrogate public data for privacy-utility tradeoff estimation (i.e. "setting the privacy budget").

The code used to generate the surrogate public data and execute all experiments is publicly available.[2] Additionally, we emphasize that the approaches to surrogate public data we will discuss are a *fallback* for settings lacking a matched traditional public sample. If a sufficiently similar traditional public dataset exists, it should generally be preferred, as it also incurs no privacy cost under Def. 2.

## 2 Preliminaries

Differential privacy (DP) ensures that the presence or absence of a single individuals data has only a limited influence on an output statistic; in other words, it restricts how much any single record can affect the outcome of an analysis. To define this, we consider two datasets $D, D' \in \mathcal{X}^n$, which are *neighboring* if they differ in at most one data entry. Let $\mathcal{X}$ denote the universe of records.

**Definition 1** (Differential Privacy [38]). *An algorithm* $\mathcal{M} : \mathcal{X}^n \to \mathbb{R}$ *satisfies* $(\varepsilon, \delta)$-*differentially private if, for every pair of neighboring datasets* $D, D' \in \mathcal{X}^n$, *and for every subset of possible outputs* $\mathcal{S} \subseteq \mathbb{R}$,

$$\Pr[\mathcal{M}(D) \in \mathcal{S}] \leq e^{\varepsilon} \Pr[\mathcal{M}(D') \in \mathcal{S}] + \delta .$$

The following definition of public data is inspired by [14].

**Definition 2** (Public Data). *A dataset* $\hat{D} \in \mathcal{X}^m$ *is public if incorporating it into any computation does not incur additional privacy loss. That is, for any sensitive dataset* $D \in \mathcal{X}^n$ *and for every* $(\varepsilon, \delta)$-*differentially private mechanism* $\mathcal{M}$, *the privacy guarantee is identical whether* $\hat{D}$ *is used or not, i.e.,* $\mathcal{M}(D, \cdot)$ *and* $\mathcal{M}(D, \hat{D})$ *both satisfy identical* $(\varepsilon, \delta)$-*differential privacy guarantees.*

## 3 Producing Public Data Surrogates

We evaluate multiple methods for generating surrogates to public data, categorizing them into baseline and LLM-based approaches. For these methods, we assume that the private data's metadata – consisting of the dataset schema and a brief description of its topic (e.g., demographics, epidemiology) – is publicly available. **All methods we introduce rely solely on this metadata.**[3] A schema provides a description of the dataset domain and structure, specifying for each variable: (1) its name, (2) a very brief description, (3) the data type (e.g., integer, string), and (4) either allowed values and their meanings for categorical columns or value ranges for continuous columns. This metadata is typically extracted from the dataset's accompanying README file or codebook (see, e.g., [69] on ICPSR). Figure 5 is an excerpt from a schema. Each LLM-based method is applied to the three models presented in Table 2. Below we briefly summarize each approach; see Appendix C for full details, and Figure 1 for an overview.

**Baselines.** We evaluate three baseline methods: `Uniform` and `Arbitrary` rely solely on the public schema; `Univariate` intentionally uses noisy one-way marginals from private data as a *non-surrogate* comparison point. The `Uniform` method samples records i.i.d. from each variables full domain, while the `Univariate` approach samples columns independently using empirical 1-way marginal distributions from the private data (this violates privacy, but serves as a competitive baseline). The `Arbitrary` method constructs a random Bayesian network (BN) over the high-dimensional domain by sampling a DAG (maximum in-degree of 5) and parameterizing each nodes conditional probability tables from symmetric Dirichlet priors; see Algorithm 1.

---

[2]`https://github.com/shlomihod/surrogate-public-data`

[3]With one exception: the *univariate* baseline, which samples directly from the sensitive data *without* correlation between variables. This method is introduced purely for comparison, and is *not* a valid public data surrogate under our working defition.

Table 1: Overview of the datasets used for evaluation.

| Dataset | Topic | Features | ✕Dims | Private Split | | | Public Split | | |
|---|---|---|---|---|---|---|---|---|---|
| | | | | Name | Size | Published | Name | Size | Published |
| ACS | Census | 7 | 116,640 | National | 23,006 | Sep 2020 | Massachusetts | 23,006 | Sep 2020 |
| EDAD | Disability | 11 | 2,188,800 | 2023 | 1,469 | Apr 2024 | 2020 | 1,469 | Apr 2022 |
| WE | Workplace | 12 | 1,924,560 | 2023 | 1,400 | Apr 2024 | 2018 | 837 | Dec 2019 |

Table 2: Large Language Models (LLMs) used in this work

| Name | Provider | Version | Cutoff Date |
|---|---|---|---|
| GPT-4o | OpenAI | `gpt-4o-2024-08-06` | October 2023 |
| Claude 3.5 Sonnet | Anthropic | `claude-3-5-sonnet-20241022` | April 2024 |
| Llama 3.3 70B Instruct-Turbo | Meta via TogetherAI | `Llama-3.3-70B-Instruct-Turbo` | December 2023 |

**CSV (direct generation).** The `CSV` method prompts LLMs to generate CSV records that strictly adhere to the schema, including exact header rows, data types, allowed values, and realistic inter-field relationships (see Figure 6). The prompt specifies rules to ensure statistical plausibility and the inclusion of realistic edge cases, while the generation is executed in batches with schema-based validation of each record. This process relies *solely* on the LLMs pretrained knowledge without any direct access to the private data. When the schema specifies numeric ranges (e.g., float in $[a, b]$), the prompt and validator accept continuous values; in practice Claude/GPT emit decimals within range that we can naturally retain when downstream mechanisms support continuous features.

**Agent (state machine) approach.** The `Agent` method (implemented as a state machine, see Figure 7) is a multi-step process to construct a structural causal model (SCM) from the schema meta-data. It begins by describing the full set of variables and domain-specific constraints, then sequentially constructs a causal DAG – first identifying root nodes and then establishing edges (ensuring acyclicity deterministically) before mapping variables to structural equations. For numeric nodes, the Agent selects a parametric family (e.g., Gaussian, Log-Normal, Gamma) and emits full Pyro sampling code, yielding continuous values by construction. We validate acyclicity and node coverage with NetworkX before code generation. Then, the final output is an integrated Pyro Python program that enforces variable ranges and constraints. The `Agent` method has two variants: we experiment with generating multiple expert models whose records are re-sampled (using uniform or facility location-based sampling [111]).

## 4 Evaluation Framework

Our evaluation framework assesses the viability of the surrogate public data in three DP auxiliary tasks: **(1)** classifier pretraining, **(2)** hyperparameter optimization, and **(3)** privacy-utility trade-off estimation. Each task is assessed using three datasets, and corresponding DP mechanisms. Our strategy in evaluating each task is guided by a high level question: *how useful is each **surrogate** public data method relative to **traditional** public data and relative to the **lack** of any public data?*

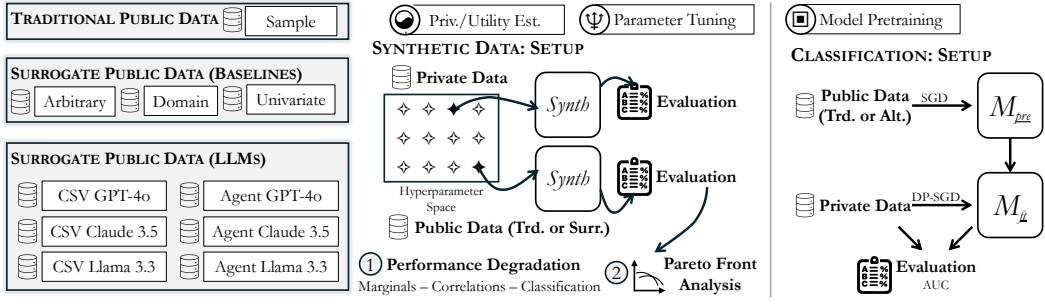

Figure 2: An overview of our evaluation framework (Section 4) We assess the usefulness of regular public data and surrogate public data (Section 3) on three tasks. Two tasks are related to synthetic data generation – hyperparameter tuning and privacy-utility estimation – and one to classification model pretraining.

These three auxiliary tasks (public pretraining, hyper-parameter tuning, and privacy-utility estimation) are repeatedly emphasized as high-impact components in deployed DP pipelines [89]. We therefore concentrate our benchmark on them and report $(\epsilon, \delta)$ and other settings inline (full grids in Appendix D.4). We report standard error over 10 seeds where appropriate.

**Task 1: model pretraining for classification.** We assess the benefit of surrogate public data as a pretraining step for binary classification on tabular data using an FTTransformer model [46, 95]. Public and private datasets are split into train, validation, and test sets (72:8:20), and performance is measured via AUC along with an AUC Advantage metric comparing models with and without public pretraining, aggregated over multiple experiment seeds. See Appendix D.1.1 for the detailed specification of this task.

**Task 2: hyperparameter tuning for synthetic data generation.** We evaluate whether surrogate public datasets can effectively guide hyperparameter selection for DP synthetic data generators across varying privacy budgets $\varepsilon$. For each synthesizer and hyperparameter configuration, we generate synthetic datasets matching the size of the private data and measure their performance on the private data using multiple metrics (marginals, correlations, and classification-based; see Table 3). We quantify performance degradation by comparing results obtained when tuning hyperparameters on surrogate public data versus directly on private data (optimal choice of hyperparameters), aggregating outcomes across multiple experimental seeds via Pareto frontier analysis. See Appendix D.1.2 for the detailed specification of this task.

**Task 3: privacy-utility estimation for synthetic data generation.** Finally, we assess how accurately surrogate public datasets can approximate the privacy-utility trade-off curve of a DP synthetic data mechanism fitted to private data. For each combination of synthesizer, dataset, and metric group, we select the best-performing hyperparameters based on surrogate public datasets, and then evaluate the DP mechanism across a range of $\varepsilon$ values. This produces paired performance curves: one based on the private data (true curve) and others based on surrogate public datasets. We quantify the dissimilarity between these curves using the $\ell_1$ and $\ell_2$ distances. See Appendix D.1.3 for the detailed specification of this task.

**Datasets.** We run the experiments on three datasets (ACS, EDAD, and WE; high-level details presented in Table 1). Each dataset has a private, sensitive split; additionally, we pair each dataset with a reasonable public analogue. These public datasets have inherent distribution shift between them; for ACS this is a geographical variation (assuming the Massachusetts sample is publicly available, while a more diverse national sample is private) or, for EDAD and WE, temporal differences (versions of the same survey from prior years). All datasets contain only categorical features to ensure compatibility with synthetic data generation methods. The private split serves as ground truth to benchmark the contribution of the "traditional" approach of using a public split compared to our surrogate generation methods. To mitigate the risk of data memorization in LLMs, we specifically selected the private splits for EDAD and WE to be *recently* published, i.e., after the training data cutoff of some of the LLMs we evaluate. To this end, we include a memorization analysis in Appendix D.2.4, based on the methodology of Bordt et al. [18]. Concretely, EDAD and WE private splits were publicly released on April 30, 2024, after GPT-4o (Oct 2023) and Llama-3.3 (Dec 2023) cutoffs, and effectively aligned with Claude 3.5 Sonnets "up until April 2024" window.[4] For the complete details for each dataset and an in-depth discussion of LLM memorization, refer to Appendix D.2.

**Mechanisms.** Our private mechanisms encompass differentially private classification (Task 1) and data synthesis (Tasks 2 & 3). As discussed previously, for classification, we employ an FTTransformer model [46] – a transformer-based architecture tailored for tabular data that rivals gradient boosting methods like XGBoost – by adapting it with minor modifications to support DP-SGD for private fine-tuning and allowing pretraining with public data via standard gradient updates [1, 95]. For private data synthesis, we evaluate three state-of-the-art methods – `PrivBayes`, `GEM`, and `AIM` – that follow the "Select-Measure-Project" paradigm: they privately select statistical queries (e.g., $k$-way marginals or correlations) on sensitive data, add noise to these measurements, and then project the results onto a synthetic distribution [120, 73, 80]. See Appendix D.3 for complete model details, with detailed hyperparameter settings provided in Appendix D.4.

---

[4]At worst there is a one-day overlap; in addition, our verbatim-memorization tests [18] show no evidence of row-level leakage; see Appendix D.2.4.

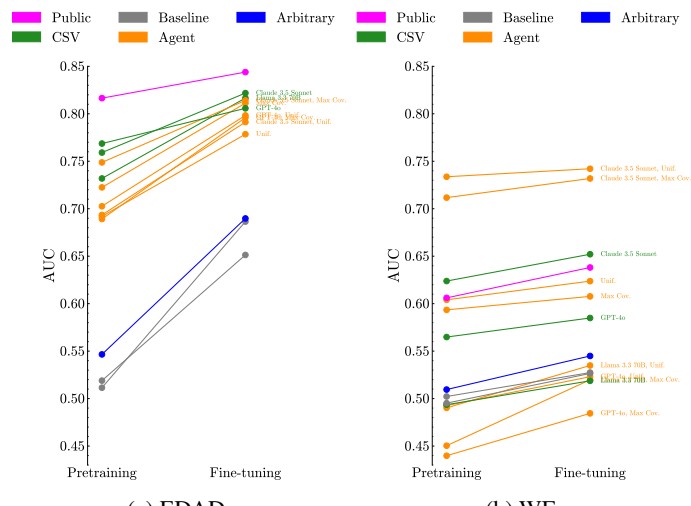

| Method | Class. | Corr. | Marg. |
|---|---|---|---|
| Public | 0.008 | 0.047 | 0.097 |
| CSV (Claude) | 0.033 | 0.046 | 0.227 |
| Agent (Claude, Unif.) | 0.004 | 0.134 | 0.225 |

**Table 1:** Pareto Efficient Methods for PrivBayes on EDAD.

| Method | Class. | Corr. | Marg. |
|---|---|---|---|
| CSV (Claude) | 0.013 | 0.002 | 0.045 |
| Agent (Claude, Max Cov.) | 0.010 | 0.003 | 0.125 |
| Agent (Claude, Unif.) | 0.004 | 0.003 | 0.024 |

**Table 2:** Pareto Efficient Methods for AIM on ACS.

| Method | Class. | Corr. | Marg. |
|---|---|---|---|
| Arbitrary (Baseline) | 0.043 | 0.052 | 0.056 |
| Agent (All, Max Cov.) | 0.016 | 0.096 | 0.172 |
| Agent (Claude, Unif.) | 0.019 | 0.040 | 0.070 |

**Table 3:** Pareto Efficient Methods for PrivBayes on WE.

Figure 3: Task 1 – Pretraining: **(a, b)** Comparing the mean AUC on the *test* subset *private* split for the **Pretraining** model vs. the **Fine-tuned** model, grouped by generation method (mean calculated across DP finetuning parameter space when the best configuration is chosen with the *validation* subset of the *public* split for the pretraining step, across 10 runs). Note how the **starting** point of model AUC differs, while the **improvement** from private finetuning (i.e. the increase in AUC) is relatively **stable**. Task 2 – Hyperparameter tuning: **(Tables 1, 2 and 3)** show some Pareto frontiers for the *performance degradation* metric when hyperparameter tuning using (surrogate) public data methods for tuning relative to tuning on private data. Note how CSV and Agent methods are competitive with tuning on the regular public data. See Section 5 and Appendix E for complete results. Note that we adopt the Olympic medal convention in each table in our paper: gold , silver and bronze cells signify first, second and third best performance, respectively.

## 5 Results

**Task 1: Model Pretraining for Classification.** Our experiments provide strong evidence that LLM-based methods – both CSV and Agent generation (notably with Claude 3.5 Sonnet) – offer a competitive alternative to traditional public data in the small dataset regime (fewer than 10K records). **In many settings, the surrogate public data matched (or even occasionally exceeded) the performance of regular public data as a starting point for DP fine-tuning.** Figure 3 presents our experimental results on the EDAD and WE datasets, demonstrating how pretraining on the surrogate public data can vastly improve the starting point of model performance. For the ACS dataset, we do not observe a benefit from pretraining, either with regular or surrogate public data (see Figure 14). However, when the ACS dataset is sub-sampled to a smaller dataset (e.g., 5% of the records), we observe a similar pattern regarding the usefulness the traditional and surrogate public data as with the EDAD and WE datasets. See Section 5.1 for a further analysis of the role of dataset size. See Appendix E.1 for an in-depth treatment of all pretraining results.

**Task 2: Hyperparameter Tuning for Synthetic Data Generation.** No single surrogate public-data strategy dominates across all evaluation criteria (Figure 3; Tables 1-3). Each generation method achieves superior performance on a subset of metrics – whether predictive accuracy, preservation of pairwise correlations, or marginal distribution fidelity. Pareto-frontier analysis suggests that the critical determinant is the extent to which a surrogate captures higher-order dependence structure; even the Arbitrary baseline appears on the frontier *even though it does not encode similar statistical relationship of the private data* (for evidence, see Figure 3; Table 3). Nonetheless, the LLM-generation methods (CSV and Agent) exhibit the most favorable overall trade-offs, underscoring their utility for hyperparameter tuning of DP synthetic data generators. See Appendix E.2 for an in-depth treatment of all hyperparameter tuning results.

*Remark: Why might* Arbitrary *be performant here? A plausible explanation is that PrivBayes benefits from BN-structured surrogates when prioritizing $k$-way interactions; even randomly param-*

*eterized BNs can provide a coarse proxy for how distributional structure matters during public-space hyper-parameter selection. In contrast, `Arbitrary`'s misspecified marginals undercut classifier pre-training (Task 1) and offer no consistent gains for privacy-utility curve matching (Task 3).*

**Task 3: Privacy-utility Estimation for Synthetic Data Generation.** The story for this task is less clear-cut. While surrogate public data generally provides a reasonable approximation for the privacy-utility tradeoff curves, the differences between various generation methods were not pronounced. We observed that regular public data provided the best or second-best estimation of the privacy-utility tradeoff curve in the vast majority of cases. This observation suggests that data similarity may be an important contributing factor. However, a subsequent analysis (Section 5.2) examining the role of similarity did not reveal a clear pattern that explains this result. See Appendix E.4 for an in-depth treatment of all privacy-utility estimation results.

## 5.1 ACS and the Role of Dataset Size

For the ACS dataset, we do not observe any benefit from pretraining, either with traditional or surrogate public data (e.g., as Figure 14b shows for $\varepsilon = 1$). However, **a follow-up analysis reveals that this is due to the relatively large size of the dataset.** Dataset size is a key factor in differentially private mechanisms, as it directly influences the noise level added to achieve a specific level of privacy protection [38]. The relatively large size of the ACS dataset partly explains why the benefit of regular public data pretraining appears marginal in, e.g., Table 8a; as privacy sensitivity scales inversely with dataset size, when the private dataset is sufficiently large, the magnitude of noise necessary for a DP guarantee decreases.

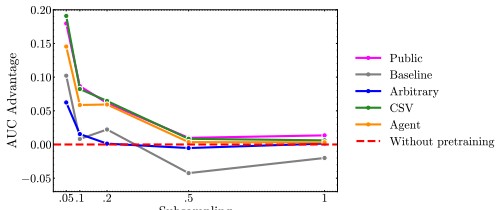

Figure 4: Mean AUC Advantage of the DP classification model with $\varepsilon = 1$ after pretraining for each subsampled dataset, grouped by generation method category. The mean is calculated across the DP finetuning hyperparameter space when best pretraining hyperparameter configuration is chosen for the pretraining step, with 10 runs per hyperparameter configuration.

To investigate this effect, we repeated the full pretraining experiment on four ACS subsets obtained by subsampling at 5%, 10%, 20%, and 50%. In these experiments we focus on the AUC advantage at $\varepsilon = 1$, where the benefit of public data is most pronounced. Figure 14b (in Appendix E.1) shows that with 5% subsampling, the ACS dataset exhibits a similar pattern of performance to the one we found with the EDAD and WE datasets.

In fact, the LLM-based methods (using Claude Sonnet 3.5) outperformed the traditional public dataset. Figure 4 presents the relationship between the (subsampled) dataset size and the AUC advantage per generation method category to $\varepsilon = 1$. As we examine smaller datasets, the differences we observe align with results on the EDAD and WE datasets. Both the `CSV` and `Agent` surrogate datasets perform on average similarly to traditional public data. This observation may also help explain the negative findings reported by Swanberg et al. [100] regarding the use of LLM-generated public data for DP synthetic data generation on the Adult dataset, which is substantially larger than some of the datasets considered here. We hypothesize that with smaller datasets, LLM-generated public data surrogates could provide some benefit in pretraining differentially private data synthesizers, but leave a closer examination of that DP auxiliary task to future work.

## 5.2 Dataset Similarity May Be Less Important Than You'd Think

By using public data in DP auxiliary tasks, we implicitly assume statistical similarity to the private, sensitive data. Our results generally back this up; in pretraining (Task 1) and privacy-utility trade-off estimation (Task 3), we observe consistently better traditional public data performance. To explore whether the traditional public data dominance (and the relative performance rankings of the surrogate public data) could be explained by dataset similarity, we measure the similarity between all datasets using two common metrics from the DP literature (see Appendix G.1): Total Variation Distance (TVD) and average error across all 3-way marginal queries (3WM) [73, 80].

**Comparing Private vs. Public.** The first dataset similarity question we ask is: **how similar is a public data variant to the true, private data?** Both *traditional* and *surrogate* private vs. public data similarity results are shown in Table 25. In general, our metrics suggest that the traditional public dataset and the `Univariate` baseline (recall, this baseline samples independently with a

little noise from the true private distribution) are most similar to the private data. For EDAD and WE datasets, we can explain the lower overall similarity scores due to their higher dimensionality (defined as the Cartesian product of possible unique variable values; see the "× Dims" column in Table 1); the dataset distance is exacerbated by sparsity (particularly for TVD). However, even accounting for the limitations of these metrics, *we did not observe a clear relationship between the similarity rankings of public datasets and their usefulness rankings in the pretraining and privacy-utility tasks*. Our explanatory hypothesis: common similarity metrics, like TVD and 3WM, may not adequately capture dataset characteristics relevant to the DP auxiliary tasks we frame. We leave further exploration of suitable metrics to future research.

**Comparing Among (Traditional or Surrogate) Public Data.** The second dataset similarity question we ask is: **how similar are public data variants to each other, and does this partially explain their relative performance rankings?** To this end, we compared similarity metrics among traditional and surrogate public data, with heatmap plots provided in Appendix G. The most consistent pattern observed across datasets and metrics is the strong similarity between `Agent` pairs using the same LLM but differing only in mixing methods (`Unif.` vs. `Max Cov.`) (this is barring TVD for EDAD and WE, where most entries are zero due to the aforementioned dimensionality constraints). This pattern extends to similarities between the overall mixing datasets and individual `Agent` datasets (with the exception of the Llama datasets on EDAD). This is expected since these pairs share the same underlying source of sampled records. Interestingly, we did not find stable similarities across generated data between different LLMs within either `Agent` or `CSV` methods, or between the same LLM across these two methods. Again, this could be an artifact of the metrics we use, but we leave a deeper exploration of this for future work.

# 6   Related Work

**Public data in differential Privacy.**   Historically, the notion of using surrogate (independently generated) public data for downstream private learning appears in earlier work on e.g. on importance-weighted DP learning [60], which we view as philosophically aligned with our setup. Extensive prior work has then further shown how public data can improve DP machine learning via pretraining followed by private fine-tuning, particularly in NLP and vision tasks [106, 7, 118, 44, 43, 53, 20, 104, 63]. Public data is integrated directly into differentially private computations for tasks such as private estimation [15], statistical queries [12, 71], and predictive learning [11, 111, 13, 61, 123, 85, 14, 48, 86, 110, 17, 76]. Differentially private fine-tuning of pretrained LLMs with DP-SGD [1] has been employed to generate synthetic data for downstream tasks [67, 119, 8, 115] and is comprehensively reviewed in Cummings et al. [31].

**Related LLM approaches.**   Contemporary work by Swanberg et al. [100] is closely related to ours; they evaluate LLM-generated public data in a single setting for public pretraining of private synthetic data mechanisms. However, their work focuses on a narrow experimental setup on pretraining private synthetic data models; in contrast, our study assesses surrogate data for several DP auxiliary tasks, spans multiple datasets and metrics, and includes memorization tests [18].

**Generating tabular data.**   Transformer-based neural models have been applied to generate synthetic tabular data either by training models from scratch [97, 49, 122] or by pretraining a tabular foundation model [77]. Pretrained LLMs have also been adapted via fine-tuning [19] or in-context learning [96, 66], but these methods are unsuitable since they condition directly on sensitive data.

# 7   Limitations

The overarching goal of this work is to address a significant barrier to the real-world adoption of differential privacy [38]. Enabling more widespread deployment of differential privacy, when appropriate, can promote the protection of individuals privacy [31]. However, our work does have several risks and limitations. First, there is a risk that LLM memorization may lead to overly optimistic performance estimates (we attempted to mitigate this risk by carefully checking for evidence of memorization, see Appendix D.2.4).

Second, the normative implications of employing LLMs to generate surrogate public data should be carefully analyzed in this context. Recent work by Tramèr et al. [107] cautions against treating web-scraped LLM training data as "public" or non-sensitive. Traditionally, differentially private algorithms have assumed data is either fully private (restricted) or fully public (freely available and safe to reuse). However, Tramèr et al. [107] emphasize a messier reality; social media and other

sources of personally identifiable information, for example, may be both accessible to language models for training data *and* contain sensitive information specific to individuals. When an LLM is trained on such data, it may memorize fragments of it; regurgitating these private fragments could be interpreted as a privacy violation. Indeed, even if a final model is fine-tuned under DP constraints, privacy violations may originate from the pretrained model (e.g., a base model memorized private details during pretraining, and a subsequent DP fine-tuning step does not noise those probabilities sufficiently to obfuscate). This undermines trust, as an individual may be told that the entire pipeline is "privacy preserving," yet see their personal data re-emerge in the final models outputs.

We carefully position our work under the paradigm shift identified by Tramèr et al. [107]. Using LLMs to emulate *expert-driven* data-generating processes risks inadvertently exposing sensitive information that is publicly available, as mediated by the LLM. Thus, we propose that best practice is to *report empirical measurements of memorization levels*. We do this by leveraging work by Bordt et al. [18] on verbatim memorization of tabular data by LLMs; see Appendix D.2.4. Additionally, we report on datasets (see Table 1) that post-date LLM training (for the models we evaluate; see Table 2). Choosing tasks where the LLMs prior knowledge is outdated or non-existent demonstrates performance on truly unseen data [29]. We stress the importance of communicating these nuances, and of reporting, to the best of one's knowledge/ability, the empirical level of memorization and the potential LLM data regurgitation risks when presenting these methods.

Additionally, given substantial evidence that LLMs encode biases [41], these biases could be reflected in the generated data – either implicitly in `CSV` generation or explicitly via the causal relationships in the `Agent`-based approach. For instance, a stereotypical correlation could persist through pretraining and DP fine-tuning, ultimately resulting in an unfair classifier. We leave a detailed investigation of these issues for future work.

Finally, we note that our methods require minimal metadata per column: (1) name, (2) one-line description (if available), and (3) categorical domain or plausible/known numeric range. Such schemas are extremely common in social-science surveys (ICPSR, UKDS) and medical data models (FHIR, OMOP). We defer an investigation of, e.g., noisy logs or sensor data that lack a coherent schema to future work. The CSV and Agent natively support continuous values (validator-checked CSV; parametric numeric nodes in Agent). We bin continuous columns in our experiments to align with current synthesizers; principled discretization design, potentially with a small DP budget, remains a possible extension. Because surrogates are mechanism-agnostic, they could easily be extended to support federated or local DP pipelines; we leave empirical validation to future work.

## 8 Conclusion

In this work, we asked whether LLMs can be used to generate effective surrogate public data for solving DP auxiliary tasks in settings where traditional public tabular data is limited or unavailable. Each approach we considered leveraged schema-level metadata to generate surrogate public data in the same domain as the private, sensitive data. We considered LLM data generation methods like directly prompting for tabular `CSV` records, and through an `Agent` that constructs an SCM over the schema variables using the LLM as an expert prior. Our evaluations demonstrated that LLM-generated public data surrogates can be used to significantly improve the DP auxiliary task of private classifier pretraining with public data. The LLM-generated public data surrogates were also useful for the tasks of hyperparameter tuning and privacy/utility tradeoff estimation, albeit with less impressive performance relative to a strong baseline (`Arbitrary`). Overall, our results provide an affirmative answer: for the DP auxiliary tasks we considered, generating surrogate public data with LLMs *can* overcome tabular public data scarcity.

## Acknowledgement

The authors thank Lucius EJ Bynum for helpful discussions during the design of the generation methods, and Ran Canetti, Adam Smith, and Marco Gaboardi for valuable comments regarding the analysis of the result. This research was supported by computational resources provided through the National AI Research Resource (NAIRR) pilot program (Award 240327). S.H. is supported in part by DARPA under Agreement No. HR00112020021. L.R. is supported by the NSF GRFP Grant No. DGE-2234660. J.S. is supported in part by NSF Awards No. 2312930 and 2326193. Any opinions, findings, and conclusions or recommendations expressed in this material are those of the authors and do not necessarily reflect the views of the United States Government.

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

# Contents

## A  Future Work

The strong performance of the `Arbitrary` method in hyperparameter tuning is intriguing, as it suggests that finding good-enough hyperparameter configuration might depend on the record domain and the synthetic data generators, and not necessarily on the private data. This raises questions about potential theoretical justifications for this observation.

The fact that traditional public data often performs best for privacy/utility trade-off estimation would lead us to believe that dataset similarity plays an important role for this task. We hypothesize that the two similarity metrics used in this work, while being natural candidates, may not adequately capture dataset characteristics relevant to estimating the behavior of data synthesizers across privacy budget settings. Identifying metrics that better predict which surrogate data provides accurate trade-off estimations would be beneficial. Such a metric could enable, e.g., the exponential mechanism [82] to select similar datasets (or combinations of datasets), if such a metric had low sensitivity with respect to the private dataset.

Several additional DP auxiliary tasks remain unexplored in our study, such as using public data for seeding synthetic data generation [100] and assessing the success rate of privacy attacks as a function of $\varepsilon$ [32]. We leave these avenues for future research.

We propose three approaches to improve the quality of surrogate data produced by `Agent`-based methods, making it more closely resemble private data. First, subject matter experts can review and refine the generated SCM to better encode experts' domain knowledge. Second, Retrieval-Augmented Generation (RAG) [70] could be beneficial to surface specific knowledge from scientific literature, enabling the model to incorporate both accurate causal relationships and their quantitative

parameters as established in peer-reviewed research. Third, recent advancements in reasoning LLMs [99, 58, 34] may enhance LLMs' ability to consider causal relationships.

Finally, some recent work on Sequence Driven Structural Causal Models (SD-SCMs) shows how to simulate counterfactual outcomes and treatment scenarios that are often inaccessible in sensitive datasets (by allowing an LLM to specify structural equations implicitly, given a topological order and a specific prompting structure) [22]. Similarly to the surrogate public data approaches explored in this paper, the SD-SCM approach does *not* require access to a downstream private dataset of interest; instead, it only requires a schema over the data to be generated, and a user to specify the prompting structure and topological order over variables (which could be generated e.g., by the first few steps of the `Agent` procedure given in Figure 7). There may be many potential uses for SD-SCM generated surrogate public data for private causal algorithms; for example, we believe that future work could explore how it can be used to improve the performance of hyperparameter tuning for private causal effect estimators.

## B   Additional Related Work

**Public data in differential privacy**   Empirical evidence demonstrates that public data can improve the performance of differentially private machine learning models through a two-stage approach: pretraining on public data followed by differentially private fine-tuning on sensitive data. This approach has been extensively studied across NLP and vision tasks [106, 7, 118, 44, 43, 53, 20].

Ganesh et al. [42] identify two phases in neural network optimization within non-convex loss landscapes. The first locates an optimal basin, where public data suffices and using the privacy budget is unnecessary. The second performs local optimization within that basin; here, if the public and target distributions differ – as they often do – consuming privacy budget to update weights with sensitive data is beneficial. Supporting this, Thaker et al. [104] show that public pretraining outperforms fully private training in vision tasks, even under significant distribution shifts. This advantage holds even when private fine-tuning is limited to the final layer, as in Ke et al. [63].

Another research direction incorporates public data directly into differentially private computations, rather than treating it as a separate preprocessing step. This approach spans private estimation [15], statistical queries [12, 71], and learning and optimization [11, 111, 13, 61, 123, 85, 14, 48, 86, 110, 17, 76]. An emerging line of research finetunes pretrained, open-source LLMs on private, sensitive data with DP-SGD [1] to generate training data for downstream models, such as classifiers or other LLMs [67, 119, 8, 115]. For a broader survey on recent advances in privacy research, see [31].

Finally, contemporary work by Swanberg et al. [100] is closely related to ours, but with three key differences. First, while they evaluate LLM-generated public data in a single experimental setting (for public pretraining of private synthetic data mechanisms), we assess its utility across several DP auxiliary tasks — including hyperparameter tuning *for* synthetic data generation, privacy/utility tradeoff estimation, and private *classifier* pretraining. Second, our evaluation is broader in scope, incorporating multiple datasets (with different data-origins), diverse metrics and additional baselines / methods for leveraging an LLM to produce surrogate public data. Third, we designed our experiments to mitigate the risk of positive results due to memorization, including an explicit test based on Bordt et al. [18], and provide results and analysis to assess the impact of data leakage on the performance of our methods.

**Generating tabular data with LLMs**   Transformer-based models can be used to generate synthetic samples from tabular data. The fundamental approach involves treating each record as a "sentence" for the transformer architecture to process. Two overall strategies exist: training transformers from scratch specifically for tabular data and adapting pretrained LLMs for tabular generation tasks. For the first strategy, one variant trains a transformer on an individual dataset or distribution to produce synthetic records [97, 122, 49, 122]; another variant pretrains a general tabular foundation model on multiple datasets and then adapts this model to novel unseen datasets through in-context learning [77]. The second strategy uses existing pretrained LLMs, adapting them for tabular data generation through either fine-tuning [19] or in-context learning [96, 66]. These methods *cannot* be directly applied to our setting, as we only consider DP auxiliary tasks (e.g., pretraining, hyperparameter tuning) that *do not* consume privacy budget. Both approaches condition on sensitive data and thus require accounting for privacy loss.

Recent work has explored the potential of LLMs for causal modeling tasks, including pairwise causal discovery, causal model generation, and counterfactual reasoning [65, 27]. While causality itself is not the primary focus of our project, the ability to produce plausible causal models is highly relevant since causal models are also generative, capable of producing realistic records. LLM-based causal model discovery methods can operate either with metadata alone (using only dataset descriptions and schema) [108, 75, 74, 121, 33, 22] or with additional observations [2, 68] – with the metadata-only approach being particularly relevant to our project as we have no access to observations. Most literature in this area that operates without observations focuses solely on discovering causal *graphs* – descriptions of causal dependencies without specifying conditional distributions. However, [22] extends this approach by adding a second step that prompts LLMs with the topological order over variables, embedded in a prompt structure, to generate records directly.

# C  Details of Surrogate Public Data Generation

## C.1  Baselines

Before discussing the LLM-based approach, we present a series of baseline generation processes to systematically evaluate which aspects of public data characteristics are useful for differential privacy tasks: pretraining, hyperparameter tuning, and estimating the privacy-utility trade-off. The baselines differ in statistical structure and in the information available about the private data.

### C.1.1  Uniform Distribution over the Domain

The dimensionality of the data plays a critical role in differentially private algorithms [79, 93], as it could affect, for example, the magnitude of noise introduced to satisfy DP *or* the ratio between signal and that noise (e.g., when tuning data synthesizers like PrivBayes or AIM). This `Uniform` distribution baseline captures the scenario where we have no prior knowledge about the underlying data distribution beyond the schema itself by using the maximum entropy probability distribution [59]: for each record, `Uniform` samples i.i.d. from either the set of possible values (for categorical columns) or the specified range (for continuous columns), both given in the schema.

### C.1.2  Univariate Distribution

Beyond knowledge of the record domains, organizations and researchers might have access to prior information about the *univariate* distributions of individual columns, either precisely or approximately. This prior knowledge is available in cases where organizations may have released various statistical measures of private data, such as histograms, means, medians, and standard deviations, with or without differential privacy [94, 52]. As a facsimile for data generated with knowledge of the distributions along individual columns, the `Univariate` baseline samples independently from each column according to the empirical univariate distribution *drawn directly from the private data*. To make this baseline more realistic – assuming only an approximate PDF (e.g., the distributions "shape") is known – we round the probabilities to two decimal places, normalize to 1, and rescale during sampling.

### C.1.3  Arbitrary Distribution

The previous two baselines are limited by *column independence* in their sampling, preventing them from capturing complex statistical structures needed for higher-order analysis and predictive tasks [93]. To isolate the role of structural dependencies in our *DP auxiliary tasks* with surrogate public data, we consider whether *only* capturing the existence of relationships between columns could make surrogate public data a useful prior. To test this, we generate an *arbitrary* dataset from a random but structured distribution that adheres to the schema.

Algorithm 1 details the full `Arbitrary` baseline procedure; here, we provide a high-level overview of the two-step generation process. First, we construct a random Directed Acyclic Graph (DAG) representing a Bayesian network over the column variables. The DAG is built sequentially, with each new node potentially connecting to any previously added nodes, subject to a maximum in-degree (here we used 5). This ensures a structured yet arbitrary dependency pattern between variables. Second, we parameterize the network by sampling conditional probability tables for each node. For a given node, we use a Dirichlet distribution with concentration parameter $\alpha = 1$ to generate probability distributions for each configuration of its parent variables. Specifically, for each parent value combination, we sample a categorical distribution from the $k$-simplex, where $k$ is the cardinality of the node's domain. This yields a distribution with meaningful dependencies (e.g., correlations) while remaining *entirely independent* of the true empirical distribution of the private data.

## C.2  CSV Direct Generation

We evaluate a direct approach to data generation using LLMs. The generation process involves prompting the LLM to create `CSV`s – tabular records that adhere to the schema while following specific guidelines [6]. These guidelines instruct the model to ensure realistic value distributions and relationships between fields, maintain real-world patterns and constraints, and incorporate edge cases at frequencies that mirror their natural occurrence. Similarly to the other surrogate public data methods we evaluate, this approach operates *without* access to the private dataset, relying *solely* on the LLM's pretrained knowledge.

To ensure data quality, each generated record is validated against the schema, and only valid records are retained. Due to context window limitations and API constraints, the generation process is executed in multiple batches until the desired number of records is obtained [87, 9, 105]. Note that,

due to the autoregressive nature of LLMs (see e.g., Section 2), records within the same generation batch are *not* sampled independently, in contrast to the baseline methods.

## C.3   Agent (State Machine) Approach

As a final approach, we employ a multi-step, `Agent`-based process to elicit a structural causal model (SCM) from an LLM given only text-based access through prompts and responses. Our goal is to arrive at a coherent directed acyclic graph (DAG) that captures the inter-dependencies among variables in the schema, along with associated structural equations (e.g., the actual distributional parameters, probabilities, etc.). Each step concludes with an automated validation of the LLM's output; so, if any contradictions or omissions are detected, the `Agent` (implemented as a state machine, see Figure 7) automatically refines our prompt and re-queries the LLM.

First, we prompt the LLM to  list out all variables (keys) from the provided schema, ensuring the response exactly matches the schemas variable set. Next, we ask it to  propose realistic consistency constraints among these variables; these constraints should capture domain knowledge such as permissible value ranges (e.g., "age must be at least 0") or logical relationships (e.g., "an individual who is 10 years old must have fewer than 10 years of education"). We then instruct the LLM to identify a subset of variables that can serve as the "root nodes" in a causal graph, typically those deemed exogenous or less likely to be influenced by other variables in the schema. From there, the LLM proposes parentchild relationships  from root nodes to non-root nodes, and then  among all remaining variables,  ensuring no cycles are introduced so that the final structure is a DAG (which we validate with a graph library to confirm it contains all variables exactly once and remains acyclic [50]).

Having obtained a DAG, we prompt the LLM to  map each variable to a structural equation that references its parents. For instance, if a node depends on two parents, the LLM might generate a formula specifying a probabilistic distribution conditional on parent values. These structural equations encode marginal distributions for root variables and conditional distributions for their descendants. Sometimes the structural equations are not fully specified (e.g., the probability parameter in Bernoulli distribution is parameterized), so we instruct the LLM to  assign values to all parameters. Then,  we combine the DAG and structural equations automatically into a single code snippet (we use the Pyro library [16]), which lets us generate synthetic data automatically. Finally, we ask the LLM to amend the Python code to  enforce the range or valid values for each column, and  include the constraints elicited at the beginning of the interaction. This entire interaction is a stateful, automatic, closed loop, allowing the LLM to act on its own as an "expert" to design a plausible causal model *solely* from schema level information (containing short descriptions of each variable), *without* a need to inspect any real-world sensitive records.

To extend this approach from a "single expert" to a "panel of experts," we execute the complete generation workflow multiple times to produce a *collection* of datasets, inspired by prior work on "self-consistency" prompting methods [112]. These datasets are then combined to yield a single mixed dataset using two approaches. The first approach, `Unif.`, involves uniform sampling of records across all generated datasets. The second approach, `Max Cov.`, solves the *Facility Location* submodular problem [111] by finding a subset of datasets that maximizes the sum of pairwise Total Variation similarities. This optimization selects a subset of datasets that aims to represent the space of all generated datasets [111]. Then, similarly to the `Unif.` approach, we sample records uniformly from the *selected datasets*.

One important advantage of agent-generated SCMs is that domain experts can modify the causal structure, structural equations, and constraints based on their expertise, scientific literature, and common sense. We leave this for future work.

```
{
    . . .
    "RELACT": {
        "description": "Main labour market activity status",
        "dtype": "int64",
        "values": {
            "1": "Employed",
            "2": "Unemployed",
            "3": "Retired",
            "4": "Student",
            "5": "Unable to work",
            "6": "Doing unpaid social work or charitable activities
                ↪ ",
            "7": "Other inactive person"
        }
    },
    "CERTIG": {
        "description": "Degree of disability",
        "dtype": "int64",
        "values": {
            "1": "0−32%",
            "2": "33−44%",
            "3": "45−64%",
            "4": "65−74%",
            "5": "75% or more",
            "6": "Not known"
        }
    },
    "AUDI_7_1": {
        "description": "Has significant difficulty hearing a
            ↪ conversation with several people without a hearing
            ↪ aid",
        "dtype": "int64",
        "values": {
            "1": "Yes",
            "2": "No"
        }
    },
    . . .
}
```

Figure 5: Excerpt from the schema of the EDAD dataset (Spanish disability, autonomy, and dependency survey) [56].

---

**Algorithm 1** Random Bayesian network generation for the arbitrary dataset.

---

1: **procedure** GENERATERANDOMBN($\mathcal{S}, d_{\max}, \alpha$)

    *Input:*

    $\mathcal{S} = \{(v_1, \mathcal{D}_1), \ldots, (v_n, \mathcal{D}_n)\}$: Schema where $v_i$ is a variable and $\mathcal{D}_i$ is its domain of possible values

    $d_{\max}$: Maximum parent degree

    $\alpha$: Dirichlet concentration parameter

    *Output:*

    Bayesian network $\mathcal{B} = (\mathcal{G}, \Theta)$ where:

      $\mathcal{G} = (\mathcal{V}, \mathcal{E})$: Directed acyclic graph with nodes $\mathcal{V}$ and edges $\mathcal{E}$

      $\Theta = \{\theta_{v|\Pi_v} : v \in \mathcal{V}\}$: Set of conditional probability distributions, where $\theta_{v|\Pi_v}$ represents the distribution of $v$ given its parent set $\Pi_v$

    *Initialization:*

2:     Extract variables $\mathcal{V} = \{v_1, \ldots, v_n\}$ from schema $\mathcal{S}$

3:     Define indexing function $\phi_v : \mathcal{D}_v \to \{1, \ldots, |\mathcal{D}_v|\}$ for each $v \in \mathcal{V}$

    *Network Structure Generation:*

4:     Randomly permute the ordering of variables in $\mathcal{V}$

5:     Initialize edge set $\mathcal{E} \leftarrow \emptyset$

6:     Initialize parameter set $\Theta \leftarrow \emptyset$

7:     **for** $i = 1$ to $n$ **do**

8:         Define candidate parent set $\mathcal{C}_i = \{v_1, \ldots, v_{i-1}\}$

9:         Select $\Pi_i \subseteq \mathcal{C}_i$ randomly with $|\Pi_i| \leq \min(d_{\max}, i-1)$

10:        Add edges $\{(u, v_i) : u \in \Pi_i\}$ to $\mathcal{E}$

       *Parameter Generation:*

11:        Let $\Omega_{\Pi_i}$ be the set of all configurations of $\Pi_i$ where each configuration $\pi \in \Omega_{\Pi_i}$ is a tuple of values

12:        Let $k_i = |\mathcal{D}_{v_i}|$ be the cardinality of variable $v_i$'s domain

13:        **if** $\Pi_i = \emptyset$ **then**

14:           $\theta_{v_i} \sim \mathrm{Dir}(\alpha \cdot \mathbf{1}_{k_i})$              ▷ Sample from Dirichlet with symmetric $\alpha$ parameter

15:        **else**

16:           **for all** $\pi \in \Omega_{\Pi_i}$ **do**

17:              $\theta_{v_i|\pi} \sim \mathrm{Dir}(\alpha \cdot \mathbf{1}_{k_i})$    ▷ Conditional probability distribution of $v_i$ given parent configuration $\pi$

18:           **end for**

19:        **end if**

20:        $\Theta \leftarrow \Theta \cup \{\theta_{v_i|\Pi_i}\}$

21:     **end for**

22:     **return** $\mathcal{B} = ((\mathcal{V}, \mathcal{E}), \Theta)$

23: **end procedure**

---

```
System: You are an expert in {domain} who generates synthetic
    ↪ data that closely mirrors real-world {domain} data.
    ↪ Your goal is to create data that would be
    ↪ indistinguishable from real {domain} records.

Follow exactly these rules:
1. Only output the CSV data with no additional text or
    ↪ explanations
2. Always include a header row matching the schema exactly
3. Strictly adhere to the provided schema's data types and
    ↪ possible values for all fields
4. Use comma as the separator
5. Ensure all values and relationships between fields are
    ↪ realistic and statistically plausible
6. Generate diverse data while maintaining real-world
    ↪ patterns and constraints
7. Include occasional edge cases at realistic frequencies

User: Generate {num_rows} rows of data with these fields:

{schema}
```

Figure 6: The prompt template used for CSV generation with an LLM.

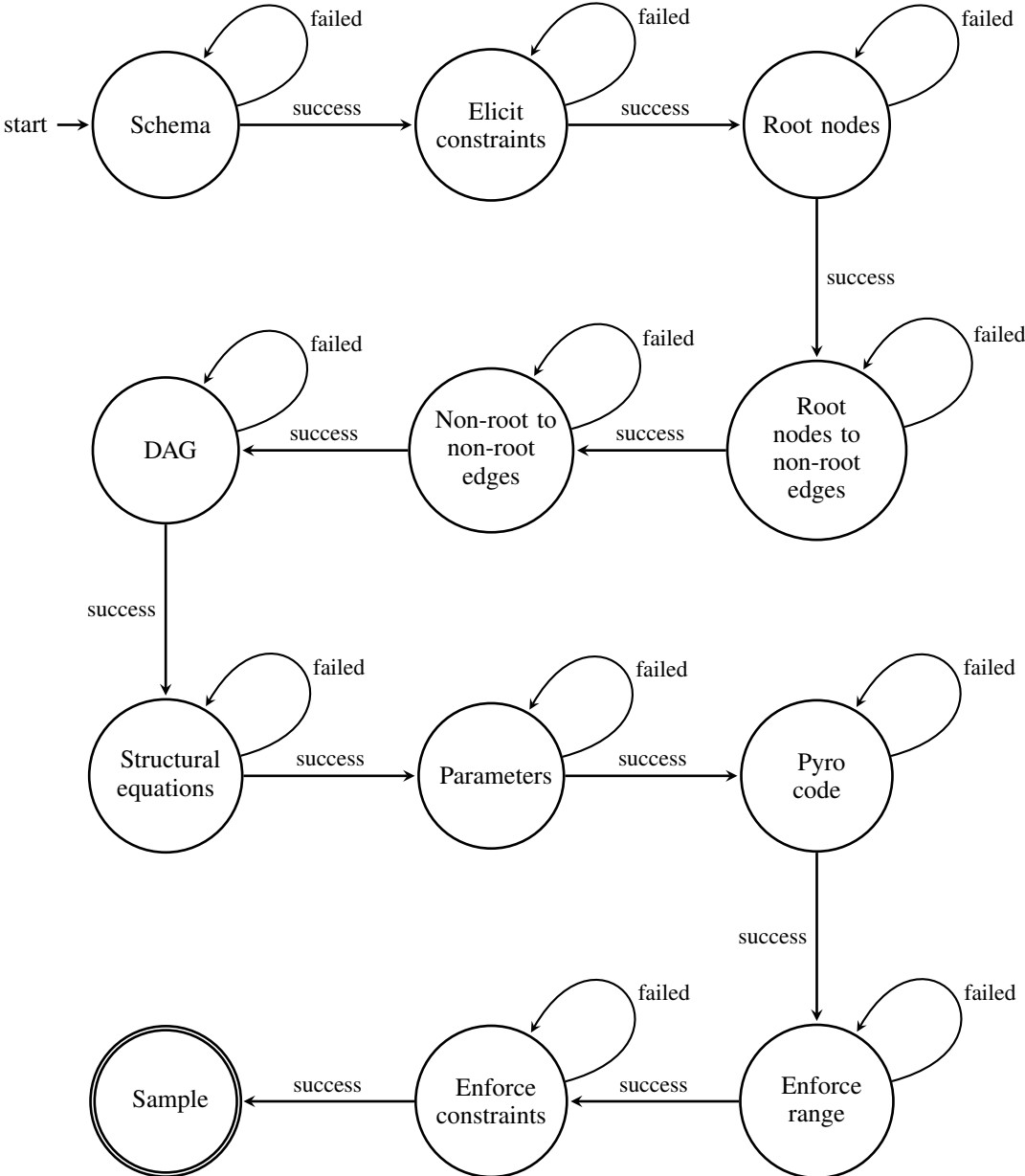

Figure 7: State machine for the SCM `Agent` showing state transitions. Each state can transition to itself upon failure or advance to the next state upon success, following a zigzag pattern.

# D   Details of Evaluation Framework

| Category | Metric | Description |
|---|---|---|
| Marginals | Total Variation Distance | Distance between the joint distributions of the original and synthetic datasets. |
| | Max 3-Way Marginal Error | Maximum absolute difference error for 3-way marginals between original and synthetic datasets, normalized by dataset size. |
| | Avg. 3-Way Marginal Error | Average absolute difference error for 3-way marginals between original and synthetic datasets, normalized by dataset size and query count. |
| | Max Binarized Marginal Error | Maximum absolute difference error for 3-way marginals after thresholding continuous variables to binary values, normalized by dataset size. |
| | Avg. Binarized Marginal Error | Average absolute difference error for 3-way marginals after thresholding continuous variables to binary values, normalized by dataset size and query count. |
| Correlations | Max Pearson Correlation Diff | Maximum absolute difference between Pearson correlation coefficients of original and synthetic datasets. |
| | Avg. Pearson Correlation Diff | Average absolute difference between Pearson correlation coefficients of original and synthetic datasets. |
| | Max Cramer's V Diff | Maximum absolute difference between Cramer's V correlation coefficients of original and synthetic datasets. |
| | Avg. Cramer's V Diff | Average absolute difference between Cramer's V correlation coefficients of original and synthetic datasets. |
| Classification | Error Rate Diff | Difference in classification error rates between models trained on original vs. synthetic data and evaluated on the same test set. |
| | AUC Diff | Difference in Area Under the ROC Curve (AUC) between models trained on original vs. synthetic data and evaluated on the same test set. |

Table 3: Overview of quality evaluation metrics for a synthetic dataset against the original dataset. All metrics range from 0 to 1, with lower values indicating better synthetic data quality.

## D.1   Tasks

### D.1.1   Task 1: Model Pretraining for Classification

A common practice in machine learning with DP is to first *pretrain* a model on public data (incurring no privacy loss) before *fine-tuning* it privately on sensitive data (using e.g., DP-SGD, incurring fixed $(\varepsilon, \delta)$-privacy loss). We apply this method to evaluate surrogate public data for binary classification tasks on tabular data (recall that this is a less common setting than public pretraining with image data [42, 104], due to a general lack of publicly available priors for tabular datasets).

We divided public and private datasets into train, validation, and test subsets using a $72 : 8 : 20$ ratio, and used an FTTransformer deep neural attention based classification model architecture (Gorishniy et al. [46]; see Appendix D.3.1 for more details). Our classification evaluation framework follows three steps: (1) standard pretraining, updating model weights with gradients calculated from (surro-

gate) public data; **(2)** DP fine-tuning on private training data; and **(3)** performance assessment on the private test data. For comparison, we include a control condition that omits the pretraining phase. We measure classification performance using AUC metric and ensure balanced datasets by downsampling the majority class to match the minority class size. We also consider an **AUC Advantage** metric, which we define as the difference in AUC between models *with* public data pretraining and a model *without* pretraining, which directly quantifies the incremental benefit provided by pretraining before private finetuning.

To account for the multiple hyperparameters in both pretraining and fine-tuning stages, we conduct a comprehensive grid search, further running each configuration 10 times to mitigate variations inherent to differential privacy training and model initialization. We analyze results using two complementary approaches: averaging performance across all hyperparameter combinations, and simulating a real-world scenario by selecting the optimal pretraining hyperparameters based on public validation data before averaging results across fine-tuning hyperparameters. Refer to Appendix D.4 for the complete hyperparameter space details.

### D.1.2 Task 2: Hyperparameter Tuning for Synthetic Data

Hyperparameters play an important role in training machine learning models, especially when differential privacy is involved [89]. While selecting the best performing hyperparameters in the non-private setting can be done with many model training runs using a validation split or cross-validation, this is not feasible in a straightforward manner with differential privacy due to the privacy loss incurred on each run. Public data may be helpful in this case, allowing researchers to run multiple experiments without consuming the privacy loss budget [57, 26].

To assess the usefulness of surrogate public data for this DP auxiliary task, we run a large-scale DP synthetic data evaluation across multiple dimensions: **(1)** datasets (including private and public splits, and various public data surrogates); **(2)** privacy loss budget $\varepsilon$; **(3)** different DP synthetic data generators (see Section D.3.2; GEM [72], AIM [80], PrivBayes [120]); and **(4)** their associated hyperparameter spaces. For each configuration, we fit a synthetic data generator and produce a synthetic dataset of the same size as the original, private data. We then evaluate across a variety of metrics, which fall into three general categories: marginal-based metrics, correlational metrics, and classification-based metrics, as shown in Table 3.

We conduct our analysis **(1)** *per synthetic data generator*, because each has a different hyperparameter space and different sensitivity to changes in hyperparameter configuration; **(2)** *per metric*, because the best-performing hyperparameter is defined with respect to a specific metric; and **(3)** *per privacy loss budget $\varepsilon$*. We quantify the degradation in performance when using the synthetic generator *on the private data* by comparing the best hyperparameter setting that we would have chosen *with the private data* (i.e., the optimal case) relative to the hyperparameters we would have chosen *with each of the (potentially surrogate) public datasets*.

To aggregate the usefulness of public data in choosing hyperparameter configurations across different evaluation metrics, we computed a *relative* performance degradation metric for each configuration. Concretely, for every private synthetic data generator, privacy level $\varepsilon$ and dataset (ACS, EDAD, and WE), we first identified the hyperparameter configuration that yielded the best performance on the private reference dataset (i.e. the real data). We then determined, for each candidate surrogate public dataset (and the regular public data), the hyperparameter configuration that *would have been chosen* based solely on its corresponding performance. Our benchmark quantifies degradation as **the relative difference between the performance achieved by the surrogate-chosen hyperparameters on the private reference and the optimal performance on the reference dataset** (measured as either absolute error or percent degradation, depending on the metric). We conducted this process independently for each metric – across classification, correlation, and marginal-based metrics. We averaged across multiple experimental seeds to obtain aggregate performance with standard error; we then conducted a Pareto frontier analysis [39] across the frontier defined by aggregating into the three metric categories: classification, correlation and marginal-based metrics.

### D.1.3 Task 3: Privacy-Utility Estimation for Synthetic Data

Understanding the privacy-utility trade-off of a mechanism for a specific private dataset is *extremely* useful for producing a differentially private release in the real world [94]. For example, it may provide guidance on setting the privacy loss budget by exposing its impact on the fidelity of private synthetic data (e.g., [4, 55]).

In this task, we evaluate how well a public dataset – either traditional or surrogate – can estimate the privacy-utility *curve* for each utility metric. This experiment is, in a sense, the "dual" of the hyperparameter tuning task described in the previous section: here, we compare the privacy-utility curve computed on the public data with the curve obtained on the private data. To mimic real-world usage, we run the DP mechanism with the best-performing hyperparameters determined from the public data (Table 3), selecting the optimal configuration independently at each tested $\varepsilon$ value.

For each dataset, synthetic data generator, and evaluation metric, we created both public-based and private-based curves over a range of privacy loss budgets $\varepsilon$. To aggregate the results across different evaluation metrics, we first compute, for each metric group (classification, correlation, and marginals) and each synthesizer (PrivBayes, AIM, and GEM), an aggregated performance value that is the average "chosen value" across all metrics in that group. For a given synthesizer and for each $\varepsilon$, we group the results by dataset and reference dataset and then pivot these averages so that each row corresponds to a dataset and each column to an $\varepsilon$ level. This representation enables us to generate line plots to visually assess the similarity between performance curves (see, e.g., Figure 28 for an example with the PrivBayes synthesizer).

Since the line plots alone are insufficient to quantify aggregate closeness, we compute both $\ell_1$ and $\ell_2$ distances between each pair of curves. The $\ell_1$ distance is more interpretable – being in the same units as the evaluation metric – while the $\ell_2$ distance is less sensitive to outliers. We average the $\ell_1$ and $\ell_2$ distances across the different metric categories (weighting each category equally). To reduce variability, each configuration is run 10 times. Finally, we perform a Pareto frontier analysis across both $\ell_1$ and $\ell_2$ distances for each dataset [39].

## D.2 Datasets

### D.2.1 ACS

The ACS data excerpt was released by the US Census Bureau in September 2020 and provided by the NIST CRC to assess synthetic data generation methods. We designated the "National" dataset (27,254 records) as the private split and the "Massachusetts" dataset (7,634 records) as the public split. Since the differential privacy synthetic data generators assessed in this project are primarily designed for categorical data, we used the "demographic" subset containing 7 categorical features provided by NIST CRC. After removing records with missing values, we retained 23,006 and 6,514 records for the private and public splits, respectively. The public split was up-sampled to match the size of the private split. For a complete description of the dataset and its curation, refer to its documentation [103].

### D.2.2 EDAD

The EDAD (Survey on Disability, Personal Autonomy and Dependency Situations) datasets were released by the Spanish National Statistics Institute (INE) in April 2022 and April 2024, containing responses from their 2020 (164,254 records) and 2023 (12,518 records) surveys respectively. We designated the 2023 survey responses as the private split and the 2020 survey responses as the public split. Since our synthetic data generators are primarily designed for categorical data, we used a subset of 11 categorical features from both surveys. After removing records with missing values, we retained 8,922 and 1,469 records for the private and public splits, respectively. The private split was down-sampled to match the size of the public split. For a complete description of the datasets and their curation, refer to the documentation given by [56].

### D.2.3 WE

The Workplace Equity Survey datasets (WE) consist of responses from two global surveys conducted in 2018 (released December 2019) and 2023 (released April 2024) by the Coalition for Diversity and Inclusion in Scholarly Communications C4DISC). We designated the 2023 survey responses (1,755 records) as the private split and the 2018 survey responses (1,182 records) as the public split. Since our synthetic data generators are primarily designed for categorical data, we used a subset of 12 categorical features from both surveys. In this dataset, we kept the missing values as another category. We retained 837 and 1,400 records for the public and private splits, respectively, and no upsampling or downsampling was done. The slight reduction in records is due to filtering response with high levels of missingness and only using respondents from the top 10 most common country affiliations in the survey (to reduce dimensionality). For a complete description of the datasets and their curation, refer to their documentation [98, 69].

### D.2.4   Dataset Memorization by the LLMs

Recent research has highlighted growing concerns that, because LLMs are exposed to benchmark data from the internet during training, their performance those and other benchmarks may be inflated when assessing performance post-training [78, 45, 91, 116, 36]. For example, it is well known that LLMs have a large capacity for training data memorization [24, 62, 25]; this is one mechanism by which they could "hack" existing benchmarks, by simply memorizing the examples and their answers. This memorization consideration is particularly relevant for our experimental setup, where we utilize LLMs to generate records both directly and indirectly. Thus, any prior exposure to our evaluation datasets (ACS, EDAD, and WE) could significantly impact model performance in our evaluations (of particular concern is exposure to the split of these datasets that we consider *private* in our evaluations, e.g., the national version of the ACS dataset). We address this memorization concern through two mitigation strategies.

First, we considered the temporal relationship between dataset releases and *model knowledge cutoff dates* when selecting two of our datasets for evaluation. Namely, the private splits of EDAD and WE were released in April 2024, which is later than the knowledge cutoff dates of most models used in our study (Table 2): GPT-4o (October 2023), Llama 3.3 70B (December 2023), and Claude 3.5 Sonnet (April 2024). While there is a one-month overlap with Claude, the analysis of Cheng et al. [29] suggests that the effective knowledge cutoff dates of LLMs typically *precede* their reported dates.

Second, we executed the LLM memorization assessment methodology proposed by [18]; they provide an extensive package & benchmark for LLM memorization detection *specific to tabular data*. We ran their assessment across all private and public splits. In the data generation tests from [18] – the most relevant to our setting – both header tests (generating the first few rows) and row completion tests (generating random-location rows) indicated *no evidence* of record-level memorization by any of the three LLMs across all datasets. Refer to Figure 8 for an example of the header test results for the ACS dataset with Claude 3.5 Sonnet.

Additional tests examining an LLM's metadata knowledge of tabular datasets, rather than record generation capabilities, revealed varying levels of dataset familiarity. The models unsurprisingly demonstrated strong familiarity with ACS, but limited knowledge of EDAD and minimal recognition of WE. This pattern aligns with the relative public visibility of these datasets: ACS is a core and official product of the US Census, EDAD is an official product of the Spanish National Statistics Institute, and WE is a small-scale survey conducted by a coalition of professional and trade organizations.

When provided with header columns and the first few rows, all models successfully identified the name of the ACS dataset, and sometimes could identify the EDAD dataset name (where the 2020 public split consists of multiple raw files). However, for the WE dataset, even when given headers and first rows, no model generated the correct dataset name  instead, they provided thematically related names such as "work-life-and-career-survey" and "publishing-industry-diversity-survey." We hypothesize that this pattern emerges from the survey questions themselves serving as column names, which inherently reveal the overall topic of the survey (e.g., "How long have you worked in publishing and/or related industries?").

We observed similar patterns regarding column name completion. When given the dataset name and the first few features, all models failed to generate the correct column names for both EDAD and WE datasets. For ACS, the models could generate some of the column names, but not in the correct order. We hypothesize that this is due to the fact that the ACS datasets we used were sub-sampled, modified, and adopted from the US Census release by NIST.

### D.3   Private Mechanisms

### D.3.1   Classification

Differentially private pretraining is usually conducted in domains where strong, publicly available priors with matching data-dimensionality are available (e.g., text or image data). In these fields, neural transformer models dominate [114, 64].

For an adequate analog to this space in the tabular setting, we consider an FTTransformer model [46], which is a transformer based architecture for tabular data classification. FTTransformer has demonstrated strong performance against established powerful gradient boosting approaches such

```
PUMA,AGEP,SEX,MSP,HISP,RAC1P,NOC,NPF,HOUSING_TYPE,OWN_RENT,DENSITY,INDP,
    INDP_CAT,EDU,PINCP,PINCP_DECILE,POVPIP,DVET,DREM,DPHY,DEYE,DEAR,PWGTP,
    WGTP
01-01301,18,2,6,0,9,N,N,3,0,2731.2,N,N,7,0.0,0,N,N,2,2,2,2,79,0
01-01301,27,1,6,0,1,N,N,3,0,2731.2,3291,4,7,15400.0,4,116,N,2,2,2,2,5,0
01-01301,74,2,3,0,2,N,N,2,0,2731.2,N,N,9,12900.0,3,N,N,2,1,2,2,19,0
01-01301,22,1,6,0,1,N,N,3,0,2731.2,N,N,7,0.0,0,N,N,2,2,2,2,10,0
01-01301,18,2,6,0,1,N,N,3,0,2731.2,N,N,7,0.0,0,N,N,2,2,2,2,15,0
01-01301,52,2,1,0,1,N,N,1,1,2731.2,7860,8,10,52000.0,8,433,N,2,2,2,2,25,0
01-01301,54,1,1,0,1,N,N,1,1,2731.2,7860,8,10,55000.0,8,458,N,2,2,2,2,25,0
01-01301,20,2,6,0,1,N,N,3,0,2731.2,N,N,7,35400.0,0,N,N,2,2,2,2,12,0
01-01301,48,2,1,0,1,N,N,1,1,2731.2,8680,9,10,45000.0,7,375,N,2,2,2,2,20,0
01-01301,49,1,1,0,1,N,N,1,1,2731.2,7860,8,9,48000.0,7,400,N,2,2,2,2,20,0
01-01301,15,1,6,0,1,N,N,3,0,2731.2,N,N,6,9300.0,0,N,N,2,2,2,2,18,0
01-01301,45,2,1,0,1,N,N,1,1,2731.2,N,N,5,27860.0,8,N,N,2,1,0,2,1420
01-01.01,27,1,350,1,N,2,2,2,27,1.8,0
```

Figure 8: The header test output on the ACS dataset on Claude 3.5 Sonnet. The LLM is prompted with the column names as well as a few *first rows* of the dataset (black), and its completion is presented. The output is colored according to its Levenshtein string distance compared to the original records: correct, incorrect, and missing. We observe that the LLM failed to reproduce the header, as many errors occur within columns with variability.

as XGBoost [28]. Its effectiveness stems from specialized data transformations that mitigate information loss in transformer-based attention layers [47]. Prior work shows how simple it can be to adapt FTTransformer to the private setting [95] by making minor modifications to its architecture to support DP-SGD [1]. Importantly, it can also be easily *pre-trained* with public data through standard gradient updates *before* private training. The differentially private variant of FTTransformer is $(\varepsilon, \delta)$-DP, for which we set $\delta = 10^{-5}$.

### D.3.2 Data Synthesis

We considered three representative state-of-the-art private data release methods: PrivBayes [120], GEM [73] and AIM [80]. Each of these synthesizers follows the "Select-Measure-Project" paradigm, in that they *privately* select statistical queries (marginals or correlations) to run on a sensitive distribution, *privately* measure these queries, and then as *post-processing* project these measurements onto a synthetic distribution (from which we can draw arbitrary samples) that approximates the original, sensitive distribution.

PrivBayes builds a Bayesian network (BN) and adds noise to all $k$-way correlations to ensure differential privacy. Despite having been published in 2017, PrivBayes is still considered state-of-the-art and was chosen to produce the differentially private release of the Israel National Live Birth Registry [55]. GEM parameterizes a neural model to represent a synthetic distribution that approximates the true distribution by minimizing a linear query error based loss (with linear queries implemented as $k$-way marginals, where by default $k = 3$). AIM relies on the Private-PGM graphical model [79] to parameterize the underlying distribution, and utilizes an iterative process to take advantage of higher values of $\varepsilon$. Both AIM and GEM are considered the state-of-the-art approaches to generating private synthetic data [102, 93]. Outside of these methods, we acknowledge that many other methods exist for generating DP data [37, 51, 109, 81, 117, 92, 10, 23], but we believe that PrivBayes, GEM and AIM are a representative set of what can be currently considered state-of-the-art.

PrivBayes and GEM are $\varepsilon$-DP, whereas AIM is $(\varepsilon, \delta)$-DP, for which we set $\delta = 10^{-9}$. All three methods come with hyperparameters that need to be tuned. Detailed lists of hyperparameters per-synthetic data generator, and their associated values, are given in Appendix D.4.

## D.4  Hyperparameter Spaces

Table 4: Hyperparameters for FTTransformer Classifier

| Hyperparameter | Description | Values |
|---|---|---|
| pre_num_epochs | Number of epochs for pre-training | $\{1, 9\}$ |
| pre_batch_size | Batch size for pre-training | $\{32, 128\}$ |
| pre_lr | Learning rate for pre-training | $\{3 \times 10^{-4}, 3 \times 10^{-5}\}$ |
| dp_num_epochs | Number of epochs for differential private fine-tuning | 20 |
| dp_batch_size | Batch size for differential private fine-tuning | 128 |
| dp_lr | Learning rate for differential private fine-tuning | $\{3 \times 10^{-3}, 3 \times 10^{-4}\}$ |

Table 5: Hyperparameters for GEM

| Hyperparameter | Description | Values |
|---|---|---|
| k | Maximum degree of measured marginals | $\{2, 3\}$ |
| T | Number of iterations | $\{50, 100\}$ |
| alpha | Learning rate | $\{0.1, 0.5\}$ |
| ema_weights_beta | EMA weights coefficient | $\{0.1, 0.9\}$ |

Table 6: Hyperparameters for AIM

| Hyperparameter | Description | Values |
|---|---|---|
| degree | Maximum degree of measured marginals | $\{2, 3\}$ |
| rounds | Number of iterations | $\{20, 40\}$ |

Table 7: Hyperparameters for PrivBayes

| Hyperparameter | Description | Values |
|---|---|---|
| theta | SNR heuristic to set max node degree | $\{2, 8, 32, 64\}$ |
| epsilon_split | Prop. of privacy budget allocated to structure learning | $\{0.1, 0.5, 0.75\}$ |

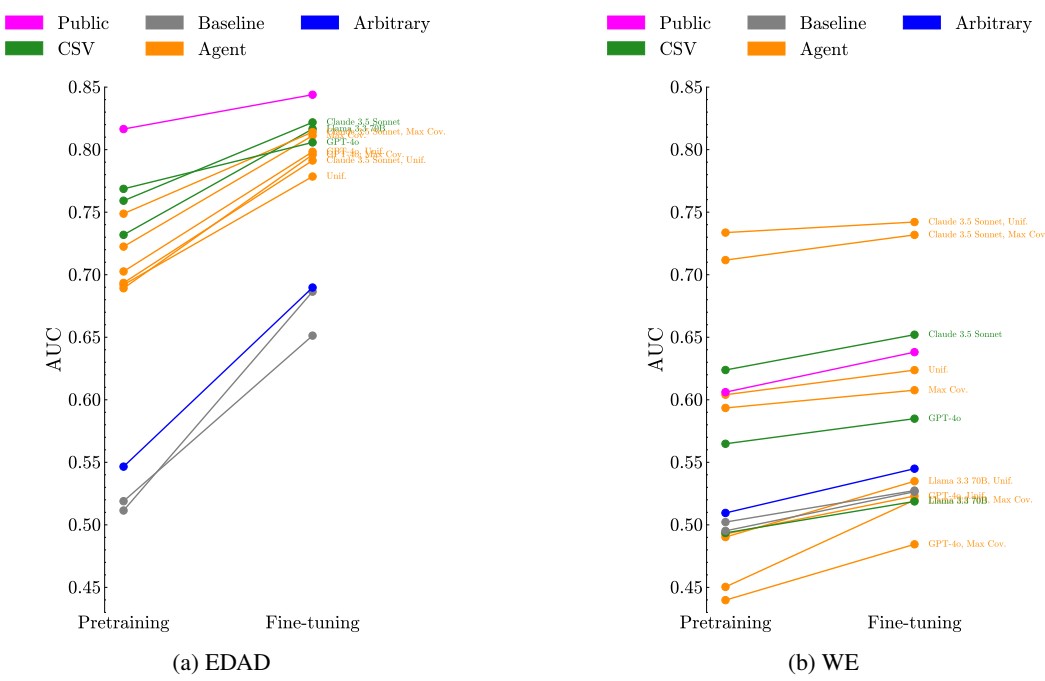

(a) EDAD        (b) WE

Figure 9: Mean AUC on the test subset of the private dataset split for the pretraining model and the fine-tuned model, grouped by generation method. The mean is calculated across the DP finetuning hyperparameter space when best pretraining hyperparameter configuration is chosen for the pretraining step, with 10 runs per hyperparameter configuration.

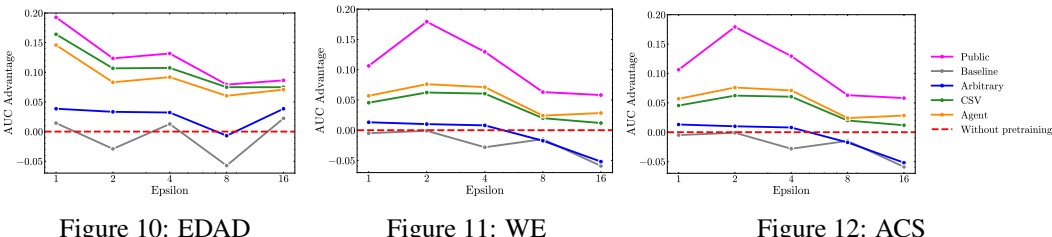

Figure 10: EDAD      Figure 11: WE      Figure 12: ACS

Figure 13: Mean AUC Advantage of the DP model after pretraining, grouped by generation method. The mean is calculated across the DP finetuning hyperparameter space when best pretraining hyperparameter configuration is chosen for the pretraining step, with 10 runs per hyperparameter configuration.

# E    Additional Result Discussion

In this section, we present the results of our evaluation framework (Section 4) for the following DP auxiliary tasks: pretraining (Section E.1), hyperparameter tuning (Section E.2), and estimating the privacy-utility trade-off (Section E.4). Appendix F provides additional results details.

All of the experiments were done with $\varepsilon \in \{1, 2, 4, 8, 16\}$, and each hyperparameter configuration (Appendix D.4) was run 10 times.

### E.1    Results for Task 1: Pretraining for DP Classification

In our analysis, the best pretraining hyperparameter configuration was selected based on the public validation subset (see Figure 15 and Table 24a for full hyperparameter averaging results, which show similar trends).

**EDAD and WE.**    Overall, we find strong evidence that LLM-based methods – both `CSV` and `Agent` surrogate public data generation (particularly with Claude 3.5 Sonnet) – offer a competitive alterna-

Table 8: Mean AUC Advantage (AUC in parentheses) of the DP model after pretraining, grouped by generation method. The mean is calculated across the DP finetuning hyperparameter space *when the best pretraining hyperparameter configuration is chosen* for the pretraining step, with 10 runs per hyperparameter configuration.

(a) ACS

| Method | $\varepsilon = 1$ | $\varepsilon = 2$ | $\varepsilon = 4$ | $\varepsilon = 8$ | $\varepsilon = 16$ |
|---|---|---|---|---|---|
| Without pretraining | .00 (.74) | .00 (.74) | .00 (.74) | .00 (.75) | .00 (.75) |
| Public | .01 (.75) | .01 (.76) | .01 (.76) | .00 (.75) | .00 (.75) |
| Baseline (Domain) | -.03 (.71) | -.03 (.71) | -.03 (.71) | -.03 (.72) | -.05 (.70) |
| Baseline (Univariate) | -.01 (.73) | .00 (.74) | -.03 (.71) | -.02 (.73) | .00 (.75) |
| Arbitrary | .00 (.74) | .01 (.75) | .00 (.74) | .00 (.75) | .00 (.75) |
| CSV (Claude 3.5 Sonnet) | .01 (.74) | .01 (.75) | .01 (.75) | .00 (.75) | .01 (.76) |
| CSV (GPT-4o) | .01 (.74) | .01 (.75) | .01 (.76) | .00 (.75) | .01 (.76) |
| CSV (Llama 3.3 70B) | .01 (.75) | .01 (.75) | .01 (.75) | .00 (.75) | .01 (.76) |
| Agent (Claude 3.5 Sonnet, Unif.) | .01 (.74) | .01 (.75) | .01 (.75) | .00 (.75) | .00 (.75) |
| Agent (Claude 3.5 Sonnet, Max Cov.) | .01 (.74) | .00 (.75) | .01 (.75) | .00 (.75) | .00 (.75) |
| Agent (GPT-4o, Unif.) | .00 (.74) | .00 (.75) | .01 (.75) | .00 (.75) | .00 (.75) |
| Agent (GPT-4o, Max Cov.) | .00 (.74) | .00 (.74) | .00 (.75) | .00 (.75) | .00 (.75) |
| Agent (Llama 3.3 70B, Unif.) | -.01 (.73) | .00 (.74) | .01 (.75) | .00 (.75) | .00 (.75) |
| Agent (Llama 3.3 70B, Max Cov.) | .00 (.74) | .01 (.75) | .01 (.75) | .00 (.75) | .01 (.76) |
| Agent (All, Unif.) | .01 (.75) | .01 (.75) | .01 (.75) | .01 (.76) | .01 (.76) |
| Agent (All, Max Cov.) | .01 (.74) | .01 (.75) | .01 (.75) | .00 (.75) | .00 (.75) |

(b) EDAD

| Method | $\varepsilon = 1$ | $\varepsilon = 2$ | $\varepsilon = 4$ | $\varepsilon = 8$ | $\varepsilon = 16$ |
|---|---|---|---|---|---|
| Without pretraining | .00 (.65) | .00 (.69) | .00 (.71) | .00 (.74) | .00 (.76) |
| Public | .19 (.84) | .12 (.81) | .13 (.85) | .08 (.82) | .09 (.85) |
| Baseline (Domain) | .00 (.65) | .00 (.69) | -.02 (.70) | -.04 (.70) | .02 (.78) |
| Baseline (Univariate) | .04 (.69) | -.06 (.63) | .05 (.76) | -.07 (.67) | .03 (.79) |
| Arbitrary | .04 (.69) | .03 (.72) | .03 (.75) | -.01 (.74) | .04 (.80) |
| CSV (Claude 3.5 Sonnet) | .17 (.82) | .12 (.80) | .10 (.81) | .07 (.82) | .07 (.83) |
| CSV (GPT-4o) | .15 (.81) | .09 (.77) | .11 (.83) | .07 (.81) | .07 (.83) |
| CSV (Llama 3.3 70B) | .17 (.82) | .12 (.80) | .12 (.83) | .08 (.82) | .08 (.84) |
| Agent (Claude 3.5 Sonnet, Unif.) | .14 (.79) | .10 (.79) | .10 (.81) | .06 (.81) | .07 (.82) |
| Agent (Claude 3.5 Sonnet, Max Cov.) | .16 (.81) | .10 (.79) | .09 (.81) | .06 (.80) | .08 (.84) |
| Agent (GPT-4o, Unif.) | .15 (.80) | .05 (.74) | .10 (.81) | .06 (.81) | .07 (.83) |
| Agent (GPT-4o, Max Cov.) | .14 (.80) | .08 (.77) | .07 (.78) | .04 (.79) | .07 (.83) |
| Agent (All, Unif.) | .13 (.78) | .09 (.78) | .08 (.79) | .07 (.81) | .07 (.83) |
| Agent (All, Max Cov.) | .16 (.81) | .07 (.76) | .12 (.84) | .07 (.81) | .07 (.83) |

(c) WE

| Method | $\varepsilon = 1$ | $\varepsilon = 2$ | $\varepsilon = 4$ | $\varepsilon = 8$ | $\varepsilon = 16$ |
|---|---|---|---|---|---|
| Without pretraining | .00 (.53) | .00 (.55) | .00 (.58) | .00 (.63) | .00 (.66) |
| Public | .11 (.64) | .18 (.73) | .13 (.71) | .06 (.69) | .06 (.72) |
| Baseline (Domain) | -.01 (.53) | -.01 (.54) | -.06 (.52) | -.06 (.57) | -.07 (.59) |
| Baseline (Univariate) | .00 (.53) | .02 (.58) | .00 (.58) | .01 (.64) | -.05 (.61) |
| Arbitrary | .01 (.55) | .01 (.56) | .01 (.59) | -.02 (.61) | -.05 (.61) |
| CSV (Claude 3.5 Sonnet) | .12 (.65) | .09 (.65) | .09 (.67) | .02 (.65) | .05 (.70) |
| CSV (GPT-4o) | .05 (.58) | .08 (.64) | .06 (.64) | .05 (.69) | .03 (.69) |
| CSV (Llama 3.3 70B) | -.01 (.52) | .01 (.57) | .04 (.61) | .00 (.63) | -.04 (.62) |
| Agent (Claude 3.5 Sonnet, Unif.) | .21 (.74) | .15 (.70) | .17 (.75) | .07 (.70) | .11 (.77) |
| Agent (Claude 3.5 Sonnet, Max Cov.) | .20 (.73) | .17 (.72) | .15 (.73) | .06 (.69) | .07 (.73) |
| Agent (GPT-4o, Unif.) | -.01 (.52) | .02 (.58) | -.01 (.57) | -.04 (.59) | -.06 (.60) |
| Agent (GPT-4o, Max Cov.) | -.05 (.48) | -.05 (.50) | -.07 (.51) | -.05 (.58) | -.02 (.63) |
| Agent (Llama 3.3 70B, Unif.) | .00 (.54) | .03 (.59) | .04 (.62) | .02 (.66) | .02 (.68) |
| Agent (Llama 3.3 70B, Max Cov.) | -.01 (.52) | -.01 (.54) | .02 (.60) | .02 (.65) | .02 (.68) |
| Agent (All, Unif.) | .09 (.62) | .15 (.71) | .12 (.70) | .05 (.68) | .07 (.73) |
| Agent (All, Max Cov.) | .08 (.61) | .14 (.69) | .13 (.71) | .07 (.71) | .06 (.72) |

tive to traditional public data. Figure 9 presents our experimental results on the WE and EDAD ($\varepsilon = 1$), demonstrating how pretraining on the surrogate public data can vastly improve the starting point of model performance.

Figure 13 shows a diminishing pretraining advantage when increasing $\varepsilon$ for both EDAD and WE. This is an expected behavior: high epsilon allows for the extraction of more signal from the private dataset, and may reduce the usefulness of public data, regular or surrogate [104].

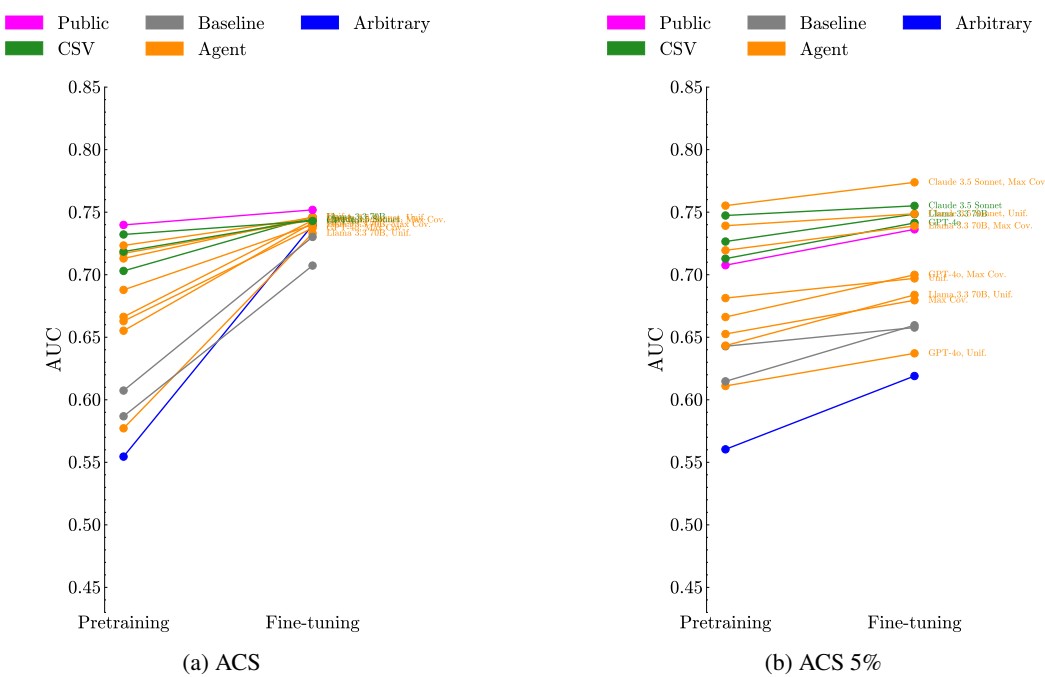

(a) ACS                  (b) ACS 5%

Figure 14: Mean AUC on the test subset of the private dataset split for the pretraining model and the fine-tuned model, grouped by generation method. The mean is calculated across the DP finetuning hyperparameter space when best pretraining hyperparameter configuration is chosen for the pretraining step, with 10 runs per hyperparameter configuration.

Under a more granular analysis, the EDAD dataset benefits substantially from pretraining, with average AUC advantages per method ranging from 0.09 to 0.19. Here, the traditional public dataset delivers the highest improvement across $\varepsilon$ values. When aggregated by generation method, `CSV`-based methods perform slightly worse than the regular public dataset, followed by the `Agent`-based method. A more careful examination of surrogate approaches in Table 8b reveals that the `CSV` (`Claude`) (AUC advantages ranging from 0.07-0.17) and `CSV` (`Llama`) (ranging from 0.08-0.17) perform on par with or slightly worse than the regular public data (ranging from 0.09-0.19). For example, at $\varepsilon = 1$, the AUC advantages of traditional public data, `CSV` (`Claude`), `CSV` (`Llama`) are 0.19 and 0.17, respectively. As expected, pretraining with baselines (`Uniform` and `Univariate`) and `Arbitrary` yields almost no benefit, because they contain essentially no signal about the relationship between the target variable and the features in the classification task.

The WE dataset exhibits trends similar to EDAD. Although the traditional public dataset achieves the best advantage at $\varepsilon = 2$, its performance is not consistently top-ranked across all privacy levels. In fact, for $\varepsilon = 1, 4, 16$, it is not in the top three. Notably, the two Claude `Agent`-based variants have the best performance across most $\varepsilon$ values, with AUC improvements ranging from 0.07 to 0.21.

Table 9: Dataset similarity assessment against the private data for ACS, EDAD and WE. The datasets are evaluated based on two distance metrics (Section G.1): (1) Total Variation Distance (TVD); and (2) Average error on 3-Way Marginals (3WM). Both metrics are in range $[0, 1]$, inverted to represent similarity $(1 - x)$, and scaled by 100. Zero values (rounded) are omitted for readability.

| Method | ACS | | EDAD | | WE | |
|---|---|---|---|---|---|---|
| | 1-TVD | 1-3WM | 1-TVD | 1-3WM | 1-TVD | 1-3WM |
| Public | 48.5 | 50.4 | 4.9 | 26.1 | 6.7 | 34.1 |
| Baseline (Domain) | 4.3 | | 0.1 | | 0.2 | |
| Baseline (Univariate) | 44.6 | 63.8 | 7.1 | 66.7 | 15.4 | 78.5 |
| Arbitrary | 2.8 | | 0.1 | | | |
| CSV (Claude 3.5 Sonnet) | 14.4 | 15.0 | | | | 10.9 |
| CSV (GPT-4o) | 25.7 | 30.2 | | 11.5 | | 14.2 |
| CSV (Llama 3.3 70B) | 16.6 | 10.0 | | | | 2.4 |
| Agent (Claude 3.5 Sonnet, Unif.) | 41.5 | 48.3 | | 5.5 | | 11.7 |
| Agent (Claude 3.5 Sonnet, Max Cov.) | 40.1 | 40.0 | | 6.8 | | 8.0 |
| Agent (GPT-4o, Unif.) | 27.3 | 23.3 | | 7.2 | | |
| Agent (GPT-4o, Max Cov.) | 27.4 | 20.4 | | 6.9 | | |
| Agent (Llama 3.3 70B, Unif.) | 13.8 | | | | | |
| Agent (Llama 3.3 70B, Max Cov.) | 10.3 | | | | | |
| Agent (All, Unif.) | 30.5 | 26.6 | | | | |
| Agent (All, Max Cov.) | 24.6 | 15.7 | | | | |

| Method | Classification | Correlation | Marginals |
|---|---|---|---|
| CSV (Claude) | 0.002 | 0.033 | 0.121 |
| CSV (GPT) | 0.001 | 0.149 | 0.096 |
| CSV (Llama) | 0.003 | 0.052 | 0.041 |
| Agent (Llama, Unif.) | 0.002 | 0.061 | 0.086 |

Table 10: Pareto Efficient Methods (Task 2: Hyperparameter tuning for private synthetic data) for PrivBayes on ACS.

| Method | Classification | Correlation | Marginals |
|---|---|---|---|
| Public | 0.008 | 0.047 | 0.097 |
| CSV (Claude) | 0.033 | 0.046 | 0.227 |
| Agent (Claude, Unif.) | 0.004 | 0.134 | 0.225 |

Table 11: Pareto Efficient Methods (Task 2: Hyperparameter tuning for private synthetic data) for PrivBayes on EDAD.

| Method | Classification | Correlation | Marginals |
|---|---|---|---|
| Arbitrary (Baseline) | 0.043 | 0.052 | 0.056 |
| Agent (All, Max Cov.) | 0.016 | 0.096 | 0.172 |
| Agent (Claude, Unif.) | 0.019 | 0.040 | 0.070 |

Table 12: Pareto Efficient Methods (Task 2: Hyperparameter tuning for private synthetic data) for PrivBayes on WE.

| Method | Classification | Correlation | Marginals |
|---|---|---|---|
| CSV (Claude) | 0.013 | 0.002 | 0.045 |
| Agent (Claude, Max Cov.) | 0.010 | 0.003 | 0.125 |
| Agent (Claude, Unif.) | 0.004 | 0.003 | 0.024 |

Table 13: Pareto Efficient Methods (Task 2: Hyperparameter tuning for private synthetic data) for AIM on ACS.

## E.2 Results for Task 2: Hyperparameter Tuning for DP Synthetic Data Generation

**ACS.** On ACS, where LLMs are likely to possess well-calibrated priors due to extensive training on U.S. Census data (see Appendix D.2.4), the AIM synthesizer (Table 13) shows that `Agent (Claude, Unif.)` is best for both classification (0.004) and marginal consistency (0.024), while `CSV (Claude)` has the best correlation metric (0.002) (although the `Agent` based Claude methods here are close behind). For the PrivBayes synthesizer (Table 10), the CSV-based approaches are impressive: `CSV (GPT)` achieves the best in terms of classification metrics (0.001), `CSV (Llama)` is best in marginal metrics (0.041), and `CSV (Claude)` is best for correlation metrics (0.033). For GEM on ACS, the `Agent (Claude, Max Cov.)` approach is dominant along with the `Arbitrary` baseline. Recall that the `Arbitrary` baseline *directly* encodes relationships into the data (via the Bayesian approach described in Appendix C.1.3). In the case of GEM, whether relationships between variables are accurate to *true* relationships in the private data is less important when tuning its hyperparameters.

**EDAD.** We now turn to the EDAD dataset (a Spanish disability survey); EDAD was published after many LLMs training cutoffs, so we expect the LLMs to have less, if any, prior exposure to tabular data in the same domain as the schema we present. For the AIM synthesizer (Table 14), *several* agent-based methods (e.g., `Agent (All, Unif.)`) are similarly strong for classification metrics (0.001). Although the correlation metrics are tightly grouped (ranging from 0.014 to 0.019), the overall Pareto frontier is defined by a mix of the CSV and Agent approaches. For the PrivBayes synthesizer (Table 11), the agent-based method `Agent (Claude, Unif.)` again leads on classification (0.004) while `CSV (Claude)` remains on the Pareto frontier for correlation (0.046); meanwhile, the real `Public` data yields the best marginal consistency (0.097).

**WE.** For WE – the Workplace Equity survey dataset, also from a period after many LLM training cutoffs – for the AIM synthesizer (Table 15) the best-performing methods are exclusively Agent-based methods. Here, `Agent (Claude, Unif.)` leads in classification metrics (0.016), `Agent (GPT, Unif.)` attains the best correlation metrics (0.007), and `Agent (GPT, Max Cov.)` provides the strongest marginal consistency (0.025). In contrast, for the PrivBayes synthesizer (Table 12), although the `Arbitrary` baseline dominates on marginal consistency (0.056) and is competitive on correlation (0.052), agent-based methods (both `All, Max Cov.` and `Claude, Unif.`) yield a substantial improvement in classification performance (0.016 - 0.019).

## E.3 DP-GAN Results

We also tested DP-GAN as a potential DP synthesizer for hyper-parameter tuning. However, the poor performance of the model on the tasks we considered was such that we decided not to pursue additional results or consider it along the mainline, state-of-the-art DP synthetic data methods (which included GEM, a similar neural generative adversarial synthetic data method). We include some limited results here in case they are of interest to the reader, in Tables 19 and 20.

| Method | Classification | Correlation | Marginals |
|---|---|---|---|
| CSV (Claude) | 0.004 | 0.014 | 0.037 |
| CSV (Llama) | 0.003 | 0.018 | 0.010 |
| Agent (All, Max Cov.) | 0.001 | 0.019 | 0.013 |
| Agent (All, Unif.) | 0.001 | 0.018 | 0.040 |
| Agent (Claude, Max Cov.) | 0.004 | 0.014 | 0.010 |
| Agent (Claude, Unif.) | 0.003 | 0.014 | 0.025 |
| Agent (GPT, Max Cov.) | 0.003 | 0.017 | 0.012 |
| Agent (GPT, Unif.) | 0.003 | 0.015 | 0.011 |

Table 14: Pareto Efficient Methods (Task 2: Hyperparameter tuning for private synthetic data) for AIM on EDAD.

| Method | Classification | Correlation | Marginals |
|---|---|---|---|
| Agent (Claude, Unif.) | 0.016 | 0.016 | 0.198 |
| Agent (GPT, Max Cov.) | 0.033 | 0.013 | 0.025 |
| Agent (GPT, Unif.) | 0.020 | 0.007 | 0.030 |
| Agent (Llama, Unif.) | 0.017 | 0.010 | 0.047 |

Table 15: Pareto Efficient Methods (Task 2: Hyperparameter tuning for private synthetic data) for AIM on WE.

| Method | Classification | Correlation | Marginals |
|---|---|---|---|
| Arbitrary (Baseline) | 0.002 | 0.023 | 0.043 |
| Agent (Claude, Max Cov.) | 0.002 | 0.039 | 0.072 |

Table 16: Pareto Efficient Methods (Task 2: Hyperparameter tuning for private synthetic data) for GEM on ACS.

| Method | Classification | Correlation | Marginals |
|---|---|---|---|
| Public | 0.008 | 0.222 | 0.146 |
| CSV (GPT) | 0.004 | 0.172 | 0.166 |
| Agent (Claude, Max Cov.) | 0.007 | 0.104 | 0.147 |

Table 17: Pareto Efficient Methods (Task 2: Hyperparameter tuning for private synthetic data) for GEM on EDAD.

| Method | Classification | Correlation | Marginals |
|---|---|---|---|
| CSV (LLaMA) | 0.025 | 0.057 | 0.521 |
| Agent (All, Unif.) | 0.007 | 0.071 | 0.059 |
| Agent (GPT, Unif.) | 0.028 | 0.056 | 0.058 |

Table 18: Pareto Efficient Methods (Task 2: Hyperparameter tuning for private synthetic data) for GEM on WE.

Table 19: Performance of surrogate and baseline methods for **DP-GAN** hyperparameter tuning on **EDAD**. Each value denotes percentage degradation (lower is better) relative to the private-data optimum for (i) downstream classification, (ii) pairwise correlation, and (iii) marginal preservation.

| Method | % Degradation (Class.) | % Degradation (Corr.) | % Degradation (Marg.) |
|---|---|---|---|
| Public | 0.144 | 0.027 | 0.006 |
| CSV (Claude 3.5 Sonnet) | 0.179 | 0.028 | 0.028 |
| Agent (Max Cov.) | 0.193 | 0.038 | 0.034 |
| Agent (Claude 3.5 Sonnet, Max Cov.) | 0.199 | 0.028 | 0.025 |
| CSV (GPT-4o) | 0.207 | 0.025 | 0.021 |
| Agent (Claude 3.5 Sonnet, Unif.) | 0.212 | 0.058 | 0.039 |
| Agent (Unif.) | 0.262 | 0.034 | 0.015 |
| Arbitrary | 0.265 | 0.073 | 0.028 |
| Baseline (Univariate) | 0.242 | 0.081 | 0.011 |
| Baseline (Domain) | 0.277 | 0.083 | 0.031 |
| Agent (GPT-4o, Unif.) | 0.295 | 0.057 | 0.027 |
| CSV (Llama 3.3 70B) | 0.296 | 0.024 | 0.024 |
| Agent (GPT-4o, Max Cov.) | 0.351 | 0.032 | 0.028 |

Table 20: Pareto-optimal methods for DP-GAN hyperparameter tuning on **EDAD**. These methods form the frontier for the three degradation metrics.

| Method | Classification | Correlation | Marginals |
|---|---|---|---|
| Public | 0.144 | 0.027 | 0.006 |
| CSV (GPT-4o) | 0.207 | 0.025 | 0.021 |
| CSV (Llama 3.3 70B) | 0.296 | 0.024 | 0.024 |

### E.4 Results for Task 3: Privacy-Utility Trade-off Estimation for DP Synthetic Data Generation

As shown in Table 21, Table 22, and Table 23, the distances between the performance vectors – measured in both $\ell_1$ and $\ell_2$ norms – vary considerably across datasets and synthesizers. For example, in the AIM synthesizer (Table 21), methods such as `Agent (All, Max Cov.)` achieve an ACS $\ell_1$ of 0.039 and an ACS $\ell_2$ of 0.023, while `CSV (Llama)` attains similar values (ACS $\ell_1$: 0.044, ACS $\ell_2$: 0.023). In the GEM setting (Table 22), a similar trend is observed. Here, the `Arbitrary` baseline exhibits impressively low EDAD $\ell_1$ (0.028) and EDAD $\ell_2$ (0.013) distances, while other methods, such as `Agent (All, Unif.)` and `CSV (GPT)`, also display competitive performance on certain metrics. For the PrivBayes synthesizer (Table 23), `CSV (GPT)` achieves an ACS $\ell_1$ of 0.091 and an ACS $\ell_2$ of 0.042 – values that are generally lower than those produced by several agent-based approaches on other metrics.

| Method | ACS $\ell_1$ | EDAD $\ell_1$ | WE $\ell_1$ | ACS $\ell_2$ | EDAD $\ell_2$ | WE $\ell_2$ |
|---|---|---|---|---|---|---|
| Arbitrary (Baseline) | 0.353 | 0.510 | 0.367 | 0.184 | 0.231 | 0.166 |
| Agent (All, Max Cov.) | 0.364 | 0.258 | 0.330 | 0.186 | 0.116 | 0.148 |
| Agent (All, Unif.) | 0.519 | 0.126 | 0.373 | 0.274 | 0.057 | 0.168 |
| Agent (Claude, Unif.) | 0.705 | 0.257 | 0.254 | 0.355 | 0.115 | 0.124 |
| Agent (Llama, Max Cov.) | 0.543 | 0.559 | 0.260 | 0.288 | 0.251 | 0.119 |
| Agent (Llama, Unif.) | 0.337 | 0.696 | 0.295 | 0.176 | 0.312 | 0.133 |

Table 21: Priv/Util Pareto Efficient Methods (Task 3: Privacy/utility tradeoff estimation) for AIM.

| Method | ACS $\ell_1$ | EDAD $\ell_1$ | WE $\ell_1$ | ACS $\ell_2$ | EDAD $\ell_2$ | WE $\ell_2$ |
|---|---|---|---|---|---|---|
| Univariate (Baseline) | 0.321 | 0.028 | 0.294 | 0.144 | 0.013 | 0.133 |
| CSV (GPT) | 0.091 | 0.155 | 0.402 | 0.042 | 0.070 | 0.180 |
| Agent (All, Max Cov.) | 0.094 | 0.071 | 0.318 | 0.043 | 0.033 | 0.144 |
| Agent (All, Unif.) | 0.112 | 0.051 | 0.280 | 0.051 | 0.024 | 0.126 |
| Agent (GPT, Max Cov.) | 0.127 | 0.061 | 0.232 | 0.058 | 0.027 | 0.105 |

Table 22: Priv/Util Pareto Efficient Methods (Task 3: Privacy/utility tradeoff estimation) for GEM.

| Method | ACS $\ell_1$ | EDAD $\ell_1$ | WE $\ell_1$ | ACS $\ell_2$ | EDAD $\ell_2$ | WE $\ell_2$ |
|---|---|---|---|---|---|---|
| CSV (Llama) | 0.044 | 0.387 | 0.376 | 0.023 | 0.188 | 0.171 |
| Agent (All, Max Cov.) | 0.039 | 0.100 | 0.191 | 0.023 | 0.051 | 0.092 |
| Agent (All, Unif.) | 0.082 | 0.091 | 0.167 | 0.041 | 0.056 | 0.081 |
| Agent (Claude, Max Cov.) | 0.068 | 0.152 | 0.111 | 0.033 | 0.085 | 0.063 |
| Agent (Claude, Unif.) | 0.070 | 0.151 | 0.114 | 0.035 | 0.082 | 0.059 |
| Agent (GPT, Max Cov.) | 0.065 | 0.164 | 0.194 | 0.034 | 0.099 | 0.092 |
| Agent (Llama, Max Cov.) | 0.048 | 0.332 | 0.158 | 0.024 | 0.180 | 0.073 |
| Agent (Llama, Unif.) | 0.042 | 0.442 | 0.177 | 0.023 | 0.216 | 0.082 |

Table 23: Priv/Util Pareto Efficient Methods (Task 3: Privacy/utility tradeoff estimation) for PrivBayes.

# F Additional Detailed Results

In this section, we present additional detailed results of our evaluation framework (Section 4 and Appendix D) for the following DP auxiliary tasks: pretraining, hyperparameter tuning, and estimating the privacy-utility trade-off.

## F.1 Results for Task 1: Private Pretraining for Classification

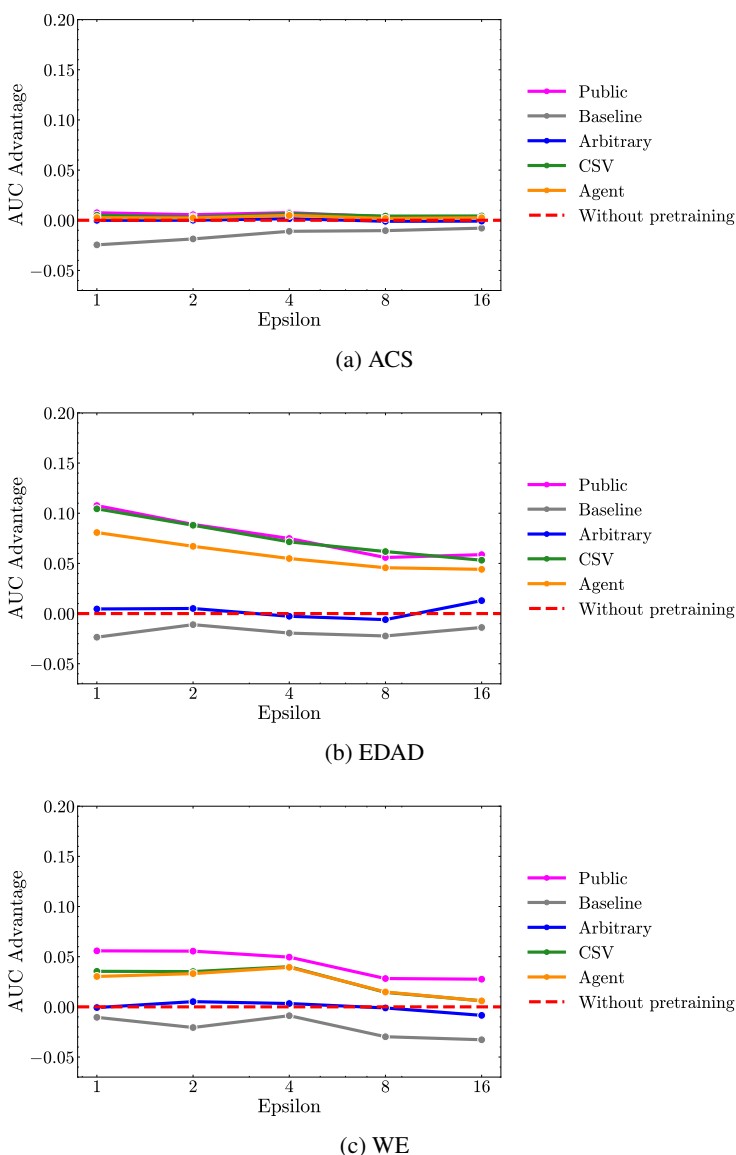

(a) ACS

(b) EDAD

(c) WE

Figure 15: Mean AUC Advantage of the DP model after pretraining, grouped by generation method. The mean is calculated across the hyperparameter space, with 10 runs per hyperparameter configuration.

Table 24: Mean **AUC Advantage** (AUC in parentheses) of the DP model after pretraining, grouped by generation method. The mean is calculated across the hyperparameter space, with 10 runs per hyperparameter configuration.

(a) ACS

| Method | $\varepsilon = 1$ | $\varepsilon = 2$ | $\varepsilon = 4$ | $\varepsilon = 8$ | $\varepsilon = 16$ |
|---|---|---|---|---|---|
| Without pretraining | .00 (.74) | .00 (.74) | .00 (.74) | .00 (.75) | .00 (.75) |
| Public | .01 (.75) | .01 (.75) | .01 (.75) | .00 (.75) | .00 (.75) |
| Baseline (Domain) | -.03 (.71) | -.02 (.72) | -.01 (.73) | -.01 (.74) | -.01 (.74) |
| Baseline (Univariate) | -.02 (.72) | -.02 (.73) | -.01 (.73) | -.01 (.74) | -.01 (.74) |
| Arbitrary | .00 (.74) | .00 (.74) | .00 (.74) | .00 (.75) | .00 (.75) |
| CSV (Claude 3.5 Sonnet) | .00 (.74) | .00 (.74) | .01 (.75) | .00 (.75) | .00 (.75) |
| CSV (GPT-4o) | .00 (.74) | .00 (.74) | .01 (.75) | .00 (.75) | .00 (.75) |
| CSV (Llama 3.3 70B) | .01 (.74) | .01 (.75) | .01 (.75) | .00 (.75) | .00 (.75) |
| Agent (Claude 3.5 Sonnet, Unif.) | .01 (.74) | .00 (.75) | .01 (.75) | .00 (.75) | .00 (.75) |
| Agent (Claude 3.5 Sonnet, Max Cov.) | .01 (.74) | .00 (.75) | .01 (.75) | .00 (.75) | .00 (.75) |
| Agent (GPT-4o, Unif.) | .00 (.74) | .00 (.74) | .01 (.75) | .00 (.75) | .00 (.75) |
| Agent (GPT-4o, Max Cov.) | .00 (.74) | .00 (.74) | .00 (.75) | .00 (.75) | .00 (.75) |
| Agent (Llama 3.3 70B, Unif.) | .00 (.74) | .00 (.74) | .00 (.75) | .00 (.75) | .00 (.75) |
| Agent (Llama 3.3 70B, Max Cov.) | .00 (.74) | .00 (.74) | .01 (.75) | .00 (.75) | .00 (.75) |
| Agent (Allm Unif.) | .01 (.74) | .01 (.75) | .01 (.75) | .00 (.75) | .00 (.75) |
| Agent (All, Max Cov.) | .00 (.74) | .00 (.75) | .01 (.75) | .00 (.75) | .00 (.75) |

(b) EDAD

| Method | $\varepsilon = 1$ | $\varepsilon = 2$ | $\varepsilon = 4$ | $\varepsilon = 8$ | $\varepsilon = 16$ |
|---|---|---|---|---|---|
| Without pretraining | .00 (.65) | .00 (.69) | .00 (.71) | .00 (.74) | .00 (.76) |
| Public | .11 (.76) | .09 (.78) | .07 (.79) | .06 (.80) | .06 (.82) |
| Baseline (Domain) | -.02 (.63) | -.01 (.67) | -.02 (.69) | -.02 (.73) | -.01 (.75) |
| Baseline (Univariate) | -.03 (.62) | -.01 (.68) | -.02 (.70) | -.03 (.71) | -.02 (.74) |
| Arbitrary | .01 (.66) | .01 (.69) | .00 (.71) | -.01 (.74) | .01 (.77) |
| CSV (Claude 3.5 Sonnet) | .11 (.76) | .09 (.78) | .08 (.79) | .07 (.81) | .06 (.82) |
| CSV (GPT-4o) | .09 (.74) | .08 (.77) | .06 (.78) | .06 (.80) | .05 (.81) |
| CSV (Llama 3.3 70B) | .11 (.76) | .09 (.78) | .08 (.79) | .07 (.81) | .05 (.81) |
| Agent (Claude 3.5 Sonnet, Unif.) | .08 (.73) | .07 (.76) | .06 (.77) | .05 (.80) | .04 (.81) |
| Agent (Claude 3.5 Sonnet, Max Cov.) | .09 (.74) | .07 (.76) | .06 (.78) | .04 (.79) | .05 (.81) |
| Agent (GPT-4o, Unif.) | .07 (.72) | .06 (.74) | .05 (.77) | .04 (.78) | .04 (.80) |
| Agent (GPT-4o, Max Cov.) | .07 (.72) | .06 (.75) | .04 (.76) | .04 (.78) | .04 (.80) |
| Agent (All, Unif.) | .08 (.73) | .07 (.75) | .05 (.77) | .05 (.79) | .04 (.81) |
| Agent (All, Max Cov.) | .09 (.74) | .07 (.76) | .06 (.78) | .05 (.79) | .05 (.81) |

(c) WE

| Method | $\varepsilon = 1$ | $\varepsilon = 2$ | $\varepsilon = 4$ | $\varepsilon = 8$ | $\varepsilon = 16$ |
|---|---|---|---|---|---|
| Without pretraining | .00 (.53) | .00 (.55) | .00 (.58) | .00 (.63) | .00 (.66) |
| Public | .06 (.59) | .06 (.61) | .05 (.63) | .03 (.66) | .03 (.69) |
| Baseline (Domain) | -.02 (.51) | -.03 (.52) | -.02 (.56) | -.04 (.59) | -.04 (.62) |
| Baseline (Univariate) | .00 (.53) | -.01 (.55) | .01 (.58) | -.02 (.61) | -.03 (.63) |
| Arbitrary | .00 (.53) | .01 (.56) | .00 (.58) | .00 (.63) | -.01 (.65) |
| CSV (Claude 3.5 Sonnet) | .06 (.59) | .06 (.61) | .06 (.64) | .03 (.66) | .02 (.68) |
| CSV (GPT-4o) | .04 (.58) | .05 (.60) | .04 (.62) | .03 (.66) | .02 (.67) |
| CSV (Llama 3.3 70B) | .00 (.53) | .00 (.56) | .02 (.59) | -.01 (.62) | -.02 (.64) |
| Agent (Claude 3.5 Sonnet, Unif.) | .10 (.64) | .10 (.65) | .10 (.68) | .06 (.69) | .05 (.71) |
| Agent (Claude 3.5 Sonnet, Max Cov.) | .11 (.64) | .11 (.66) | .09 (.67) | .07 (.70) | .05 (.71) |
| Agent (GPT-4o, Unif.) | -.01 (.53) | .00 (.55) | .01 (.59) | -.01 (.62) | -.02 (.64) |
| Agent (GPT-4o, Max Cov.) | -.04 (.49) | -.03 (.52) | -.03 (.55) | -.03 (.60) | -.03 (.62) |
| Agent (Llama 3.3 70B, Unif.) | .00 (.53) | .01 (.56) | .02 (.60) | .00 (.63) | -.01 (.65) |
| Agent (Llama 3.3 70B, Max Cov.) | -.01 (.52) | -.01 (.55) | .01 (.59) | -.01 (.62) | -.02 (.64) |
| Agent (All, Unif.) | .06 (.59) | .06 (.61) | .06 (.64) | .03 (.66) | .02 (.68) |
| Agent (All, Max Cov.) | .03 (.56) | .03 (.59) | .05 (.63) | .01 (.64) | .01 (.67) |

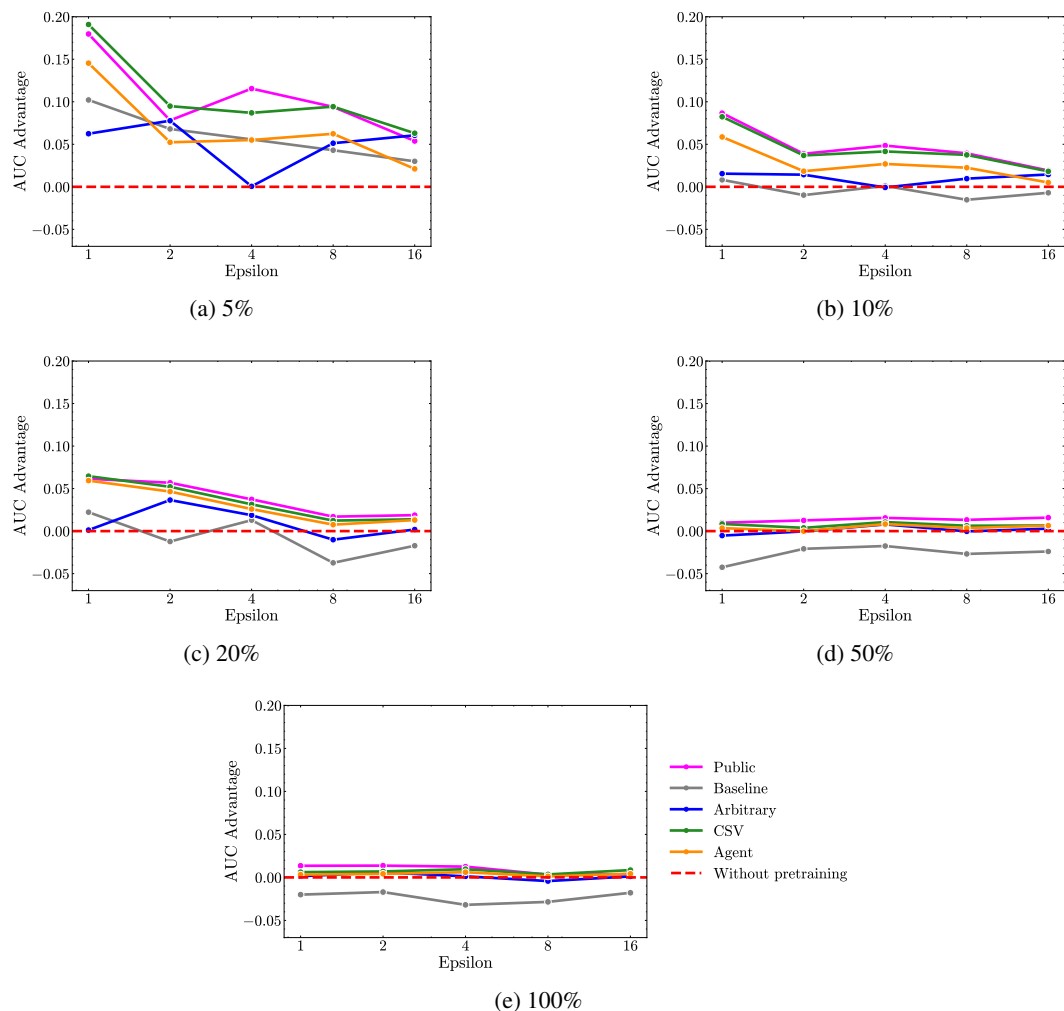

Figure 16: Mean AUC Advantage of the DP model after pretraining, grouped by generation method for the sub-sampled ACS dataset. The mean is calculated across the DP finetuning hyperparameter space when best pretraining hyperparameter configuration is chosen for the pretraining step, with 10 runs per hyperparameter configuration.

**F.2    Results for Task 2: Hyperparameter Tuning for Private Synthetic Data**

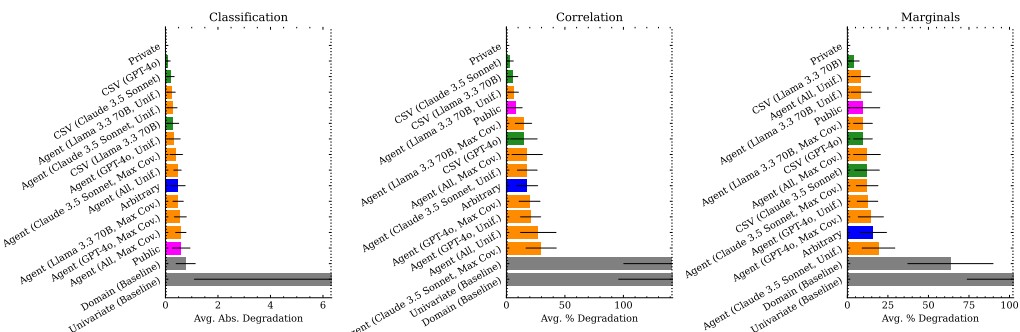

Figure 17: Granular hyperparameter tuning results for ACS on PrivBayes. Note the poor relative performances of the `Baselines` relative to the other methods; encoding relationships between variables is clearly very important to tuning hyperparameters on the PrivBayes Classifier.

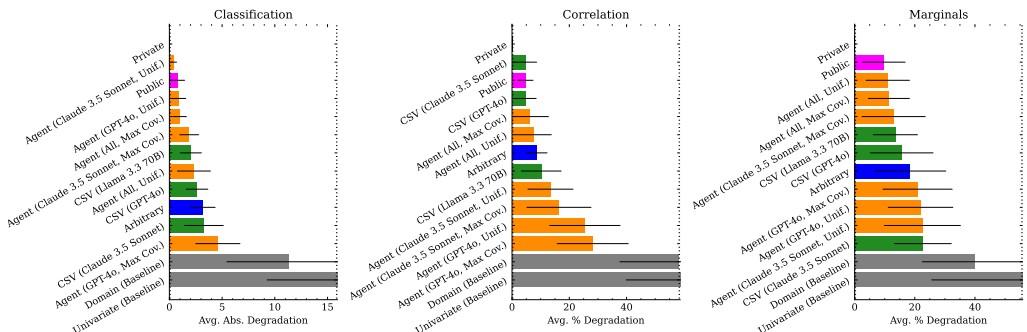

Figure 18: Granular hyperparameter tuning results for EDAD on PrivBayes. Note that the agent-based method `Agent (Claude, Unif.)` leads in classification (0.004) while `CSV (Claude)` dominates the correlation metric (0.046); meanwhile, real public data yields the best marginal consistency (0.097).

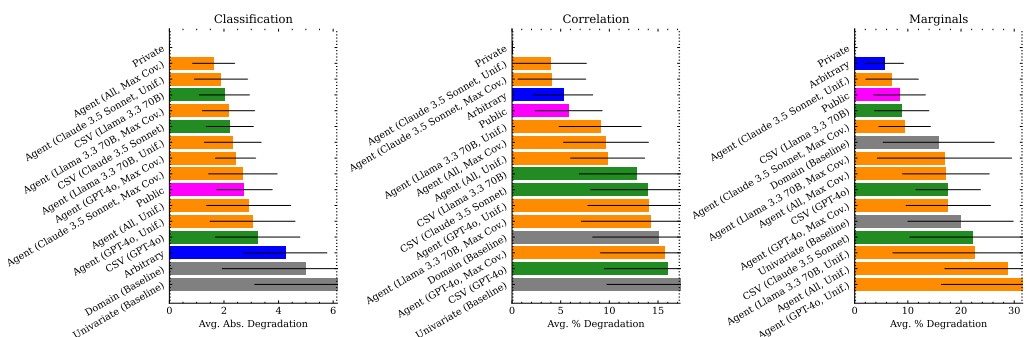

Figure 19: Granular hyperparameter tuning results for WE on PrivBayes. Observe that although the `Arbitrary` baseline excels in marginal consistency (0.056) and is competitive on correlation (0.052), `Agent`based approaches (e.g., `All, Max Cov.` and `Claude, Unif.`) offer improvement in classification performance (0.0160.019).

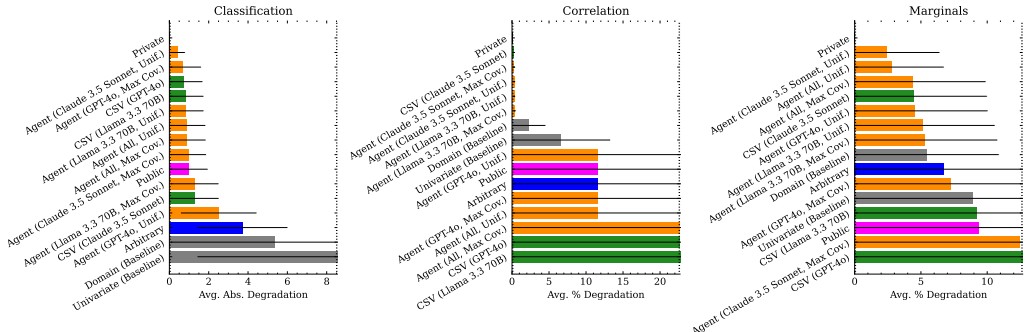

Figure 20: Granular hyperparameter tuning results for ACS on the AIM synthesizer. Here, `Agent (Claude, Unif.)` outperforms on both classification (0.004) and marginal consistency (0.024), while `CSV (Claude)` is best on correlation (0.002).

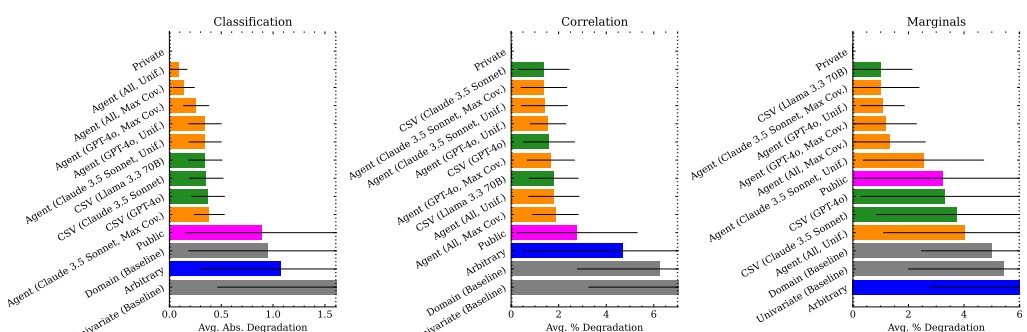

Figure 21: Granular hyperparameter tuning results for EDAD on the AIM synthesizer. Several agent-based methods, such as `Agent (All, Unif.)`, deliver strong classification performance (0.001), with the Pareto frontier defined by a mix of CSV and agentbased approaches (correlation metrics ranging from 0.014 to 0.019).

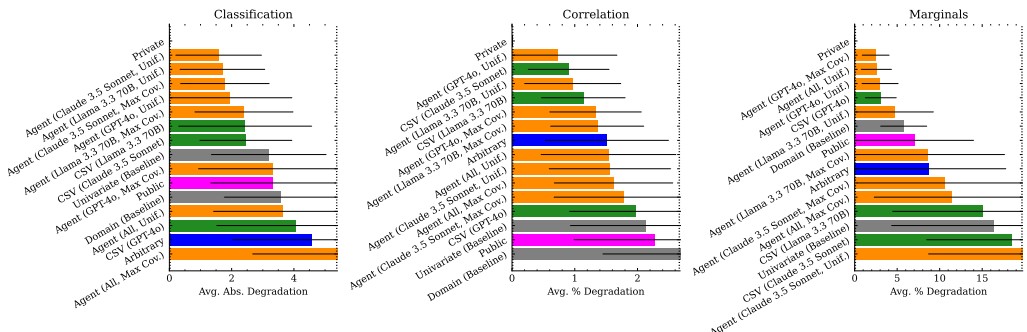

Figure 22: Granular hyperparameter tuning results for WE on the AIM synthesizer. Exclusively agentbased methods dominate, with `Agent (Claude, Unif.)` leading in classification (0.016), `Agent (GPT, Unif.)` achieving the best correlation (0.007), and `Agent (GPT, Max Cov.)` strong marginal consistency (0.025).

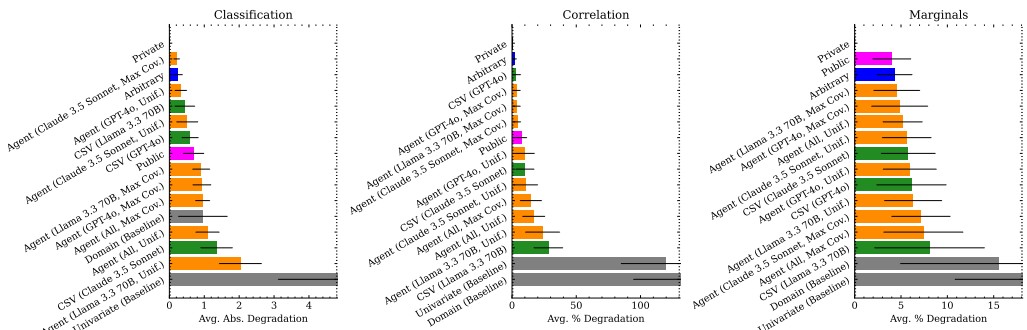

Figure 23: Granular hyperparameter tuning results for ACS on the GEM synthesizer. The `Agent` (`Claude, Max Cov.`) method, alongside the `Arbitrary` baseline that directly encodes variable relationships, is dominant – reinforcing that structure in the data is beneficial.

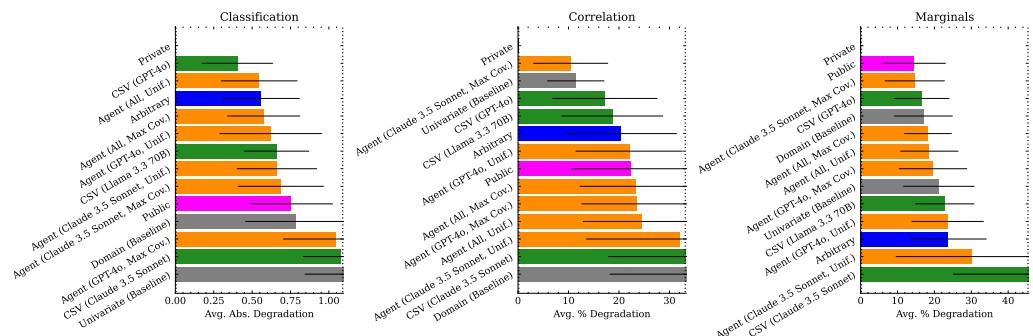

Figure 24: Granular hyperparameter tuning results for EDAD on the GEM synthesizer. As in ACS, both the agentbased approach and the `Arbitrary` baseline perform competitively.

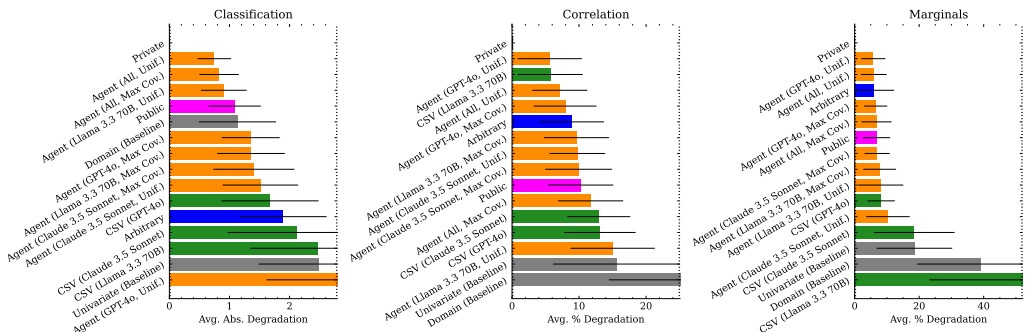

Figure 25: Granular hyperparameter tuning results for WE on the GEM synthesizer. The trends mirror those in ACS, with the `Arbitrary` baseline maintaining strong performance and `Agent`based methods showing similar competitiveness.

## F.3  Results for Task 3: Estimating the Privacy/Utility Tradeoff

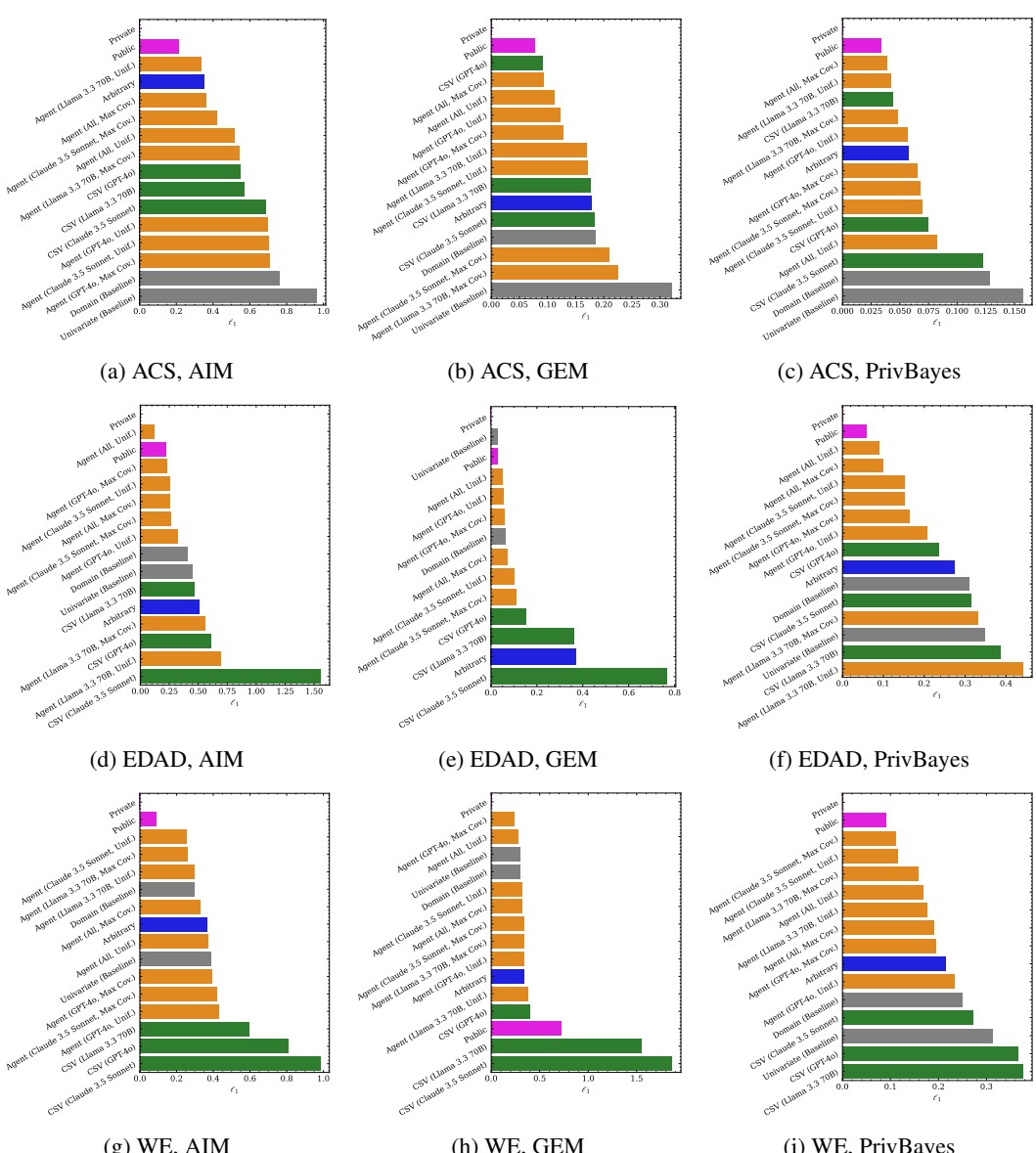

Figure 26: Privacy/utility tradeoff estimation results in terms of $\ell_1$ distance from the true sensitive data tradeoff. Note the relatively consistent performance across synthesizers for each dataset between some methods (e.g., poor privacy/utility tradeoff estimation for CSV on WE), while other methods have higher variance (e.g., Agent (Claude 3.5 Sonnet, Max Cov. on ACS, between GEM and AIM).

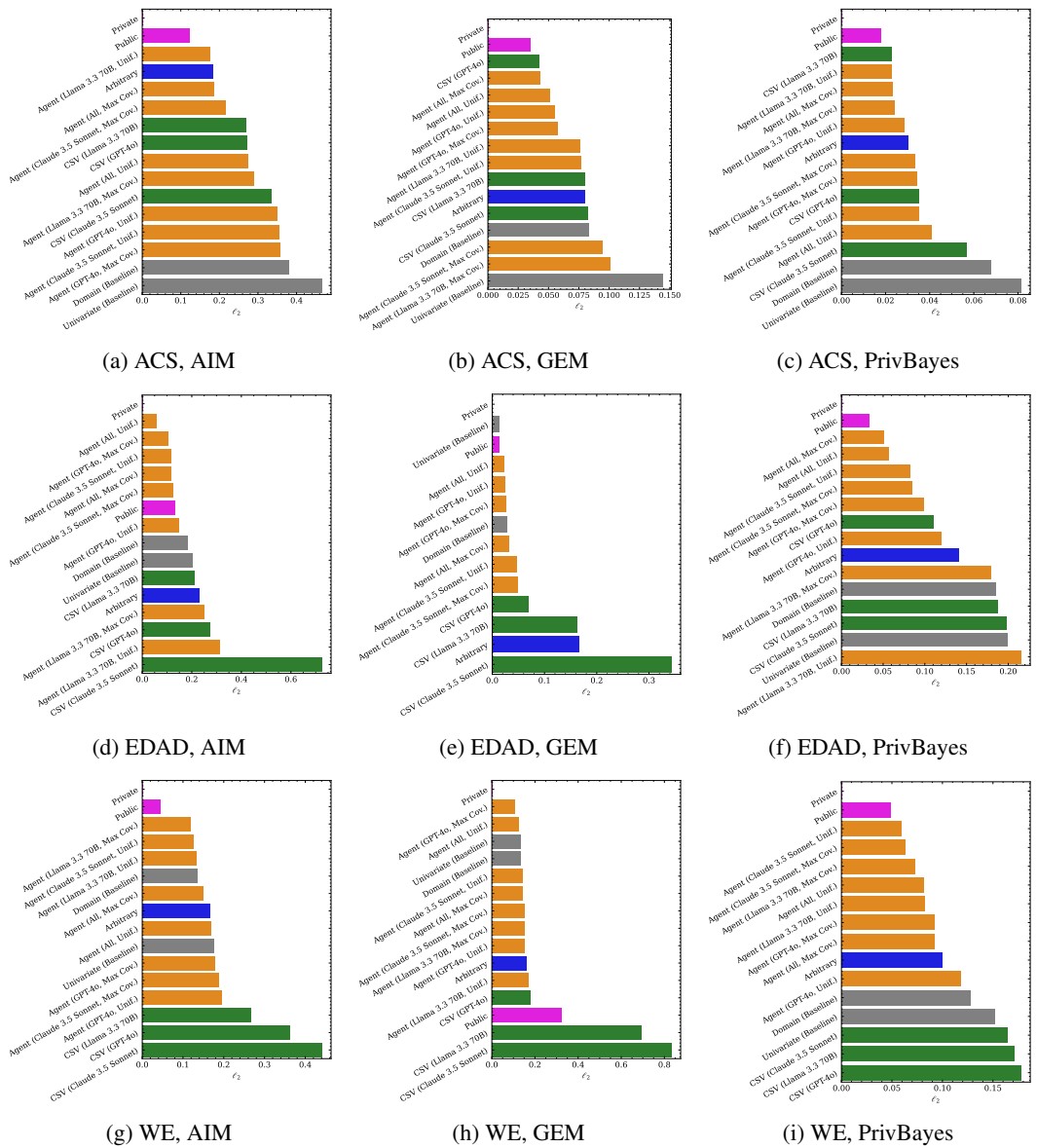

Figure 27: Privacy/utility tradeoff estimation results in terms of $\ell_2$ distance from the true sensitive data tradeoff. These results largely mirror the $\ell_1$ distance results, although the increased sensitivity to outliers leads to some interchanges of ranking (e.g., `Agent (Claude 3.5 Sonnet, Unif.)` and `CSV (GPT-4o)` interchange places on ACS PrivBayes).

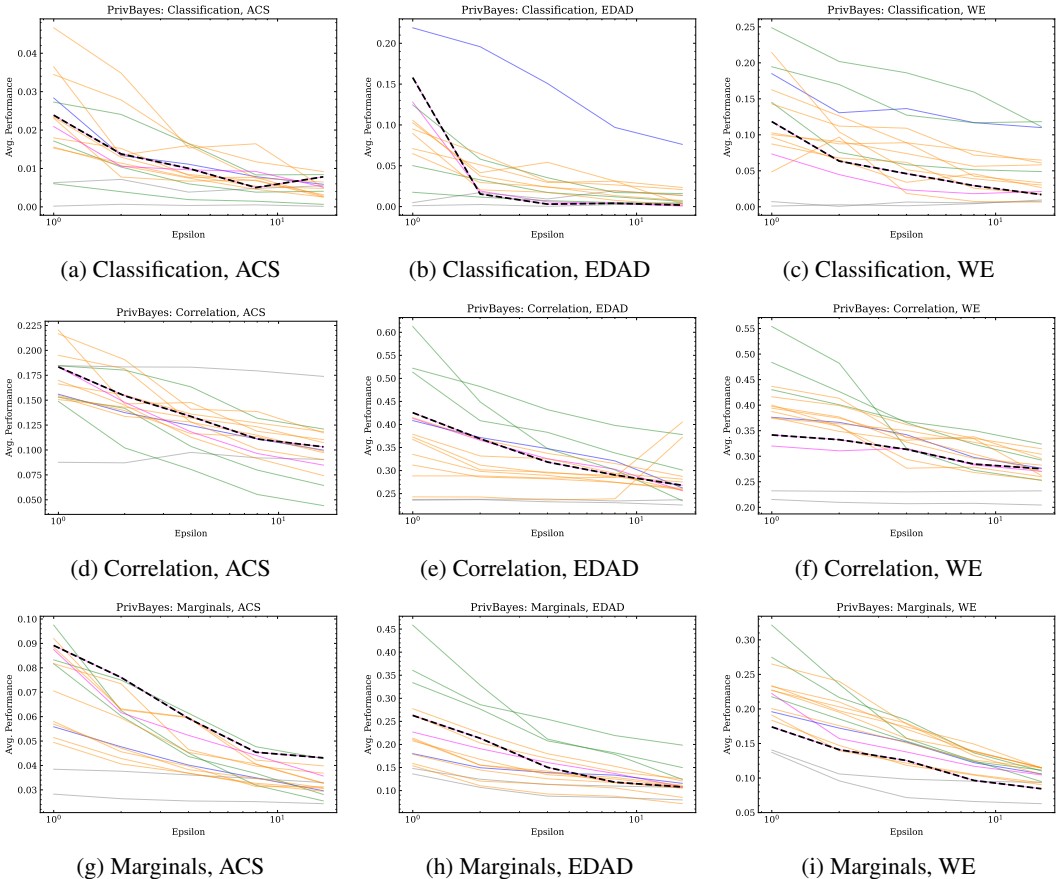

(a) Classification, ACS      (b) Classification, EDAD      (c) Classification, WE

(d) Correlation, ACS      (e) Correlation, EDAD      (f) Correlation, WE

(g) Marginals, ACS      (h) Marginals, EDAD      (i) Marginals, WE

Figure 28: To provide intuition for *exactly* what the $\ell_1$ and $\ell_2$ scores in Figures 26 and 27 attempt to capture, we plot the average performance across epsilon that constitutes each vector, relative to the true performance of the sensitive data (which, in these plots, is the black dotted line). Ideally, for privacy/utility estimation, any public data (surrogate or otherwise) would *match* the performance of the private data across privacy loss budget parameters. This would allow a practitioner to, say, choose the correct $\epsilon$ based on a performance threshold in absolute terms. Clearly, given the noisiness of the lines (which generally cluster around, but inconsistently track, the black dotted line for private data performance), this is a difficult estimation problem.

Table 25: Dataset similarity assessment against the private data for ACS, EDAD and WE. The datasets are evaluated based on two distance metrics (Section G.1): (1) Total Variation Distance (TVD); and (2) Average error on 3-Way Marginals (3WM). Both metrics are in range $[0, 1]$, inverted to represent similarity $(1 - x)$, and scaled by 100. Zero values (rounded) are omitted for readability.

| Method | ACS 1-TVD | ACS 1-3WM | EDAD 1-TVD | EDAD 1-3WM | WE 1-TVD | WE 1-3WM |
|---|---|---|---|---|---|---|
| Public | 48.5 | 50.4 | 4.9 | 26.1 | 6.7 | 34.1 |
| Baseline (Domain) | 4.3 | | 0.1 | | 0.2 | |
| Baseline (Univariate) | 44.6 | 63.8 | 7.1 | 66.7 | 15.4 | 78.5 |
| Arbitrary | 2.8 | | 0.1 | | | |
| CSV (Claude 3.5 Sonnet) | 14.4 | 15.0 | | | | 10.9 |
| CSV (GPT-4o) | 25.7 | 30.2 | | 11.5 | | 14.2 |
| CSV (Llama 3.3 70B) | 16.6 | 10.0 | | | | 2.4 |
| Agent (Claude 3.5 Sonnet, Unif.) | 41.5 | 48.3 | | 5.5 | | 11.7 |
| Agent (Claude 3.5 Sonnet, Max Cov.) | 40.1 | 40.0 | | 6.8 | | 8.0 |
| Agent (GPT-4o, Unif.) | 27.3 | 23.3 | | 7.2 | | |
| Agent (GPT-4o, Max Cov.) | 27.4 | 20.4 | | 6.9 | | |
| Agent (Llama 3.3 70B, Unif.) | 13.8 | | | | | |
| Agent (Llama 3.3 70B, Max Cov.) | 10.3 | | | | | |
| Agent (All, Unif.) | 30.5 | 26.6 | | | | |
| Agent (All, Max Cov.) | 24.6 | 15.7 | | | | |

# G   Details of Dataset Similarity

## G.1   Statistical Distance Metrics

We now introduce metrics for comparing probability distributions and datasets used throughout this paper.

**Definition 3** (Total Variation Distance). *For discrete probability distributions $P$ and $Q$ over $\mathcal{X}$, the Total Variation Distance (TVD) is defined as:*

$$TVD(P, Q) = \frac{1}{2} \sum_{x \in \mathcal{X}} |P(x) - Q(x)|$$

The Total Variation Similarity (TVS) is simply $1 - \text{TVD}(P, Q)$, representing the similarity rather than the distance between distributions. Both TVD and TVS can be naturally extended to datasets by considering the empirical probability distributions induced by the datasets over the universe $\mathcal{X}$.

Now we turn to a more specific measurement of disparity between two datasets based on the results of statistical queries.

**Definition 4** (Linear Query). *Given a predicate $\phi : \mathcal{X} \to \{0, 1\}$ that maps database records to binary values, a linear query $q_\phi : \mathcal{X}^n \to \mathbb{N}_0^+$ is a function that, for a dataset $D \in \mathcal{X}^n$, computes:*

$$q_\phi(D) = \sum_{r \in D} \phi(r)$$

*In other words, a linear query counts the number of records in dataset $D$ that satisfy the predicate $\phi$.*

**Definition 5** (Workload Error). *Given a workload $W = \{q_1, \ldots, q_k\}$ of linear queries, and a pair of datasets $D, D' \in \mathcal{X}^n$, the workload error is defined as:*

$$WError(D, D') = \sum_{q \in W} |q_i(D) - q_i(D')|$$

The *average $k$-way marginal error* can be defined as a special case of the workload error where the workload $W$ consists of all possible $k$-way marginal queries. For instance, the 3-way marginal error

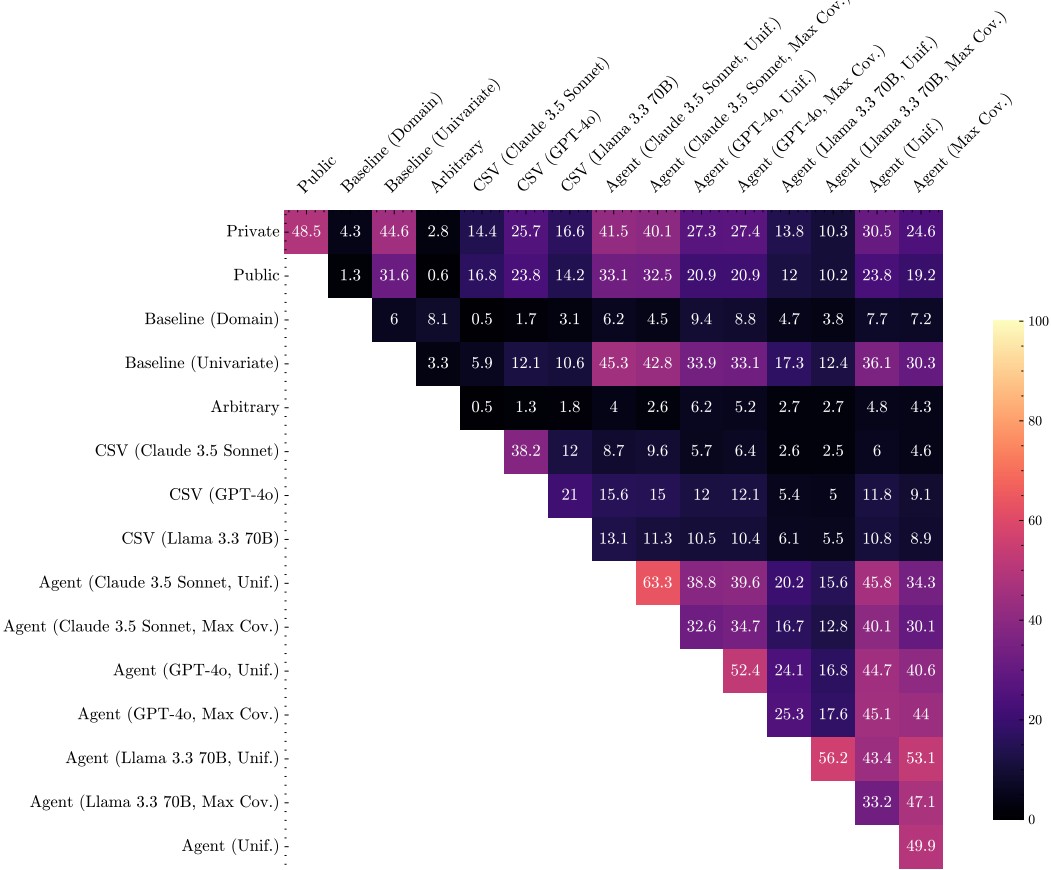

Figure 29: Heatmap of similarity metrics based on the Total Variation Distance (TVD) between the datasets based on the ACS data. The metric is in range $[0, 1]$, inverted to represent similarity $(1 - x)$, and scaled by 100, and rounded to a single digit.

uses all possible triplet combinations of attributes as queries. Assuming datasets of equal size, the average $k$-way marginal error is normalized by both the number of queries in the workload $|W|$ and the size of the datasets $|D|$:

$$\text{AvgError}_{k\text{-way}}(D, D') = \frac{1}{|W| \cdot |D|} \sum_{q \in W} |q(D) - q(D')|$$

where $W$ is the set of all $k$-way marginal queries, and $|W| = \binom{d}{k}$ for a dataset with $d$ attributes.

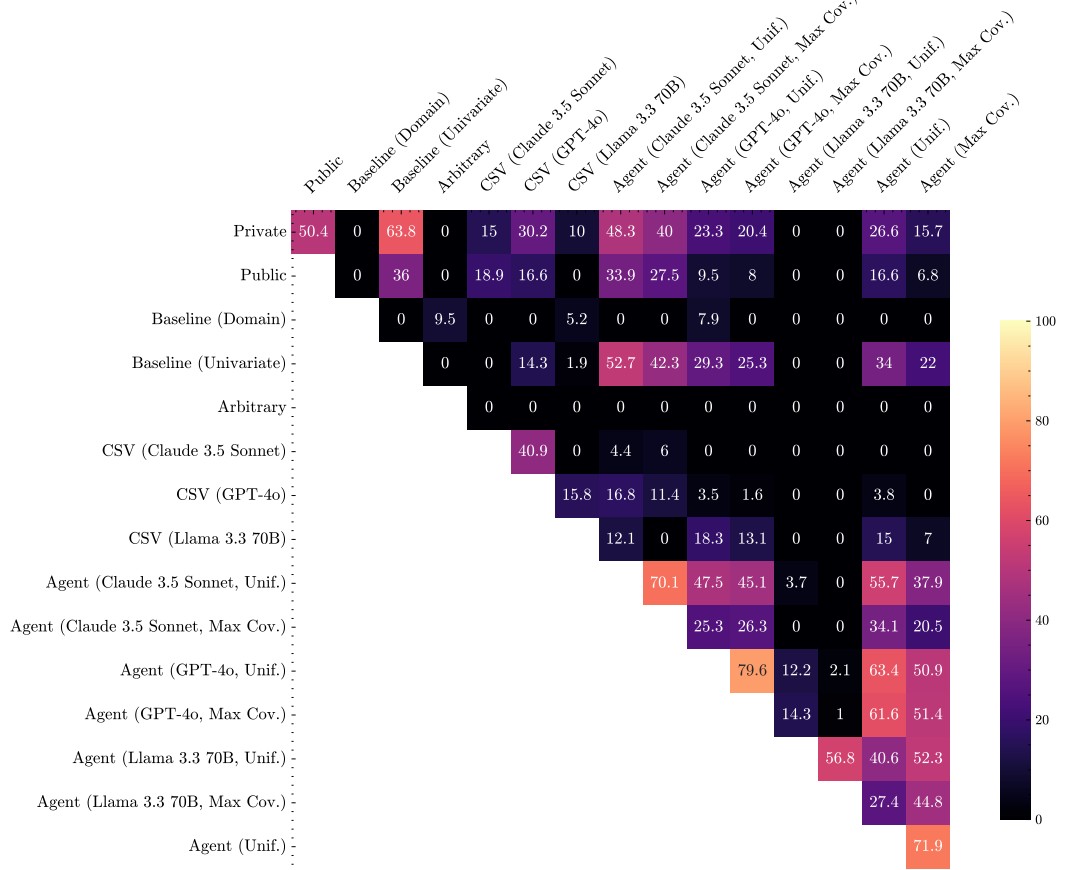

Figure 30: Heatmap of similarity metrics based on the Average Error on 3-Way Marginals (3WM) between the datasets based on the ACS data. The metric is in range $[0, 1]$, inverted to represent similarity $(1 - x)$, and scaled by 100, and rounded to a single digit.

Table 26: We also considered the following metrics when comparing the surrogate public datasets (beyond TVD and three-way marginals): distance-to-closest-record (DCR), nearest-neighbor distance ratio (NNDR), and classifier-based $\alpha$-precision/$\beta$-recall. Our light experimentation here corroborates the main papers conclusions. We include DCR and NNDR (5th percentile) from [19], and $\alpha$-precision/$\beta$-recall from [5, 90].

| Method | exact_match_prop | dcr_5th_pct | nndr_5th_pct | $\alpha$-precision | $\beta$-recall |
|---|---|---|---|---|---|
| Private | 100.0 | 100.0 | 100.0 | 100.0 | 100.0 |
| Public | 6.9 | 100.0 | 100.0 | 78.1 | 50.8 |
| Domain | 0.1 | 90.9 | 50.0 | 40.9 | 31.0 |
| Arb. | 0.0 | 81.8 | 33.3 | 23.5 | 12.9 |
| CSV (C3.5) | 1.0 | 90.9 | 33.3 | 70.9 | 13.5 |
| CSV (G4o) | 1.5 | 90.9 | 50.0 | 80.9 | 30.0 |
| CSV (L3.3) | 0.0 | 81.8 | 33.3 | 85.2 | 3.5 |
| Agent (C3.5,U) | 0.7 | 90.9 | 50.0 | 70.4 | 42.8 |
| Agent (C3.5,M) | 1.0 | 90.9 | 50.0 | 68.2 | 40.0 |
| Agent (G4o,U) | 0.2 | 90.9 | 50.0 | 88.2 | 44.2 |
| Agent (G4o,M) | 0.7 | 90.9 | 50.0 | 88.9 | 43.6 |
| Agent (L3.3,U) | 0.2 | 81.8 | 33.3 | 2.2 | 8.1 |
| Agent (L3.3,M) | 0.2 | 81.8 | 33.3 | 2.1 | 8.8 |
| Agent (U) | 0.5 | 90.9 | 50.0 | 67.4 | 37.6 |
| Agent (M) | 0.2 | 90.9 | 50.0 | 69.6 | 36.3 |

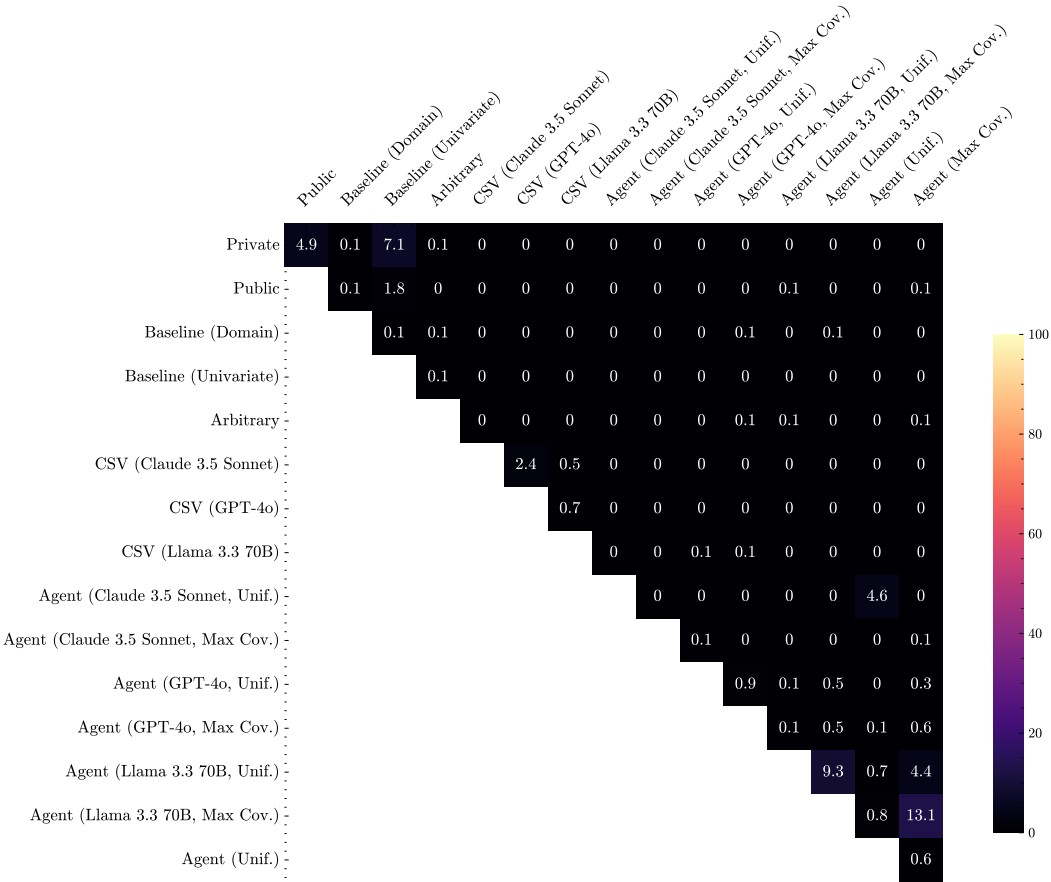

Figure 31: Heatmap of similarity metrics based on the Total Variation Distance (TVD) between the datasets based on the EDAD data. The metric is in range $[0, 1]$, inverted to represent similarity $(1 - x)$, and scaled by 100, and rounded to a single digit.

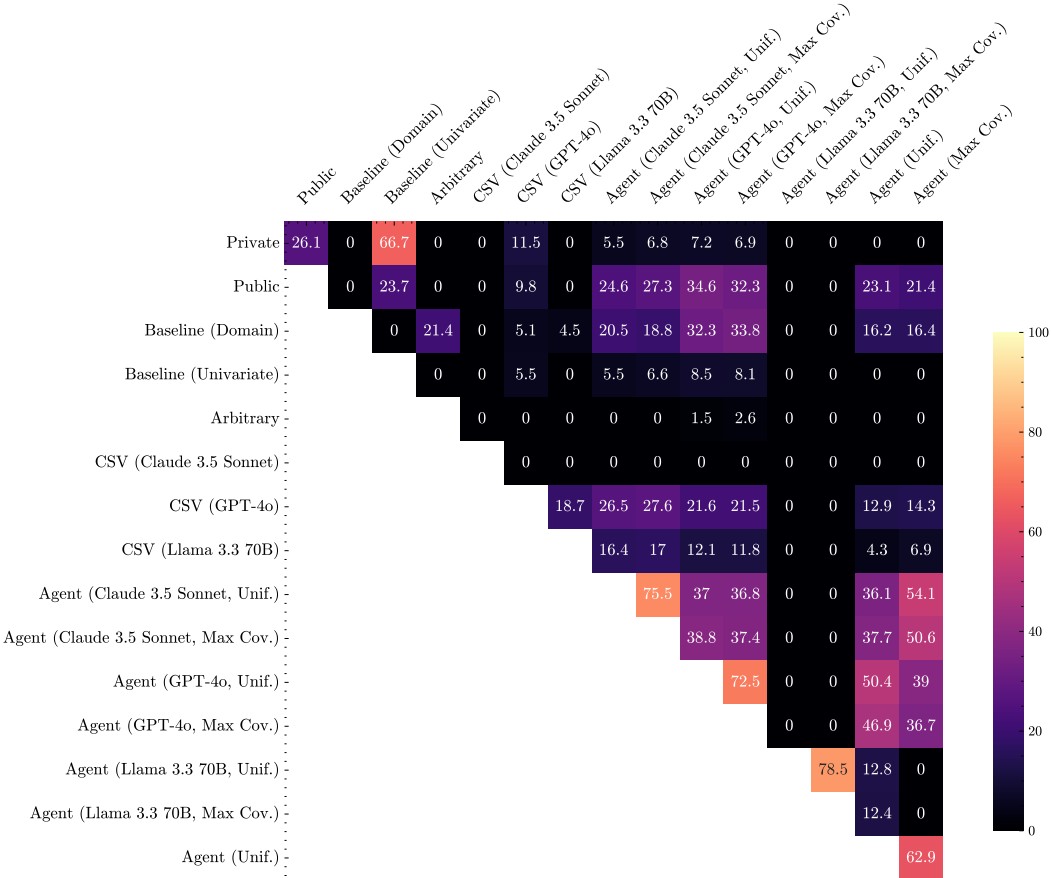

Figure 32: Heatmap of similarity metrics based on the Average Error on 3-Way Marginals (3WM) between the datasets based on the EDAD data. The metric is in range $[0, 1]$, inverted to represent similarity $(1 - x)$, and scaled by 100, and rounded to a single digit.

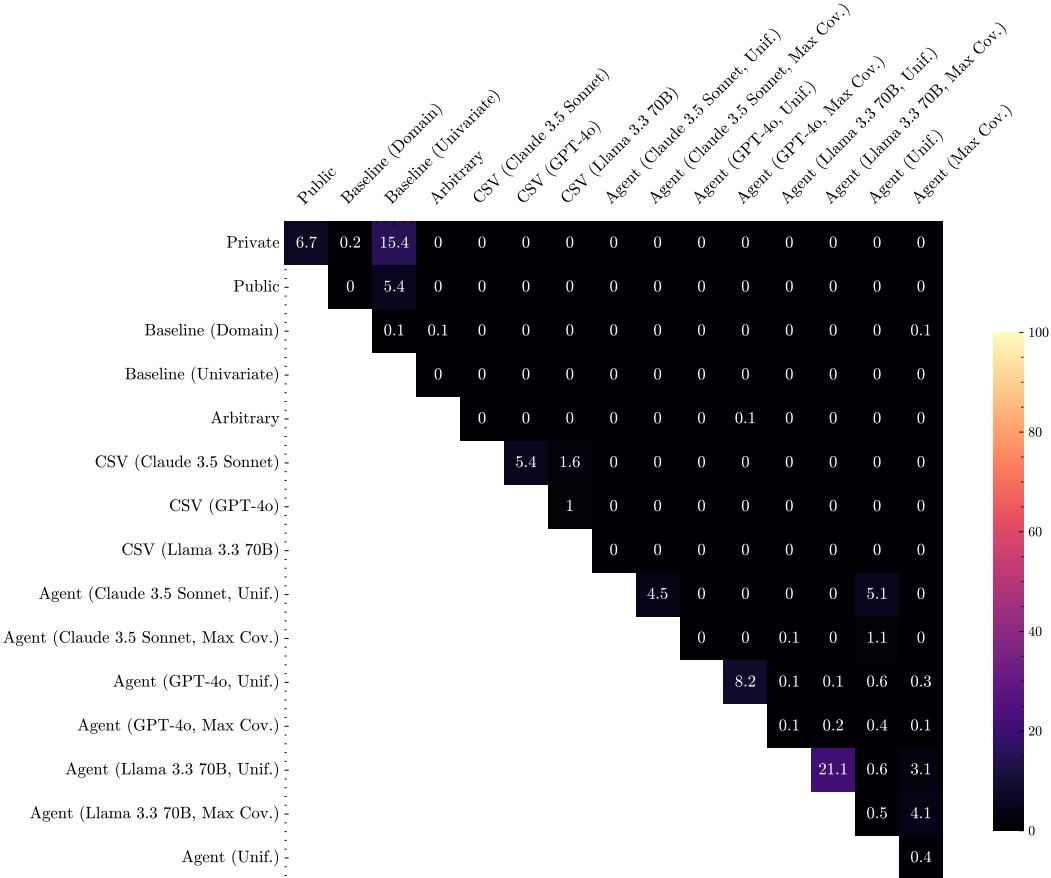

Figure 33: Heatmap of similarity metrics based on the Total Variation Distance (TVD) between the datasets based on the WE data. The metric is in range $[0, 1]$, inverted to represent similarity $(1 - x)$, and scaled by 100, and rounded to a single digit.

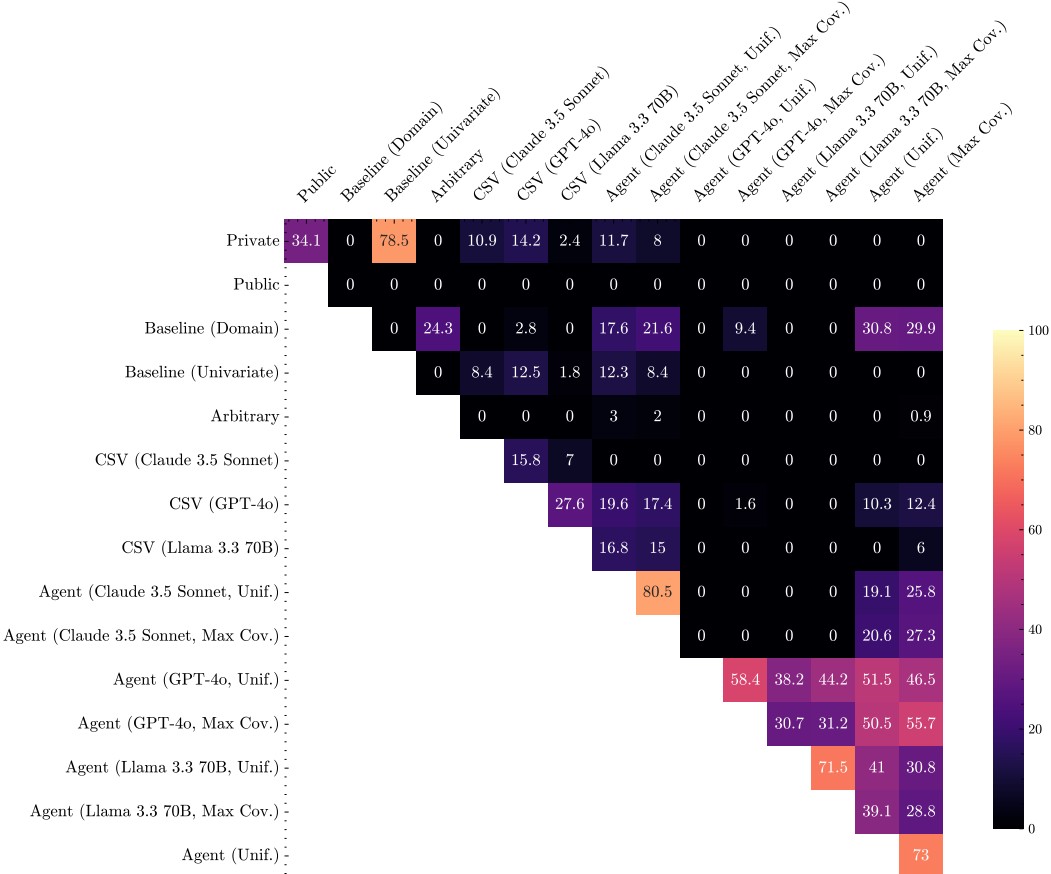

Figure 34: Heatmap of similarity metrics based on the Average Error on 3-Way Marginals (3WM) between the datasets based on the WE data. The metric is in range $[0, 1]$, inverted to represent similarity $(1 - x)$, and scaled by 100, and rounded to a single digit.

## H   Compute and Resources

Benchmarking DP synthesizers and training models for differentially private tasks is computationally intensive [93]. We executed our experiments on a combination of high-performance GPU and CPU clusters hosted on AWS EC2. Specifically, we utilized three `g4dn.12xlarge` instances – each equipped with NVIDIA T4 GPUs – for approximately 17.3 days of continuous up-time per instance, amounting to roughly 52 GPU-days in total (although it is hard to assess the true GPU utilization). In addition to local compute, we used LLM APIs provided by OpenAI, Anthropic, and TogetherAI (for the Llama 3 model) for both our direct CSV generation and multi-step Agent-based approaches. We conducted substantial inference for our experiments; as an example, during January, our queries to Claude alone amounted to a total of 38,092,225 input tokens and produced 7,099,403 output tokens, in February, we recorded 11,922,046 input tokens and 226,998 output tokens, and in March, 9,027,827 input tokens and 124,484 output tokens were consumed (imbalance between input output due to re-inputting previously generated tokens as context on each call in the state machine for the Agent). These resources allowed for extensive hyperparameter searches, multiple runs per privacy setting, and a comprehensive evaluation across DP auxiliary tasks.

