# OpenReview forum: "Do You Really Need Public Data? Surrogate Public Data for Differential Privacy on Tabular Data"
_NeurIPS.cc/2025/Datasets_and_Benchmarks_Track — NeurIPS 2025 Datasets and Benchmarks Track poster_

### Official Review · Reviewer_rgrp · 2025-06-02

**Rating:** 4
**Confidence:** 2

**Summary:**

The paper aims to answer the research question: *“Can surrogate public data generated from schema metadata using large language models (LLMs) effectively replace traditional public data for auxiliary tasks in differentially private (DP) learning on tabular data?”*.

The main contributions of the paper are two-fold:

1. **Methodological Contribution**: Proposes two methods for LLM-driven data generation:
    - **CSV Generation**: Direct generation of synthetic records in CSV format adhering to schema constraints.
    - **Agent Approach**: Constructs structural causal models (SCMs) based on schema metadata and samples from these models.

2. **Empirical Contribution**: Demonstrates through experiments that LLM-generated data improves DP classifier pretraining and performs competitively for hyperparameter tuning and privacy-utility estimation.

Note: my score of 4 should be interpreted as *"If there is a clear consensus among other reviewers that the paper should be accepted, then I won't be radically against that"*

**Additional Feedback:**

## Questions

- Q1: How to ensure there is no/minimal data leakage when generating samples with LLM?

- Q2: How to evaluate the SCM generated by LLM?

- Q3: How can the evaluation results guarantee that the surrogate public dataset is better in DP than the public dataset?

**Dataset Code Accessibility:**

Yes

**Dataset Code Comments:**

Yes, the provided datasets are accessible.

**Ethical Considerations:**

No, there are no or only very minor ethics concerns

**Final Justification:**

Throughout the rebuttal and discussion phases, I have carefully considered both the authors’ clarifications and the feedback from the other reviewers. The paper presents a conceptually meaningful contribution for privacy-preserving data synthesis and sharing. That said, if the other reviewers are satisfied with the current version of the paper, I would not strongly oppose acceptance.

**Limitations Weaknesses:**

**1. [Important] Unclear motivation:** As the authors point out in Section 1, tabular data is different due to its heterogeneity. However, it remains unclear to me why this would highlight the necessity of surrogate public data. For lines 38-41, my confusion lies primarily around the fact that it is hard for any modality to find truly public, non-sensitive samples. I am unsure if “heterogeneity” is the root of the claimed unique challenges of tabular data.

**2. [Important] Insufficient evaluation on privacy preservation:** The paper seems to provide only $\epsilon$ as the indicator for privacy preservation performance. However, it could be insufficient to prove that the surrogate public data is truly better at avoiding privacy leakage. Please refer to prior studies [1, 2, 3], where many other metrics are considered simultaneously to avoid potential bias or imprecision of a specific metric.

**3. Insufficient evaluation on synthetic data quality:** This is less important than the above points, as I do acknowledge privacy is the major concern of the paper. However, it would also be interesting to provide basic evaluation results like statistical similarity of the synthetic data compared to public data and private data. Although some low-order metrics are considered (i.e., TVD and 3WM), they may be trivial, and I would suggest that the authors check out more recent high-order evaluation metrics for tabular data [4, 5].

[1] Du, Yuntao, and Ninghui Li. "Towards principled assessment of tabular data synthesis algorithms." arXiv e-prints (2024): arXiv-2402.

[2] Kotelnikov, Akim, et al. "Tabddpm: Modelling tabular data with diffusion models." International Conference on Machine Learning. PMLR, 2023.

[3] Borisov, Vadim, et al. "Language Models are Realistic Tabular Data Generators." The Eleventh International Conference on Learning Representations.

[4] Zhang, Hengrui, et al. "Mixed-Type Tabular Data Synthesis with Score-based Diffusion in Latent Space." The Twelfth International Conference on Learning Representations.

[5] Shi, Juntong, et al. "TabDiff: a Mixed-type Diffusion Model for Tabular Data Generation." The Thirteenth International Conference on Learning Representations.

**Strengths Contributions:**

**1. Extensive coverage of DP auxiliary tasks:** The paper designs and implements an extensible benchmark across three major DP auxiliary tasks: model pretraining, hyperparameter tuning, and privacy-utility estimation

**2. Clear writing:** The paper is generally well-written and clear, especially the structure of the experiment section.

**3. Good reproducibility:** The paper provides a codebase for the reported results, and the code seems clear and accompanied by tutorial notebooks.

---

> ### Author Rebuttal · Authors · 2025-07-31
>
> Thank you so much for taking the time to review our paper and offer constructive feedback; we are sure that it will strengthen our paper. We further respond to your comments and concerns below, inline.
>
> > ***“1. Extensive coverage of DP auxiliary tasks …  2. Clear writing …  3. Good reproducibility …”***
>
> We are grateful that the reviewer valued (i) the breadth of the benchmark (ii) the effort we invested in making the paper readable, and (iii) the open‑sourced code. Thank you!
>
> > **W1: *“It remains unclear … whether heterogeneity is really the root challenge that necessitates surrogate public data.”***
>
> Thank you for prompting us to clarify this point, we hope the following will be useful in your understanding, and we will add a version of this as a minor revision in our paper.
>
> In vision or text, a single, fixed coordinate system (RGB grids; token indices) and a handful of massive “foundation” corpora provide universal priors: ImageNet, COCO, Common Crawl, etc. For example, a neural classifier pre‑trained on cats vs. dogs could readily be fine‑tuned for x‑ray pathology because the input dimensionality, structure and “semantics of image” are unchanged [Tang et al. 2023, arXiv 2306.06076]. In essence, the weights have been taught what an image “looks like” before they’ve been finetuned on a specific target distribution.
>
> We argue that tabular data is different in three ways that together motivate the surrogate public data that we generate:
>
> (1) **Schema multiplicity,** which is to say every survey, medical EHR, or transaction log defines in some sense a “bespoke” set of columns. Adding or deleting a single column can literally change both the input dimension *and* the semantic meaning of the data; for example, categorical codes (e.g., ICD‑10 vs. SNOMED) are rarely compatible across institutions.
>
> (2) **Cross‑domain priors are weak,** which is to say it's not clear that the types of meta-features like “edges” (in the image domain) or “language syntax” (in the text domain) necessarily exist in the tabular domain. Put another way, there are likely no stable high‑level features shared by, say, a hospital read‑admission tabular dataset and a credit‑risk tabular dataset (outside of basic demographic information).
>
> (3) **Public tabular samples are sparse,** because many sensitive domain tabular datasets are small and subject to legal restrictions, so close public proxies frequently don’t exist. In the image domain, for example, we can blur, down-sample, recolor, generate synthetic examples easily, etc. For tables we often cannot “morph” one schema into another without expert domain knowledge.
>
> LLM‑driven surrogate public data gives us a way to instantiate a plausible in‑domain tabular data sample from only the schema (thus not incurring privacy costs) using the strong semantic prior the LLM has learned, which is a practical approach that side‑steps the absence of naturally occurring public tabular data that’s in-distribution. We appreciate this push to clarify explicitly this motivation, and would be happy to add a concise example comparing an image task with a survey task (like adding a new Likert-scale question, which increases the dimension size and potentially breaks input compatibility) to make this distinction explicit in the introduction.
>
> > **W2: *“Only $\varepsilon$‑DP is reported; other privacy‑leakage metrics should be considered.”***
>
> We highlight that our goal is to avoid consuming any privacy budget during auxiliary tasks. From that standpoint a surrogate is either (i) generated independently of private data (zero leakage by construction) or (ii) not a public data surrogate. Formal ($\varepsilon$, $\delta$)‑DP therefore remains the gold standard: if the downstream mechanism is $\varepsilon$‑DP regardless of the surrogate, then no attack on the surrogate can worsen the guarantee. Additionally, from a purely DP perspective, the privacy level is formally *set* (via $\varepsilon$), as we do over multiple values in experimental design, and thus does not need to be measured.
>
> **Thus, we have included a few privacy metrics mentioned in the papers that the reviewer cited (the metrics are scaled 0-100, higher value means stronger similarity): "Exact-match-proportion", "DCR", and "NNDR" from [Borisov et al. 2022, arXiv 2210.06280; Kotelnikov et al. 2023, arXiv 2209.15421]. See the table below, in response to the next metric ask comment.**
>
> We have added these empirical results alongside our statement of the theoretical guarantee in our revision, stressing how a negative attack result cannot strengthen privacy beyond $\varepsilon$‑DP, but that a positive result can be useful in revealing e.g. implementation bugs.
>
> We finally note that several influential DP tabular works (e.g., [Hod et al. 2024, arXiv 2405.00267; McKenna et al. 2022, arXiv 2201.12677; Liu et al. 2021, arXiv 2106.07153]) rely principally on the formal guarantee and do not report adversarial metrics. We are happy to include empirical metrics for completeness if the reviewer believes this is merited.
>
> > **W3: *“TVD and 3‑way marginals may be trivial; consider higher‑order metrics (TabDDPM, TabDiff)”***
>
> We appreciate this reviewers point; we’d like to highlight that we already compute 15 metrics spanning univariate, pair‑wise, and classifier‑based categories, and have many overlapping metrics with the cited papers; TVD and 3WM are simply the ones we highlighted in terms of *direct* distributional comparison, and are relatively standard when considering synthetic data [Tao et al. 2022, arXiv 2112.09238; Patki et al. 2016, IEEE 7796926]. We’d also like to note that TVD and 3-way marginals are not necessarily trivial; the TVD metric we report is *multi-dimensional* (see Appendix G.1; in other works it is given only along each dimension independently), and much prior work has shown how difficult it can be to approximate 3-way marginals accurately, and how they can be a proxy for synthetic data usefulness [e.g. Tao et al. 2022, arXiv 2112.09238].
>
> **That said, we added the alpha-precision/beta-recall metric from the synthcity package [Qian et al. 2023, arXiv 2301.07573; Alaa et al. 2021, arXiv 2102.08921], also mentioned in [Du et al. 2024, arXiv 2402.06806],** which additionally appear in some of the papers the reviewer cited. After review, we appreciate some of the nuances these metrics introduce into our comparisons, but do not believe they change the main takeaways of the paper. Here is an excerpt of the results on EDAD, comparing all of the different surrogate public data methods against the real, sensitive “Private” data.
>
> |Metric|Private|Public|Baseline (Domain)|Arbitrary|CSV(Claude 3.5 Sonnet)|CSV (GPT‑4o)|CSV (Llama 3.3 70B)|Agent (Claude 3.5 Sonnet, Unif.)|Agent (Claude 3.5 Sonnet, Max Cov.)|Agent (GPT‑4o, Unif.)|Agent (GPT‑4o, Max Cov.)|Agent (Llama 3.3 70B, Unif.)|Agent (Llama 3.3 70B, Max Cov.)|Agent (Unif.)|Agent (Max Cov.)|
> |-|-|-|-|-|-|-|-|-|-|-|-|-|-|-|-|
> |exact_match_prop|100.0|6.9|0.1|0.0|1.0|1.5|0.0|0.7|1.0|0.2|0.7|0.2|0.2|0.5|0.2|
> |dcr_5th_percentile|100.0|100.0|90.9|81.8|90.9|90.9|81.8|90.9|90.9|90.9|90.9|81.8|81.8|90.9| 90.9|
> |nndr_5th_percentile|100.0|100.0|50.0|33.3|33.3|50.0|33.3|50.0|50.0|50.0|50.0|33.3|33.3|50.0| 50.0|
> |alpha_precision|100.0|78.1|40.9|23.5|70.9|80.9|85.2|70.4|68.2|88.2|88.9|2.2|2.1|67.4|69.6|
> |beta_recall|100.0|50.8|31.0|12.9|13.5|30.0|3.5|42.8|40.0|44.2|43.6|8.1|8.8|37.6|36.3|
>
>
> We’d be happy to include the other results we’ve run e.g. on PrivBayes for EDAD with all the new metrics reported in this table for hyperparameter tuning differentiation (DCR, exact_match, etc.), if you would like to review them, although we are omitting them here for space and readability, as they are extensive.
>
> > **Q1: *“How do you ensure no / minimal data leakage when the LLM generates samples?”***
>
> We would be happy to clarify this and highlight these results in the main body of the paper, as we thought carefully about this exact issue. We were very careful about selecting two datasets (WE and EDAD) that were published *after* the model cutoff dates. Furthermore, we applied the verbatim‑memorization test suite of [Bordt et al. 2024, arXiv 2404.06209] and found zero evidence of row-wise leakage (unfortunately we only briefly mentioned these tests in Section 4.2 and deferred details to Appendix D, but would be happy to promote these results back into the main body of the paper).
>
> > **Q2: *“How to evaluate the SCM produced by the LLM?”***
>
> This is a great question! Functionally, we validate the SCM explicitly as a DAG; we use the NetworkX package to *compile* each SCM and check that it does not violate a-cyclicity and that all variables are represented as nodes. Then, our general evaluation is at a task‑level: we judge an SCM surrogate by how well it supports the auxiliary objective (pre‑training, parameter tuning, privacy/utility curve estimation). Nonetheless, the SCM itself could be probed, and this is an interesting direction of future work. We are aware of some existing work on structural plausibility and domain‑expert rated plausibility from the causal community [e.g. Kiciman et al. 2023, arXiv 2305.00050; Bynum et al. 2024, arXiv 2411.08019], and would be happy to discuss this work more at length.
>
> > **Q3: *“Why is the surrogate better for DP than a public dataset?”***
>
> We’d like to clarify this point here: *surrogate public data is not always better;* it should mainly be viewed as a public data *fallback* when a properly matched public sample is unavailable. Traditional public data, if available (and if from a sufficiently similar distribution), should generally be preferred, as our results show. Both the traditional public data and the surrogate public data incur zero privacy cost even if we subsequently share it; we will clarify this point in our preliminaries, thank you for raising it!

---

> > ### Comment · Reviewer_rgrp · 2025-08-06
> >
> > I appreciate the authors’ thorough and thoughtful responses. After carefully considering their clarifications, in conjunction with the insights offered by the other reviewers, I remain confident in my original assessment. I therefore maintain my overall positive evaluation of the paper.

---

### Official Review · Reviewer_Z8aK · 2025-06-18

**Rating:** 5
**Confidence:** 3

**Summary:**

The paper introduces *surrogate public data*—datasets synthesized solely from non-sensitive schema/metadata - so that no privacy budget is consumed when they are used inside DP workflows.
Two LLM-based generators are proposed:
- CSV: prompt an LLM to emit schema-conformant tabular rows.
- Agent: an automated state-machine that elicits from an LLM a plausible causal DAG and structural equations, then samples from the resulting SCM.

A comprehensive benchmark evaluates these surrogates on three DP auxiliary tasks: (1) pre-training DP tabular classifiers, (2) hyper-parameter tuning of DP synthetic-data generators, and (3) privacy-utility curve estimation. Experiments across three real tabular datasets and three DP synthesizers show that LLM-generated surrogates can match or exceed traditional public data for DP pre-training in low-data regimes and are competitive for tuning and trade-off estimation. Code is public.

**Additional Feedback:**

Could the authors comment on how their approach generalizes to continuous domains? For example, might a predetermined discretization be viable? This consideration is particularly important when prior knowledge is extremely limited [3].

[3] Ma Y, Zhang H, Cai Y, et al. Decision tree for locally private estimation with public data[J]. Advances in Neural Information Processing Systems, 2023, 36: 43676-43705.

**Dataset Code Accessibility:**

Yes

**Dataset Code Comments:**

Github link can be found in the paper.

**Ethical Considerations:**

No, there are no or only very minor ethics concerns

**Final Justification:**

The authors have generally addressed my questions. I hope the relevant discussions can be incorporated into the revised manuscript. Given my positive assessment, I will maintain my original score.

**Limitations Weaknesses:**

- All datasets only have categorical features; effect on mixed or continuous schemas remains unknown.
- Comparison with DPGAN based methods is missing, for instance [1,2].

[1] Xie L, Lin K, Wang S, et al. Differentially private generative adversarial network[J]. arXiv preprint arXiv:1802.06739, 2018.

[2] Kunar A, Birke R, Zhao Z, et al. Dtgan: Differential private training for tabular gans[J]. arXiv preprint arXiv:2107.02521, 2021.

**Strengths Contributions:**

- Public tabular data rarely align with private data distributions; replacing them with schema-driven surrogates is novel and practically valuable.
- Three concrete contributions are clearly stated.
- The algorithms are novel. Agent method cleverly turns causal-discovery prompts into executable Pyro code without touching sensitive records. CSV shows that few-shot prompting plus schema validation yields realistic rows.
- Three auxiliary tasks, three private datasets, multiple ε values, 10× runs per hyper-parameter setting, and Pareto analyses give a convincing empirical picture.
- Ablation on dataset size strengthens insights about when surrogate data help.
- The paper is well-structured; figures/tables are readable and captions self-contained. And I like the coloring the authors use.

---

> ### Author Rebuttal · Authors · 2025-07-30
>
> We appreciate the reviewer's constructive comments and are grateful that they found the surrogate-data idea, approaches, and the overall benchmark to be a valuable contribution.
>
> > **W1/Q1: *“All datasets only have categorical features; effect on mixed or continuous schemas remains unknown.”***
>
> We agree that the question of non-categorical features is an important one! Most DP synthetic data generators, and especially state-of-the-art, are designed to accommodate categorical data only [McKenna et al. 2022, arXiv 2201.12677; Rosenblatt et al. 2023, arXiv 2208.12700], and assume some preprocessing of binning continuous columns [Ganev et al. 2025, arXiv 2504.06923]. Note that the datasets we used already include binned continuous columns (e..g, age group, degree of disability in EDAD, or person's total income rank in ACS, etc.). We will clarify this point in the paper, and discuss this further in the limitation section.
>
> That said, we also note that the prompt template includes range constraints; when (for example) the schema says “float in [0,120]”, our experience is that when prompting directly for a CSV, Claude/GPT successfully emits decimals in that range, and we could keep these continuous values if the domain required it.
>
> In the case of the Agent, our state‑machine construction explicitly asks the LLM to choose a parametric family for numeric nodes as specified by the schema (e.g., Gaussian, Log‑Normal, Gamma, etc.) and to emit the full Pyro sampling code. This means that as we sample values, they are inherently continuous, and thus are easy to keep this way (even though we did not experiment with continuous values for the above stated reasons).
>
> Furthermore, a pre-determined discretization could indeed be a viable approach especially in areas where the prior is limited! One could imagine spending a small amount of privacy budget just to learn a reasonable level of discretization, to inform the schema setting. This is an interesting idea that merits exploration in our future work, thank you.
>
> > **W2: *“Comparison with DPGAN based methods is missing, for instance [1,2].”***
>
> We appreciate this reviewers’ point that we could include a method from the general class of DP-GANs for tabular synthetic data as part of the synthetic data experiments. We made this decision due to guidance from prior work; while in image domains, variants of DP-GANs can be effective, multiple studies in the tabular domain report that DP‑GANs lag behind marginal / graphical based approaches on utility and stability, especially for small‑to‑medium‑sized tables [Tao et al. 2022, arXiv 2112.09238].
>
> **That said, in order to demonstrate the extensibility of our benchmark and to provide a representative example of this class of algorithms, we re-ran our benchmark on DP Auxiliary Task 2 (hyperparameter tuning for DP synthetic data) on DP-GAN [Xie et al. 2018, arXiv 1802.06739] as suggested.** Here, for example, are the DP-GAN results on EDAD (both the full list of aggregate values and the subsequent pareto frontier analysis). We tuned over a grid of learning rates and batch sizes, two very important and often tuned hyperparams for GAN based methods.
>
> | Method  | Classification | Correlation | Marginals |
> |---|---|---|---|
> | Public   | 0.144  | 0.027  | 0.006  |
> | CSV (Claude 3.5 Sonnet) | 0.179  | 0.028  | 0.028    |
> | Agent (Max Cov.)   | 0.193 | 0.038   | 0.034    |
> | Agent (Claude 3.5 Sonnet, Max Cov.)  | 0.199  | 0.028  | 0.025  |
> | CSV (GPT‑4o) | 0.207   | 0.025   | 0.021   |
> | Agent (Claude 3.5 Sonnet, Unif.)    | 0.212   | 0.058  | 0.039   |
> | Agent (Unif.) | 0.262   | 0.034   | 0.015  |
> | Arbitrary   | 0.265  | 0.073 | 0.028   |
> | Baseline (Univariate) | 0.242  | 0.081   | 0.011  |
> | Baseline (Domain)  | 0.277  | 0.083  | 0.031  |
> | Agent (GPT‑4o, Unif.) | 0.295  | 0.057 | 0.027  |
> | CSV (Llama 3.3 70B) | 0.296   | 0.024 | 0.024 |
> | Agent (GPT‑4o, Max Cov.) | 0.351  | 0.032   | 0.028 |
>
> Then, the Pareto frontier of the above results; the methods below dominate for DP-GAN parameter tuning on EDAD.
> | Method   | Classification | Correlation | Marginals |
> |---|---:|---:|---:|
> | **Public**  |       0.144 |    0.027 |   0.006 |
> | **CSV (GPT‑4o)**  |  0.207 |    0.025 |   0.021 |
> | **CSV (Llama 3.3 70B)**  | 0.296 |    0.024 |   0.024 |
>
> Now that we have these results, we will of course add them to our paper along with a short discussion, but they do not change the overall takeaways on DP Auxiliary Task 2 (hyperparameter tuning).
>
> Thank you again for the insightful feedback, and for your overall positivity towards our work.

---

> > ### Comment · Reviewer_Z8aK · 2025-08-02
> >
> > The authors have generally addressed my questions. I hope the relevant discussions can be incorporated into the revised manuscript. Given my positive assessment, I will maintain my original score.

---

### Official Review · Reviewer_BqMK · 2025-06-30

**Rating:** 5
**Confidence:** 3

**Summary:**

The paper focuses on generation methods for surrogate data and benchmarks that assess the usefulness of the surrogate data. The authors define the surrogate data as data generated using metadata or schemas of sensitive data but independent of the actual sensitive data and argue that this surrogate data can be useful for auxiliary tasks related to differenttially private (DP) mechanisms (such as hyper-parameter tuning). The authors propose baselines and methods to generate the surrogate data and define three benchmark tasks to assess the usefulness of the methods. These are model pre-training for classification, hyper-parameter tuning and privacy-utility estimation for synthetic data generation.

**Additional Feedback:**

- Section 3: baseline description contradictory: "We evaluate three baseline methods [...] that rely solely on the public schema. [...] the Univariate approach samples columns independently using empirical 1-way marginal distributions from the private data.", would be good to rephrase the first sentence to reflect all baselines.
- Question Section 3: What exactly is the arbitrary method doing exactly? Is it using private data?
- Question: Obvious baseline missing: Spending privacy budget to obtain intervals of variables. Why are you not considering that baseline?
- Reference suggestion for Section 5.1+5.2: Your finding is related to Tobaben et al. [4] that look at the cost of few-shot image-classification and similarity between public pre-training and fine-tuning data under DP, it is a different domain but seems like a fitting reference here.
- General reference suggestion: Ganev et al. [2] look at the effect of doing domain extraction


[2] Ganev, G., Annamalai, M. S. M. S., Mahiou, S., & De Cristofaro, E. (2025). Understanding the Impact of Data Domain Extraction on Synthetic Data Privacy. arXiv preprint arXiv:2504.08254.

[4] Tobaben, M., Shysheya, A., Bronskill, J. F., Paverd, A., Tople, S., Zanella-Beguelin, S., ... & Honkela, A (2023). On the Efficacy of Differentially Private Few-shot Image Classification. Transactions on Machine Learning Research.

**Dataset Code Accessibility:**

Yes

**Dataset Code Comments:**

There is a github repository and the data is also uploaded there. I feel like everything is accessible but it takes a bit more work as huggingface or similar platform isn't used. The documentation could be improved to provide more detail but I think as the pre-processing scripts are there it would be possible to reproduce it.

**Ethical Comments:**

The work generally aims at improving the ethical use of AI through DP and as there is no additional data being released I see little ethical concerns. The Section 7 highlights potential issues in practice and I think addressees all my concerns.

**Ethical Considerations:**

No, there are no or only very minor ethics concerns

**Final Justification:**

I read all reviews and discussions. My main concern from the beginning with this paper is the very compressed writing and lack of details but the authors have promised to address that and all my other concerns and questions (and even added another baseline). I think Z8aK raised good points that the authors addressed. To my understanding the not addressed concerns of ZYaM and rgrp are minor while most points have been addressed.

**I will raise my score from 3 to 5 based on the promises of the authors to address these points. I trust that the NeurIPS process will keep the authors accountable for making the changes if the paper gets accepted.**

**Limitations Weaknesses:**

- The introduction takes a huge chunk of the paper (essentially 3 pages) which leaves very little space to discuss the findings. The Section 4 and 5 lacks the technical detail necessary to understand the findings. E.g., important details like the epsilon, delta budget, and other training or evaluation details are seemingly hidden in the Appendix.
- The reporting of results is quite compressed and hard to understand with many results hidden in the Appendix: The results are reported with repeats but without errorbars making it hard to understand if the differences are statistically significant. The display of results in Tables 1-3 is compressed using some "olympic medal highlighting" and a lot of acronyms. I would appreciate spending more space on the actual core results and their interpretation.
- The tasks are interesting but I get the feeling that they are only teasers or examples. It would be good to explain why these got selected and what other possible tasks there are. Together with the limited details mentioned earlier they read like illustrations.
 - Section 4: "after the training data cutoff of some of the LLMs we evaluate": This is written ambiguous and needs more clarification as claude seems to perform quite well but the private training data might be part of the training data of the LLM (See April 2024 in Tables 1 and 2). The authors should discuss this more..


I will recommend a borderline reject because the paper and idea are solid but the evaluation is limited and the result description and description of technical details are very compressed making it hard to follow these details. The AC has asked me to explain why it is not a 5 (I think especially the evaluation and description are lacking) and why it is not a 2 (I think the idea is solid). I would like to clarify that I am happy to change my rating based on the author responses.

**Strengths Contributions:**

- The topic is timely: The implications of using sensitive data for auxiliary tasks on the DP budget have been explored lately (e.g., for hyperparameter tuning [e.g., 1] and for the domain extraction for DP synthetic data [2]) and the gap between pre-training and fine-tuning data [3. Also there are settings where actual public data might be not useful or not available [3, 4]
- The introduction and motivation are written very clearly, but take a lot of space.
- There are some studies that go more into detail.


[1] Papernot, N., & Steinke, T. Hyperparameter Tuning with Renyi Differential Privacy. In ICLR 2022

[2] Ganev, G., Annamalai, M. S. M. S., Mahiou, S., & De Cristofaro, E. (2025). Understanding the Impact of Data Domain Extraction on Synthetic Data Privacy. arXiv preprint arXiv:2504.08254.

[3] Tramèr, F., Kamath, G., & Carlini, N. Position: Considerations for Differentially Private Learning with Large-Scale Public Pretraining. ICML 2024.


[4] Tobaben, M., Shysheya, A., Bronskill, J. F., Paverd, A., Tople, S., Zanella-Beguelin, S., ... & Honkela, A (2023). On the Efficacy of Differentially Private Few-shot Image Classification. Transactions on Machine Learning Research.

---

> ### Author Rebuttal · Authors · 2025-07-30
>
> We appreciate the constructive feedback, thank you for taking the time to review our work; we hope to address your concerns below, inline.
>
> > ***"The topic is timely [...] The introduction and motivation are written very clearly."***
>
> We thank the reviewer for recognizing the clarity of our motivation and for highlighting the relevance of addressing the DP budget implications for auxiliary tasks. We appreciate the references you provided and agree that they are useful inclusions in our work!
>
> > **W1: *“The introduction takes a huge chunk [...] details like epsilon, delta budget, and other training or evaluation details are hidden in the Appendix.”***
>
> We appreciate this feedback and agree that while our intention was to thoroughly motivate the problem and our approach, the introduction could be condensed further. We would be happy to shorten this section in a minor revision and relocate key details, such as specific DP hyperparameters ($\varepsilon$, $\delta$) and training settings, from the appendix into the main body. We did not intend to hide these details; we chose standard, straightforward settings, and we agree that highlighting them earlier will improve clarity.
>
> > **W2: *“The reporting of results is compressed [...] Tables 1-3 are compressed.”***
>
> We are sorry if some of our presentation made the results hard to follow, and are happy to address this issue in a revision. Our approach was to focus on highlighting core findings in the main body, and to relegate more comprehensive details to the appendix due to space constraints (this seems like a common challenge with these extensive datasets and benchmarks papers, where space for results is at a premium).
>
> Regarding the tables, we note that Tables 1-3 do not use acronyms but rather what we believed were interpretable abbreviations (e.g., Corr. for correlation). That said, we understand that either spelling out the exact metrics or expanding on the abbreviations more clearly in our table captions could enhance readability, and will do so in our revision. We’d also be happy to switch from the olympic medal convention to a type face highlighting that might be more standard. Finally, we did include error bars in some of our figures / tables (e.g. Figure 1), and have the standard errors for all our results. We had removed them from some figures to aid readability (as they don’t make a big difference to the takeaways), but would be happy to add them back in.
>
> > **W3: *“The tasks are interesting but [...] read like illustrations.”***
>
> We apologize if the motivation for selecting these tasks was unclear. **We are convinced that the DP tasks of hyperparameter tuning, model pre-training, and privacy-utility trade-off estimation are significant, real-world challenges in deploying differentially private models,** as highlighted by the recent practical guide of [Ponomareva et al.; How to DP-fy ML: A Practical Guide to Machine Learning with Differential Privacy] and other DP-focused studies [hyperparameter tuning e.g. Papernot et al. 2021, arXiv 2110.03620; pre-training  e.g. Tang et al. 2023, arXiv 2306.06076; privacy budget tuning e.g. Cummings et al. 2024, arXiv 2406.12103]. We would be glad to clarify this in the revision, explicitly citing these references (and others) to underscore the practical importance of each task.
>
> > **W4: *“Claude seems to perform quite well but the private training data might be part of the training data of the LLM (April 2024).”***
>
> Ah, this is our bad with the presentation - we appreciate the reviewer raising this point! We would like to clarify explicitly that Claude 3.5 Sonnet’s training cutoff date was reported as “up until April, 2024” [see Anthropic article 8114494, “How up-to-date is Claude's training data?”] whereas both the EDAD (via Instituto Nacional de Estadística) and Workplace Equity Survey (WE) (via ICPSR) datasets used as private splits were publicly released on April 30, 2024. We interpreted “up until” as excluding April, but we will update the manuscript to explicitly state these exact dates, citing official publication sources to avoid any confusion. At worst, there is a single day overlap, and we believe it is thus unlikely there is much if any bleeding of the data, but we will add a footnote discussing all of this for due diligence.
>
> Additionally, we highlight that all our results hold for the other LLMs we evaluated, which both had cutoff dates significantly before April 30, 2024 (October 2023 for GPT-4o and December 2023 for Llama 3.3 70B IT). We also conducted extensive memorization tests as referenced in Section 7 and Appendix D.2.4 for all our datasets.
>
> > ***“the paper and idea are solid but the evaluation is limited and the result description and description of technical details are very compressed making it hard to follow these details”***
>
> We are sorry that you found some elements of the presentation hindered your clearer assessment of the work, despite you finding the core idea of the paper to be solid. We thank you for the constructive feedback and will implement the suggested revisions, clarifying technical details, improving results readability, and condensing the introduction (as well as adding some suggested results from other reviewers).
>
> > ***“It takes a bit more work as HuggingFace or similar platform isn't used.”***
>
> We’d be happy to release the datasets on HuggingFace and update the repository accordingly, and appreciate the push to do so.
>
> > **Q1: *“Baseline description contradictory [...] would be good to rephrase.”***
>
> We agree this is confusingly worded as stated, and will update to clarify that only the “Uniform” and “Arbitrary” baselines rely exclusively on the schema, while “Univariate” baseline explicitly uses the noisy empirical 1-way marginal distributions from the private data as a point of comparison.
>
> > **Q2: *“What exactly is the arbitrary method doing? Is it using private data?”***
>
> Apologies if this was confusing, we deferred a more in depth description of “Arbitrary” to Appendix C.1.3. *The “Arbitrary” baseline does **not** use private data.* Instead, it constructs a random Bayesian network by randomly generating a DAG structure with a predefined maximum parent-degree (here we set it to 5). Each node's conditional probability distribution is sampled independently from a symmetric Dirichlet distribution, resulting in structured yet random dependencies among variables, entirely independent from the true private distribution. Other reviewers asked for an extended description on this as well, and we will be sure to promote our detailed description to the main paper body in our revision.
>
> > **Q3: *“baseline missing: Spending privacy budget to obtain intervals of variables?”***
>
> To clarify this suggestion: Do you mean allocating a portion of the privacy budget to privately estimate variables' ranges or intervals, and then using the remainder to fit the final DP mechanism? If so, we would be very happy to run this as a baseline and include those results in our revision.
>
> > **Q4: *“Your finding is related to Tobaben et al.”***
>
> You’re correct to point out that our finding is related to Tobaben et al. (2023), thank you for highlighting and we will add a discussion to the revision.
>
> For the benefit of the AC: Tobaben et al. (2023) evaluate how similarity between pretraining and downstream datasets impacts few-shot DP classification, and show that greater similarity is important to good private performance, particularly when data is scarce. The major difference is that their work focuses on images, while our findings focus on tabular data, which has other challenges as we highlight in our paper and in our responses to reviewers (conceptually they are related in exploring the similarity between surrogate and real datasets regardless of domain). We will add this comparison explicitly alongside other public data approaches we discuss from the image domain and include this citation, thank you.

---

> > ### Comment · Reviewer_BqMK · 2025-08-01
> >
> > Thanks for your replies and answering to my questions.
> >
> > In general I believe that your plan for revising sounds good and that it will make the paper better, I will keep observing your discussion with the other reviewers but in general I am willing to raise my score.
> >
> > > To clarify this suggestion: Do you mean allocating a portion of the privacy budget to privately estimate variables' ranges or intervals, and then using the remainder to fit the final DP mechanism? If so, we would be very happy to run this as a baseline and include those results in our revision.
> >
> > Yes, that is what I was after.

---

> > > ### Author Response · Authors · 2025-08-04
> > >
> > > We’re very glad that you are open to raising your score, given our plan for revision! Apologies for the delay in our response, it took some time but we have gone ahead and run the baseline you suggested as well.
> > >
> > > We’ve called this baseline “DP Marginal Estimation.” We allocate a portion of the privacy budget (here, we use an allocation factor of 30% of privacy budget) to privately estimate each column’s domain (e.g. via range inference for ordinal columns) and its histogram. We use the very nice Tumult (tmlt.analytics) package to track budget and ensure valid DP guarantee with these measurements; we use ApproxDPBudget queries for bounds and group counts and thus produce DP marginals. We then construct noisy PMFs per column from those marginals and generate a synthetic dataset of the same size by sampling each column. We can run this along $\epsilon \in \\{1, 4, 8, 16\\} $ (i.e. $\epsilon \in \\{0.7, 1.2, 2.4, 4.8\\}$ respectively for the surrogate data DP marginal estimation step). We will include this baseline, its precise description, and its empirical results in the revision, thank you again for the suggestion. Here, as an example, we give results for EDAD:
> > >
> > > | $\epsilon$ | Pretrain AUC | (Post)-Private Fine-tuning AUC |
> > > | -- | -- | -- |
> > > | 1   | 0.494  | 0.512 |
> > > | 4   | 0.495  | 0.729 |
> > > | 8   | 0.518  | 0.731 |
> > > | 16  | 0.491 | 0.767 |
> > >
> > > Perhaps as expected, the baseline performs about as well (perhaps slightly better) as the other baselines we measured in our paper (in that, it doesn’t help with the pre-training classification performance). It only captures noisy one-way marginals but lacks inter-feature correlations or higher-order structure that truly aids pre-training. So, without any cross-column dependencies, its ability to help initialize the model weights before private fine-tuning is limited.
> > >
> > > Thank you again for engaging with us and our paper during this rebuttal period, and for the constructive feedback.

---

> > > > ### Comment · Reviewer_BqMK · 2025-08-06
> > > >
> > > > Thanks for taking the time for running the suggested baseline and your analysis.
> > > >
> > > > I read the rebuttal that you provided to all reviewers again and I trust that you include all your promised changes in the paper and will raise my score to accept (5). I think my main issue with the paper in the beginning was the very compressed presentation that you promised to address. The review of Z8aK highlighted that to me and it seems that you have addressed the reviewers concerns.

---

### Official Review · Reviewer_ZYaM · 2025-07-21

**Rating:** 4
**Confidence:** 4

**Summary:**

This paper presents a method for creating synthetic datasets using large language models (LLMs), based only on publicly available schema-level metadata. The goal is to support differential privacy (DP) tasks in situations where real public tabular data are missing or insufficient. The authors describe how to generate this synthetic data with LLMs and show, through benchmark tests, that it can perform as well as or better than real public data when used for model pretraining.

**Dataset Code Accessibility:**

Yes

**Ethical Considerations:**

No, there are no or only very minor ethics concerns

**Final Justification:**

Thank you for addressing most of the concerns and providing additional experiments for the other reviewers. I will raise my score and hope to see all revisions incorporated into the final version

**Limitations Weaknesses:**

1. While public tabular data may be harder to access than text or image data, the paper does not clearly articulate the unique challenges of generating **private** tabular data as compared to those other domains. It also lacks comparative results showing why methods used for text or image data generation cannot be directly applied to tabular settings.

2. The proposed method relies on clean, well-defined metadata, which may not be available or reliable in real-world datasets, especially those that are proprietary or messy.


3. Although the method is tested across multiple DP tasks, the most notable improvements are observed on relatively small datasets (e.g., WE, EDAD). It remains unclear how well the approach would transfer to larger-scale or alternative DP settings, such as federated or local DP pipelines.

4. An arbitrary baseline performs unexpectedly well in the experiments (see Figure 3/Table 3), which weakens the central claim that LLM-generated surrogate data is clearly superior.

**Strengths Contributions:**

1. The paper tackles a key challenge in deploying differential privacy (DP) for tabular data—the lack of accessible public datasets, particularly in sensitive fields like healthcare and finance.

2. It uses advanced large language models (LLMs) to automatically generate two types of synthetic datasets: (a) CSV-style tabular data and (b) structured causal models (Agent), in a systematic and scalable manner.

3. The surrogate datasets are rigorously evaluated on three different DP auxiliary tasks using three real-world tabular datasets and strong performance metrics.

4. The authors provide comprehensive reproducibility details, including open-source code, dataset sources, LLM configurations, memorization safeguards, and hyperparameter settings.

---

> ### Author Rebuttal · Authors · 2025-07-30
>
> Overall, we thank the reviewer for their time and attention! Below, we address their comments inline.
>
> > ***The paper tackles a key challenge ... provides comprehensive reproducibility details …***
>
> We thank the reviewer for recognizing (i) the practical importance of closing the “public‑data gap” for tabular DP pipelines, (ii) our consideration of two LLM‑based surrogate‑data generators, and (iii) the breadth of the experimental design (three auxiliary tasks x three datasets x three synthesizers x full hyper‑parameter sweeps with ten seeds each, etc.). We especially appreciate the acknowledgement of our memorization safeguards (Section 4.2, App. D) and the open‑sourced code/LLM prompts, which we hope will make our approaches and benchmark immediately reusable!
>
> > **W1: *“The paper ... does not clearly articulate the unique challenges of generating private tabular data as compared to those other domains, nor show results that text/image techniques cannot be directly applied.”***
>
> Thank you for bringing this up, we are sorry that this was not sufficiently clear. We call out the distinction in the introduction, where we discuss how tabular data lacks strong cross‑domain priors; each survey or EHR schema effectively defines a new high‑dimensional space whose categorical codes rarely align with any standard “foundation” dataset. In contrast, ImageNet or Common Crawl can safely be used for pretraining or other “public data” tasks. Again, we are happy to clarify this in a minor revision!
>
> We also want to clarify why we do not provide cross domain results; our contribution is not a new generative model but **a way to construct in‑domain surrogates when no real public sample exists.** Thus, the right analogous methods to ours from the image domain would be something like synthetic pre-training images generated by StyleGan [Baradad et al. 2021, arXiv 2106.05963] , which have been used for state-of-the-art pre-training results for DP-SGD on CIFAR10, ImageNet, etc [Tang et al. 2023, arXiv 2306.06076].
>
> One could view our “Arbitrary” baseline (the randomly initialized Bayesian net approach) analogously; construct a valid within-domain dataset that has random (but structural) relationships between the features, and see if it’s useful. But in general, benchmarking methods for image / text data on a mismatched modality does not inform the central question “Can we remove the need for matching public data in the tabular domain?” which is the focus of our work. Finally, we’d like to point out that if you are concerned about how our problem statement interacts with standard methods for generating synthetic images, **we have included some results at the request of Reviewer Z8aK on DP-GAN [Xie et al. 2018, arXiv 1802.06739], which is a common approach for generating DP synthetic images that also works in the tabular domain.** We will make all this scope clearer at the end of our introduction in our revision, with a version of the discussion given here, thank you.
>
> > **W2: *Reliance on “clean” metadata. “The method relies on clean, well‑defined metadata, which may not exist in messy proprietary datasets.”***
>
> We appreciate this concern, and agree that we could have done a better job of discussing it. **Our approach needs only three simple fields per column: (1) name, (2) one‑line description, and (3) a list of categories or a plausible numeric range.** This minimal schema is routinely available for social‑science surveys (ICPSR, UK Data Service) and for medical data sets using FHIR or OMOP. Even if exact ranges are unknown, users can supply loose bounds (e.g., ages 0‑120); our validation layer enforces them. We do agree that there are tabular domains where this approach would likely not work due to missing reasonable schema/metadata (e.g. extremely noisy sensor logs, where the domain is likely not in the LLM prior), and are happy to clarify this in the limitations section of our paper, thank you.
>
> > **W3: *Generality beyond small datasets / other DP settings. “Notable improvements are on small datasets; unclear performance on larger or federated/local DP pipelines.”***
>
> We agree that the improvements on small datasets for the pretraining result were most notable, and we discussed why in Sections 5.1 and Figure 4 where we study the larger ACS dataset (~23,000 rows): **when we down‑sample ACS to ~5% (around the same size as the WE and EDAD datasets), the surrogate public data pretraining gains re‑emerge, confirming the well‑known inverse relationship between dataset size and the relative magnitude of noise necessary to satisfy DP.** Meaning, it is more technically challenging to provide a DP guarantee on a smaller dataset while still achieving “good enough” utility. We would argue, however, that sensitive real‑world tables -- like hospital cohorts, municipal registries, targeted surveys, etc. -- often sit in the ~100 to ~5000 row regime, where the surrogate public data would clearly be beneficial for e.g. pre-training for classification.
>
> Finally, because the surrogate generation is independent of the downstream DP mechanism, we would like to highlight that it could be applied to federated / local DP, though testing those mechanisms was not our focus in this paper. We’d be happy to add a comment on this direction for future work, as we think it's interesting!
>
> > **W4: *“Arbitrary” baseline looks strong. “An arbitrary baseline performs unexpectedly well in Figure 3/Table 3, weakening the claim of surrogate superiority.”***
>
> We agree that the strength of the “Arbitrary” baseline (a randomly initialized Bayes net approach for generating in-domain data with spurious inter-variable relations relationships) was surprising, but only on the hyperparameter tuning task (Section 5, DP Auxiliary Task 2, and Appendix E.2). It was substantially worse for (1) pretraining (Section 5, Figures 3A / 3B) and was not significantly better for (2) privacy-utility tradeoff (Appendix Section E.3).
>
> One way we thought about why it might be reasonable for the task of hyperparameter tuning for DP synthetic data is as follows. For PrivBayes, which noisily selects a Bayesian network (BN) structure, may benefit from data drawn from a synthetic BN (even a randomly initialized one, as is the case with “Arbitrary”), which could yield a coarse but still informative proxy for which k‑way interactions matter, thus guiding the choice of hyperparameters. **However, “Arbitrary” certainly fails to model the true marginal distributions accurately, which may be important for e.g. classification [Tao et al. 2022, arXiv 2112.09238; Vietri et al. 2022, arXiv 2209.07400], and so the benefit disappears for the pre-training classification task, as we show.** We’d be happy to add a clarification on this point in a revised version of our paper.

---

> > ### Comment · Reviewer_ZYaM · 2025-08-06
> >
> > Thank you for addressing most of the concerns and providing additional experiments for the other reviewers. I will raise my score and hope to see all revisions incorporated into the final version.

---

### Decision · Program_Chairs · 2025-09-18

**Decision:**

Accept (poster)

**Comment:**

After a detailed discussion with the authors, all reviewers recommend acceptance. The topic of the paper is central for NeurIPS.

A note for the authors: Only two out of 119 references are from before 2016, but new ideas can be found in old papers. See in particular Mach Learn (2013) 93:163–183, "Differential privacy based on importance weighting" which uses the same notion of surrogate public data as here (data generated independently of sensitive data, synthetic or real, consuming no privacy budget) and shows how the surrogate data can actually replace the sensitive data for downstream use.